# PACER: Acyclic Causal Discovery from Large-Scale Interventional Data

Ramon Viñas Torné [1]  Sílvia Fàbregas Salazar [1]  Soyon Park [1]  Ivo Alexander Ban [1,2]  Artyom Gadetsky [1]
Nikita Doikov [3]  Maria Brbić [1]

## Abstract

Inferring the structure of directed acyclic graphs (DAGs) is a central challenge in causal discovery. While interventional data can improve identifiability, existing methods remain limited by soft acyclicity constraints, leading to optimization over invalid cyclic graphs, numerical instability, and reduced scalability. We introduce PACER (Perturbation-driven Acyclic Causal Edge Recovery), a scalable framework for causal discovery that guarantees *acyclicity by design*. PACER defines a distribution over DAGs through a joint model of variable permutations and edge probabilities, enabling direct optimization over valid causal structures without surrogate penalties. It supports a likelihood-based treatment of observational and interventional data, flexible conditional densities, and structural prior knowledge. For linear-Gaussian mechanisms, we derive a closed-form expression for the expected interventional log-likelihood, yielding substantial computational gains. Empirically, PACER matches or exceeds state-of-the-art methods on protein signaling and large-scale genetic perturbation benchmarks, scaling to thousands of variables and achieving up to two orders of magnitude speedups over penalty-based approaches. These results demonstrate that exact and scalable causal discovery from high-dimensional perturbation data is achievable through principled search space design.

## 1. Introduction

Inferring the structure of directed acyclic graphs (DAGs) from data is a fundamental task in causal discovery. While observational data can only identify causal graphs up to Markov equivalence, the integration of interventional data can substantially improve discovery accuracy by resolving causal ambiguities that are otherwise indistinguishable (Hauser & Bühlmann, 2012). This is particularly relevant in domains like genomics, where high-throughput perturbation assays, such as single-cell Perturb-seq screens (Dixit et al., 2016), now enable the collection of large-scale experimental interventions across thousands of variables. However, as the dimensionality of these systems grows, the computational cost of structure learning becomes a major bottleneck.

Modern differentiable causal discovery has attempted to explore the super-exponential search space of directed acyclic graphs by recasting structure learning as a continuous optimization problem. A prominent strategy involves utilizing matrix-based characterizations of acyclicity, enforced through augmented Lagrangian penalties (Zheng et al., 2018; Brouillard et al., 2020; Lopez et al., 2022). Although these frameworks are more scalable than traditional greedy or combinatorial searches, they exhibit inherent structural and computational limitations. In particular, since acyclicity is only enforced as a soft penalty, these methods evaluate and optimize over cyclic configurations that lack a well-defined joint likelihood. Moreover, as the number of nodes increases, the computational complexity of standard acyclicity constraints often leads to numerical instability (Nazaret et al., 2024) and prohibitive runtimes (Bello et al., 2022).

Here, we introduce PACER (**P**erturbation-driven **A**cyclic **C**ausal **E**dge **R**ecovery), a scalable causal discovery framework that directly searches over the space of DAGs using observational and interventional data. Unlike previous approaches that relax acyclicity during optimization, PACER guarantees acyclicity by construction. The key features of PACER are:

- **Acyclicity by design.** We introduce a distribution over DAGs to simultaneously model topological orderings and filter causal edges. This formulation ensures that every structure evaluated during optimization is strictly acyclic, eliminating the need for expensive acyclicity terms and ensuring that the search remains within the space of DAGs throughout the entire learning process.

- **Flexible differentiable causal discovery.** PACER provides a unified likelihood-based framework that jointly leverages observational and interventional data. This

---

[1] Swiss Federal Technology Institute of Lausanne (EPFL), Switzerland [2] ETH Zurich, Zurich, Switzerland [3] Cornell University, USA. Correspondence to: Maria Brbić <mbrbic@epfl.ch>.

*Proceedings of the 43rd International Conference on Machine Learning*, Seoul, South Korea. PMLR 306, 2026. Copyright 2026 by the author(s).

approach supports a variety of conditional density models, including neural parameterizations and distributions tailored to specific data modalities.

- **Exact gradient computation.** We derive a closed-form expression for an interventional log-likelihood objective under linear-Gaussian mechanisms. This analytic gradient enables PACER to scale to thousands of variables, achieving up to a two-order-of-magnitude speedup over penalty-based differentiable methods.

- **Inductive biases and prior knowledge.** PACER is a flexible framework that supports the direct integration of structural priors, such as node centrality expectations or transcription factor binding constraints when modeling genomic regulatory networks.

We evaluate PACER on observational and interventional data derived from protein signaling networks and large-scale genetic perturbation benchmarks. Our results demonstrate that PACER matches or exceeds the performance of state-of-the-art causal discovery methods. Crucially, it enables causal discovery in networks with thousands of variables, achieving a runtime decrease of up to two orders of magnitude over penalty-based differentiable methods. PACER is a versatile causal discovery framework that efficiently and effectively reconstructs causal graphs from high-dimensional interventional data.

## 2. Related work

Causal discovery is traditionally framed as a discrete search over an exponential space of DAGs. We position PACER within the landscape of modern causal discovery by contrasting it with permutation-based searches, differentiable acyclicity relaxations, and frameworks for causal discovery from interventional data.

**Permutation-based causal discovery.** A long-standing line of work exploits the correspondence between DAGs and topological orderings of variables. A prominent example is the Sparsest Permutation (SP) principle, which searches for the permutation that yields the sparsest DAG consistent with the data (Raskutti & Uhler, 2018). Practical methods perform greedy or combinatorial searches over orderings (Singh & Moore, 2005; Silander & Myllymäki, 2006; Lam et al., 2022) or estimate an ordering followed by edge selection, as in CAM (Bühlmann et al., 2014). These methods guarantee acyclicity by construction, but rely on heuristic sparsification or local decompositions, and return a single graph, thereby limiting their ability to represent structural variability. Moreover, their computational cost scales super-exponentially in the number of variables (in the worst case $\mathcal{O}(n!)$), limiting scalability. In contrast, PACER employs a probabilistic representation of orderings based on the Plackett–Luce parameterization (Luce, 1959; Plackett, 1975; Gadetsky et al., 2020) and jointly learns edge probabilities. This yields a flexible distribution over DAGs that captures structural variability while maintaining acyclicity. Compared to classical permutation search, PACER trades off exact enumeration in favor of stochastic optimization, gaining substantial scalability (quadratic rather than factorial complexity).

**Differentiable causal discovery.** Recent advances have recast DAG learning as a continuous problem, a shift largely driven by NOTEARS (Zheng et al., 2018). This approach introduces a smooth characterization of acyclicity, with extensions to nonlinear mechanisms (Zheng et al., 2020) and neural parameterizations (Lachapelle et al., 2020; Bello et al., 2022). However, the scalability of these methods is fundamentally hindered by the complexity of the acyclicity constraint and its gradients, often leading to numerical instability as the number of variables increases (Nazaret et al., 2024). Furthermore, these frameworks rely on augmented Lagrangian penalties that evaluate cyclic graphs during training and necessitate sensitive hyperparameter tuning and post-hoc thresholding. In contrast, PACER avoids surrogate regularizers by operating directly in the space of permutations, ensuring acyclicity by construction and eliminating the need for post-hoc thresholding.

**Modeling DAGs via permutations of triangular adjacency matrices.** A growing body of work addresses the combinatorial challenge of DAG discovery by learning distributions over variable orderings. Differentiable approaches such as DP-DAG (Charpentier et al., 2022), BCD Nets (Cundy et al., 2021), BCNP (Dhir et al., 2025), and BayesDAG (Annadani et al., 2023) learn distributions over orderings but typically rely on $\mathcal{O}(n^3)$ operators, limiting scalability in high-dimensional settings. Moreover, these approaches primarily focus on observational data. In contrast, PACER introduces a more computationally efficient parameterization, effectively scaling to thousands of variables, and can handle interventional data within a single likelihood-based objective.

**Causal discovery from interventional data.** Interventional data improves identifiability and has been incorporated into both discrete and differentiable frameworks. Traditional approaches extend classical frameworks to the interventional regime, including score-based methods like GIES (Hauser & Bühlmann, 2012) and IGSP (Wang et al., 2017), which utilize invariance principles to orient edges. However, these methods face scalability challenges and are susceptible to local optima due to their reliance on greedy heuristics. More recent differentiable approaches, such as DCDI (Brouillard et al., 2020), DCDFG (Lopez et al., 2022), and ENCO (Lippe et al., 2022) jointly model observational and

interventional data using neural likelihoods. However, these strategies permit cyclic configurations during optimization, which can lead to ill-defined joint distributions, and require post-hoc pruning to recover valid acyclic structures. PACER instead maintains a valid DAG factorization throughout training, enabling efficient likelihood-based learning from interventional data without auxiliary constraints.

## 3. Background

**Differentiable causal discovery from interventional data.** DCDI (Brouillard et al., 2020) and DCDFG (Lopez et al., 2022) optimize an interventional likelihood-based objective $S(\Theta, \Omega)$ subject to an acyclicity constraint $\mathcal{C}(\mathbb{E}[M(\Theta)])$:

$$\max_{\Theta, \Omega} \quad S(\Theta, \Omega) \quad \text{such that} \quad \mathcal{C}(\mathbb{E}[M(\Theta)]) = 0, \quad (1)$$

where $\Theta$ parameterizes a distribution $M(\Theta)$ over adjacency matrices and $\Omega$ denotes the conditional distribution parameters of each variable conditioned on its parents. The constraint $\mathcal{C}(\cdot) = 0$ ensures that the expected adjacency matrix corresponds to an acyclic graph, using a differentiable penalty function based on either the spectral radius or the matrix exponential (Zheng et al., 2018; Lopez et al., 2022). The score $S(\Theta, \Omega)$ is defined as:

$$\mathbb{E}_{M' \sim M(\Theta)} \Big[ \sum_{r=1}^{R} \mathbb{E}_{X \sim P_{\text{data}}^{(r)}} \sum_{j \notin \mathcal{I}_r} \log p_{\Omega}^j(X_j | M'_j, X_{-j}) \Big] \\ - \lambda \|\mathbb{E}[M(\Theta)]\|_1. \quad (2)$$

This score reflects the expected log-likelihood of non-intervened variables $j \notin \mathcal{I}_r$ across all $r \in \{1, ..., R\}$ intervention regimes, regularized by an $\ell_1$-penalty on the expected adjacency matrix to encourage sparse networks. Here, $P_{\text{data}}^{(r)}$ denotes the distribution of data points $X$ under regime $r$, and $p_{\Omega}^j$ represents the conditional density model for variable $X_j$, given its parent variables $X_{-j}$ as specified by the sampled adjacency matrix $M'_j \sim M(\Theta)$.

**Assumptions.** We operate under the following assumptions. First, we assume causal sufficiency, *i.e.*, there are no unobserved common causes (latent variables) influencing the system. Second, we consider stochastic perfect interventions, where the intervention targets a specific variable (or multiple variables) by modifying its conditional distribution without necessarily fixing it to a constant value. Third, we impose no global restrictions on the domains of the variables: the framework is functionally agnostic and can accommodate discrete, continuous, or mixed data by selecting appropriate likelihood functions. Finally, we assume that the underlying causal structure is a directed acyclic graph (DAG). Importantly, our approach returns a member from the interventional Markov equivalence class (Hauser

& Bühlmann, 2012), *i.e.*, the output should be treated as an element from the set of plausible structures rather than a single definitive graph.

## 4. PACER

We introduce PACER (Perturbation-driven Acyclic Causal Edge Recovery), a scalable causal discovery method applicable to large-scale interventional data with thousands of variables (Figure 1). The key idea in PACER is to explore the search space of DAGs by design, without the need for expensive and numerically unstable regularization terms to enforce acyclicity. PACER supports learning with structural priors over the graph, for example biologically informed constraints to guide causal graph discovery.

### 4.1. Exploring the space of DAGs by design

DCDI (Brouillard et al., 2020) and DCDFG (Lopez et al., 2022) search over the entire space of graphs using an augmented Lagrangian procedure coupled with a computationally expensive regularization term to encourage acyclicity. This presents a number of problems: (*i*) the acyclicity constraint is only softly enforced, (*ii*) the likelihood-based objective is evaluated on directed cyclic graphs, and (*iii*) a final pruning step is required to obtain a valid DAG, which may potentially discard a large fraction of edges.

PACER instead restricts the search space by *directly modeling the distribution of DAGs*. The key insight is that any DAG admits a representation as a lower-triangular adjacency matrix, and that permuting the variables within such a matrix also yields a valid DAG. Therefore, one can search the space of DAGs by (*i*) modeling a distribution over permutations of variables, which specifies a complete DAG, and (*ii*) pruning superfluous edges.

**Modeling topological orderings.** We first introduce the Plackett-Luce distribution (Luce, 1959; Plackett, 1975), which models the partial ordering of variables in a DAG, and then extend it to the Bernoulli-Plackett-Luce model, which PACER uses to specify a full distribution over DAGs.

**Definition 4.1** (Plackett-Luce distribution). Let $\pi = (\pi_1, \ldots, \pi_n)$ denote a permutation of $n$ items, where $\pi_k$ is the item ranked at position $k$. Each item $i \in \{1, \ldots, n\}$ is associated with a parameter $\theta_i \in \mathbb{R}$. The probability of observing permutation $\pi$ under the Plackett-Luce model is:

$$p(\pi|\theta) = \prod_{k=1}^{n} \frac{e^{\theta_{\pi_k}}}{\sum_{u=k}^{n} e^{\theta_{\pi_u}}}.$$

Intuitively, a permutation $\pi = (\pi_1, \ldots, \pi_n)$ from a Plackett-Luce distribution can be viewed as a sequence of categorical draws, where each draw excludes the variables that have

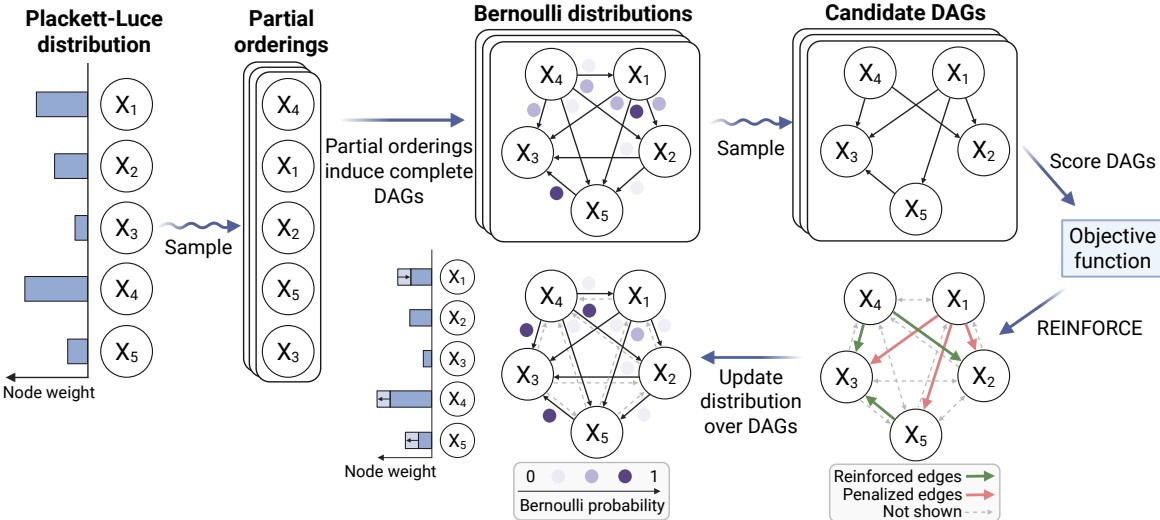

*Figure 1.* Overview of the framework. PACER models a topological ordering of variables using a Plackett-Luce distribution. Nodes with higher weight are more likely to precede nodes with lower weight in downstream DAGs. Samples from this distribution induce complete DAGs, which are further filtered via samples from independent, edge-specific Bernoulli distributions. This defines our Bernoulli-Plackett-Luce distribution over DAGs. At train time, we sample multiple candidate graphs and score them based on a likelihood-based objective function. We then optimize the parameters of the Bernoulli-Plackett-Luce model using REINFORCE gradient updates (Williams, 1992).

already been sampled. Specifically, $\pi_1$ is sampled from a categorical distribution with logits $\theta$; $\pi_2$ is then sampled from the same distribution with $\theta_{\pi_1}$ removed; and the process continues until all variables are generated.

**Pruning superfluous edges.** To prune edges, we define a binary mask $\boldsymbol{B} = (b_{ij}) \in \{0,1\}^{n \times n}$ with independent entries sampled from Bernoulli distributions with parameters $\boldsymbol{P} = (p_{ij}) \in [0,1]^{n \times n}$:

$$p(\boldsymbol{B}|\boldsymbol{P}) = \prod_{i=1}^{n} \prod_{j=1}^{n} p_{ij}^{b_{ij}} (1 - p_{ij})^{1-b_{ij}}.$$

**Modeling a distribution over DAGs.** Combining the permutation and edge mask yields our Bernoulli–Plackett–Luce distribution, constructed from conditionally independent Bernoulli distributions and a Plackett–Luce model.

**Definition 4.2** (Bernoulli-Plackett-Luce distribution). The Bernoulli-Plackett-Luce distribution with parameters $\theta \in \mathbb{R}^n$ and $\boldsymbol{P} \in [0,1]^{n \times n}$ models a distribution over DAGs $(\pi, \boldsymbol{B})$ and has probability mass function $p(\pi, \boldsymbol{B}|\theta, \boldsymbol{P}) = p(\pi|\theta)p(\boldsymbol{B}|\boldsymbol{P})$. The probability that an edge $a_{ij}$ belongs to the DAG is:

$$p(a_{ij} = 1|\theta, \boldsymbol{P}) = p(b_{ij} = 1|\boldsymbol{P}) \cdot p(\pi_i^{-1} < \pi_j^{-1}|\theta).$$

PACER utilizes the Bernoulli-Plackett-Luce distribution to model a probability distribution over DAGs (Figure 1). Note that the pair $(\pi, \boldsymbol{B})$ uniquely specifies a DAG with adjacency matrix $\boldsymbol{A} = (a_{ij}) \in \{0,1\}^{n \times n}$. Each entry of the

adjacency matrix is defined as $a_{ij} = b_{ij} \cdot \mathbb{1}\left[\pi_i^{-1} < \pi_j^{-1}\right]$, where $\mathbb{1}$ is the indicator function and $\pi^{-1}$ denotes the inverse of permutation $\pi$. We denote by $M(\Theta)$ the distribution over DAGs induced by this model. Under this framework, exploring the search space of DAGs reduces to inferring the parameters $\Theta = (\theta, \boldsymbol{P})$ of the Bernoulli-Plackett-Luce distribution $M(\Theta)$ that optimize a specific objective function.

Importantly, PACER can incorporate prior knowledge through both components of the Bernoulli-Plackett-Luce model: the Plackett-Luce parameters $\theta$ can be initialized according to the expected node centralities, while prior knowledge about the presence of an edge $i \to j$ can be encoded by setting the Bernoulli probability $p_{ij}$ to a high value. Where possible, we advocate for leveraging domain knowledge and designing informative interventions to restrict the interventional Markov equivalence class.

**Expressivity of the Bernoulli-Plackett-Luce model.** The Bernoulli-Plackett-Luce parameterization prioritizes scalability over universal expressivity. Specifically, it does not explicitly model edge dependencies that cannot be captured via a shared node ordering and independent edge probabilities. This design is deliberate and reflects a trade-off between expressivity and scalability: this parameterization provides a flexible approximation that is well-suited for high-dimensional causal discovery.

### 4.2. Objective function

We maximize the expected log-likelihood of non-intervened variables across all intervention regimes. We optimize the

adjacency parameters $\Theta = (\theta, \boldsymbol{P})$ of the joint Bernoulli-Plackett-Luce distribution $M(\Theta) = p(\pi, \boldsymbol{B}|\theta, \boldsymbol{P})$ and the parameters $\Omega$ of the conditional distribution of each child conditioned on its parents. We define our *unconstrained* objective as:

$$S(\Theta, \Omega) = \mathbb{E}_{M' \sim M(\Theta)}[f(M', \Omega)] - \lambda \|\mathbb{E}[M(\Theta)]\|_1, \quad (3)$$

where $\|\mathbb{E}[M(\Theta)]\|_1$ is the expected number of edges under the Bernoulli-Plackett-Luce model (calculated as the entrywise $L_1$ norm), $\lambda$ is a hyperparameter, and $f(M', \Omega)$ is an interventional likelihood-based objective defined as:

$$f(M', \Omega) = \sum_{r=1}^R \mathbb{E}_{X \sim P_{\text{data}}^{(r)}} \sum_{j \notin \mathcal{I}_r} \log p_\Omega^j(X_j | M_j', X_{-j}).$$

Here, $P_{\text{data}}^{(r)}$ is the data distribution under regime $r$, $\mathcal{I}_r$ represents the set of intervened variables in regime $r$, $R$ is the total number of regimes, and $p_\Omega^j$ denotes the conditional density of variable $X_j$ given its parents, as defined by the remaining variables $X_{-j}$ and the sampled DAG $M' \sim M(\Theta)$. Note that, in contrast to DCDI and DCDFG (Equation 1), we do not require any acyclicity constraints because the sampled adjacency matrices $M'$ induce DAGs by design.

### 4.3. Learning the distribution over DAGs

In this section, we introduce a generic approach to optimize our objective function via REINFORCE (Williams, 1992). We also show that, for a specific form of the causal mechanisms, we can exactly calculate the gradients of our objective with respect to the parameters of the Bernoulli-Plackett-Luce model without sampling any DAGs.

**Estimating gradients via REINFORCE.** Sampling adjacency matrices $M'$ from the distribution over DAGs $M(\Theta)$ is a non-differentiable operation. To estimate the gradients $\nabla_\Theta S(\Theta, \Omega)$ of the objective (Equation 3) with respect to the adjacency parameters $\Theta$, *i.e.* logits $\theta$ of the Plackett-Luce distribution and probabilities $p_{ij}$ of the Bernoulli distributions, we use REINFORCE (Williams, 1992) with an average baseline. For simplicity, let us consider our objective without the regularizer, $\lambda = 0$, and derive the formula for the gradient $\nabla_\Theta S(\Theta, \Omega)$:

$$\nabla_\Theta S(\Theta, \Omega) =$$
$$\mathbb{E}_{M' \sim M(\Theta)}\Big[(f(M', \Omega) - m)\nabla_\Theta \log p(M'|\Theta)\Big], \quad (4)$$
$$\text{with } m = \mathbb{E}_{M' \sim M(\Theta)}[f(M', \Omega)].$$

The average baseline $m$ allows us to reduce the variance of the estimator while keeping the estimate unbiased. We calculate this expectation using Monte Carlo samples. Intuitively, this estimator reinforces DAGs $M'$ that achieve

better-than-average scores. In the general case, $\lambda > 0$, we incorporate the subgradient of the regularizer into our update to sparsify the adjacency matrix.

**Variance of the REINFORCE estimator.** The following lemma bounds the variance of our REINFORCE estimator.

**Lemma 4.3.** *Let $p(M|\Theta)$ be the density of a Bernoulli-Plackett-Luce distribution combining a Plackett–Luce distribution over permutations and independent Bernoulli edge variables $p_{ij}$, and let $M' \sim M(\Theta)$ be a sample from it. Then, for the REINFORCE estimator with baseline $m$:*

$$g(M') \quad = \quad (f(M', \Omega) - m)\nabla_\Theta \log p(M'|\Theta)$$

*the variance is bounded as follows:*

$$\text{Var}(g) \leq \max_{M'}[(f(M', \Omega) - m)^2]\left(n + \sum_{i,j} p_{ij}(1 - p_{ij})\right).$$

*Proof.* The bound follows from the trace of the Fisher Information matrix for the Bernoulli-Plackett-Luce parameterization. See Appendix A for the full derivation. $\square$

The variance of our REINFORCE estimator therefore scales as $\mathcal{O}(n)$ for the Plackett–Luce component and as $\sum_{i,j} p_{ij}(1 - p_{ij})$ (worst-case $\mathcal{O}(n^2)$) for the Bernoulli edge component, with the latter vanishing as edges become deterministic. We provide an empirical analysis of PACER's gradient variance in Appendix B.

**Calculating the exact score without sampling DAGs.** The following theorem shows that, for a specific form of conditional distributions, we can calculate the exact score without drawing Monte Carlo samples from the distribution over DAGs. For simplicity, we set $\lambda = 0$ in the following result, analyzing the objective without the sparse regularizer.

**Theorem 4.4.** *Assume that each conditional distribution is Gaussian with mean $\mu_j$ and scale $\sigma_j$:*

$$p_\Omega^j(X_j \mid M_j', X_{-j}) = \mathcal{N}(X_j \mid \mu_j, \sigma_j^2),$$
$$\mu_j = b_j + \sum_{i=1}^n a_{ij} w_{ij} x_i,$$

*where $a_{ij} \in \{0, 1\}$ indicates whether the edge $i \rightarrow j$ is present in the DAG, $n$ is the total number of variables, and $w_{ij}$, $b_j$, and $\sigma_j$ are learnable parameters. Assume that the edges $a_{ij}$ are distributed according to our Bernoulli-Plackett-Luce model.*

*Then, the expected log-likelihood score $S(\Theta, \Omega)$ can be expressed as:*

$$S(\Theta, \Omega) = -\frac{1}{2}\sum_{r=1}^R \mathbb{E}_{\boldsymbol{x} \sim P_{\text{data}}^{(r)}} \sum_{j \notin \mathcal{I}_r} s(\boldsymbol{x}, j),$$

*with sample- and node-specific scores $s(\boldsymbol{x}, j)$ defined as:*

$$s(\boldsymbol{x}, j) = \log(2\pi\sigma_j^2) + \frac{1}{\sigma_j^2}\Bigg(\Big(x_j - b_j - \sum_{i=1}^n \mathbb{E}[a_{ij}]w_{ij}x_i\Big)^2$$

$$+ \sum_{i=1}^n \mathbb{E}[a_{ij}](1 - \mathbb{E}[a_{ij}])w_{ij}^2 x_i^2$$

$$+ \sum_{i=1}^n \sum_{k=1,k\neq i}^n (\mathbb{E}[a_{ij}]w_{ij}x_i)(\mathbb{E}[a_{kj}]w_{kj}x_k)$$

$$\left(\frac{e^{\theta_j}}{e^{\theta_i} + e^{\theta_j} + e^{\theta_k}}\right)\Bigg),$$

*and expectation $\mathbb{E}[a_{ij}] = p_{ij}\left(\frac{e^{\theta_i}}{e^{\theta_i} + e^{\theta_j}}\right)$ given by the Bernoulli-Plackett-Luce model, where $p_{ij}$ are learnable Bernoulli probabilities and $\theta_i$ and $\theta_j$ are the Plackett-Luce logits for variables $i$ and $j$, respectively.*

*Proof.* See Appendix C. □

The main implication of Theorem 4.4 is that, if we assume that the conditional distributions are Gaussian with linear mechanisms, we can exactly calculate the gradients of our objective with respect to the parameters of the Bernoulli-Plackett-Luce model without sampling from the distribution over DAGs. We generalize this result to multivariate anisotropic Normal models in Theorem C.5. We outline an extension to general models in Theorem C.7 (Appendix C), using a second-order Taylor expansion of the log-likelihood around a null graph, *i.e.*, a model with no causal relationships. We outline this approach as a direction for future research, noting that it currently lacks the scalability of the REINFORCE estimator.

**Computational and memory complexity.** PACER is designed to handle interventional data within a single likelihood-based objective and admits an exact analytic expression for the expected objective, eliminating the need for sampling. The computational cost of PACER's forward pass is $\mathcal{O}(n^2)$, allowing PACER to scale effectively to thousands of variables where $\mathcal{O}(n^3)$ methods become prohibitive. PACER's memory complexity is $\mathcal{O}(n^2)$. We provide an extended complexity analysis in Appendix D.

**4.4. Modeling the conditional distributions**

PACER supports modeling the conditional distributions $p_\Omega^j(X_j|M_j', X_{-j})$ in different ways, *i.e.* the structure of these conditionals can be chosen according to the type of data. By default, we use a Normal distribution to model

continuous data:

$$p_\Omega^j(X_j|M_j', X_{-j}) = \mathcal{N}(X_j|\mu_j, \sigma_j^2)$$
$$\mu_j = \text{MLP}_{\mu_j}(M_j' \odot X_{-j})$$
$$\sigma_j = \text{softplus}\big(\text{MLP}_{\sigma_j}(M_j' \odot X_{-j})\big).$$

Here, $\odot$ represents the element-wise vector product, $\text{softplus}(x) = \log(1 + e^x)$, and $\text{MLP}_{\mu_j}$ and $\text{MLP}_{\sigma_j}$ are multilayer perceptrons specific to variable $j$, which allow modeling non-linear causal mechanisms. PACER also supports tailored probability distributions, *e.g.* negative binomial distributions for modeling single-cell RNA-seq counts, and can be extended to accommodate more flexible likelihoods or non-parametric densities.

## 5. Results

We evaluate PACER using observational and interventional data from synthetic and real-world datasets (Sachs et al., 2005; Chevalley et al., 2025; Frangieh et al., 2021). Across experiments, the set of baselines and evaluation metrics varies to reflect differences in data scale and experimental design, and to follow the standard protocols established for each benchmark. In particular, some methods are only applicable to observational data, while others require interventional information or do not computationally scale to large graphs. For each setting, we therefore compare against strong and representative baselines that are commonly used in prior work and feasible for the corresponding dataset.

**Datasets.** We evaluate PACER on a diverse set of benchmarks spanning both synthetic and real-world datasets, including small-scale signaling networks and large-scale genetic perturbation data. We generate synthetic datasets following prior simulation protocols (Brouillard et al., 2020; Nazaret et al., 2024) using random graphs with a fixed edge density and neural network-based causal mechanisms (Appendix G). We further evaluate PACER on three real-world benchmarks. Sachs *et al.* (Sachs et al., 2005) consists of flow cytometry measurements of 11 phosphoproteins under observational and interventional conditions with a curated ground-truth network. CausalBench (Chevalley et al., 2025) is a benchmark for gene regulatory network inference using large-scale Perturb-seq datasets (RPE1 and K562 cell lines). Perturb-CITE-seq (Frangieh et al., 2021) provides multimodal single-cell perturbation data across three experimental conditions (control, co-culture, and IFN-$\gamma$ stimulation).

**Baseline methods.** We compare PACER with representative constraint-based, score-based, and differentiable causal discovery methods commonly used in prior work. These baselines span observational, interventional, and large-scale settings (Appendix E). Unless otherwise specified, all baselines are run with default hyperparameters. For PACER,

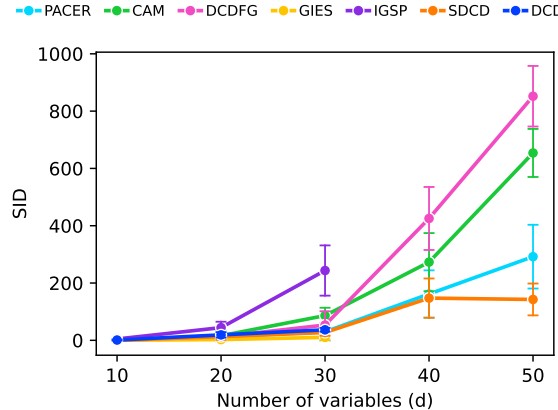

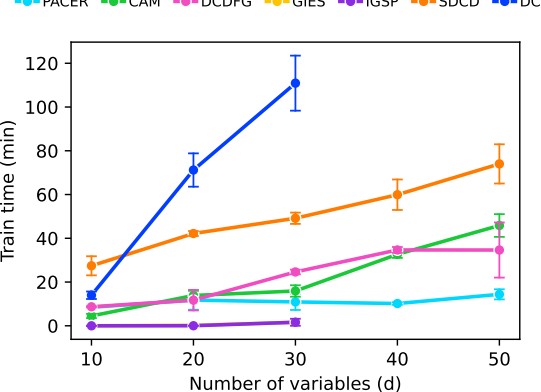

*Figure 2.* SID with increasing numbers of variables $d \leq 50$. SID score for $d = \{10, 20, 30, 40, 50\}$ is reported. Lower SID indicates better performance. Missing data points refer to runs that exceeded the 6 hr time limit. Error bars denote the standard deviation over three random seeds.

*Figure 3.* Runtime scaling with increasing numbers of variables $d \leq 50$. We report wall-clock training time for varying number of variables. Missing data points indicate runs that exceeded the 6 hour time limit. Error bars denote the standard deviation over three random seeds.

we use a single fixed set of hyperparameters across all real-world benchmarks (Appendix F). For synthetic experiments only, we match PACER's architectural capacity (*e.g.*, hidden dimensions) to that of other neural baselines to ensure a fair comparison under controlled simulation settings (Appendices G and E).

**Evaluation metrics.** We follow standard evaluation protocols for each benchmark. For synthetic data and the Sachs dataset, where the ground-truth causal graph is known, we use structure recovery metrics including the structural hamming distance (SHD), the structural intervention distance (SID), the false discovery rate (FDR), the true positive rate (TPR), and the F1 score. For CausalBench, we adopt the benchmark's Wasserstein and false omission rate metrics, and for Perturb-CITE-seq we evaluate predictive performance on held-out interventions using interventional likelihood-based metrics. We provide details in Appendix H.

**Synthetic experiments.** We study the performance and scalability of PACER by varying the number of variables $d$ in the synthetically generated data (Appendix G). We repeat experiments over three randomly generated datasets for each setting, adhering to a strict 6-hour runtime limit consistent with established benchmarks (Brouillard et al., 2020; Nazaret et al., 2024).

As the graph dimensionality increases, PACER maintains a favorable balance between structural accuracy and computational scalability, whereas other baselines exhibit rapid performance degradation or fail to execute beyond moderate graph sizes (Figures 2 and 3). Specifically, DCDI becomes numerically unstable and exceeds the 6-hour threshold for $d \geq 40$, illustrating the limitations of differentiable penalty-based methods in high-dimensional regimes. While CAM and DCDFG are capable of executing on graphs with up to

100 variables without timing out, both methods suffer from sharply increasing SID and substantially poorer scalability compared to PACER (Appendix I). PACER robustly scales to large-scale graphs, a regime that remains computationally prohibitive for existing interventional approaches. In terms of robustness to the sparsity coefficient $\lambda$, PACER remains stable across a broad range of configurations, achieving near-optimal performance around $\lambda = 1$ (Appendix J). We further compare PACER to the differentiable DAG sampling approach (Charpentier et al., 2022) in Appendix K.

**Protein signaling experiments.** We conduct the protein signaling benchmark in two case studies, corresponding to the observational and interventional regimes of the (Sachs et al., 2005) dataset (Appendix L). In the observational setting, no external perturbations are applied, *i.e.*, the data reflect the endogenous variability of the signaling network. In the interventional setting, specific proteins are chemically perturbed, providing additional constraints for causal discovery. In both settings, we evaluate against the consensus network reported in Sachs *et al.* We note that the correctness of the Sachs consensus network should be interpreted as an imperfect reference rather than the definitive ground truth (Mooij et al., 2020).

In both cases, PACER achieves competitive performance in different metrics, often outperforming existing methods (Table 1). In the observational setting, several constraint-based and score-based approaches excel in individual metrics, *e.g.* NOTEARS (Zheng et al., 2018) achieves the most accurate structure in terms of SHD, while GRaSP (Lam et al., 2022) offers the most reliable causal effect predictions in terms of SID. However, no single observational baseline dominates across all evaluation criteria. Among existing approaches, PACER stands out as the only method that consistently ranks among the top two across all five metrics, while attaining

*Table 1.* Performance summary on flow cytometry data from (Sachs et al., 2005). Network recovery metrics in observational and interventional settings. Colors highlight the two best-performing methods for each metric (darker indicates better performance). Correct: number of correctly inferred edges, total: counts all predicted edges, and DAG: whether the inferred structure is acyclic. ↓: lower is better. ↑: higher is better. We describe the metrics in Appendix H.

| | Method | SHD↓ | SID↓ | FDR↓ | TPR↑ | F1↑ | Correct | Total | DAG |
|---|---|---|---|---|---|---|---|---|---|
| Observational | GS | 28 | 51 | 0.81 | 0.35 | 0.25 | 6 | 31 | ✗ |
| | GES | 31 | 51 | 0.84 | 0.35 | 0.22 | 6 | 38 | ✗ |
| | IAMB | 26 | 49 | 0.79 | 0.41 | 0.28 | 7 | 33 | ✗ |
| | MMPC | 30 | 62 | 0.76 | 0.71 | 0.36 | 12 | 50 | ✗ |
| | GRaSP | 27 | 41 | 0.76 | 0.47 | 0.31 | 8 | 34 | ✗ |
| | BOSS | 25 | 43 | 0.74 | 0.47 | 0.33 | 8 | 31 | ✗ |
| | LiNGAM | 17 | 53 | 0.75 | 0.06 | 0.10 | 1 | 4 | ✓ |
| | NOTEARS | 16 | 57 | 0.67 | 0.12 | 0.17 | 2 | 6 | ✓ |
| | PACER | 21 | 42 | 0.64 | 0.53 | 0.43 | 9 | 25 | ✓ |
| Interventional | IGSP | 18 | 54 | 0.75 | 0.24 | 0.24 | 4 | 16 | ✓ |
| | GIES | 38 | 34 | 0.83 | 0.59 | 0.27 | 10 | 58 | ✓ |
| | CAM | 35 | 20 | 0.74 | 0.71 | 0.38 | 12 | 46 | ✓ |
| | DCDI-G | 36 | 43 | 0.85 | 0.35 | 0.21 | 6 | 40 | ✓ |
| | DCDI-DSF | 33 | 47 | 0.84 | 0.35 | 0.22 | 6 | 37 | ✓ |
| | SDCD | 24 | 28 | 0.67 | 0.53 | 0.41 | 9 | 27 | ✓ |
| | PACER | 19 | 31 | 0.58 | 0.59 | 0.49 | 10 | 24 | ✓ |

*Table 2.* Performance and runtime summary on the CausalBench benchmark. Mean Wasserstein distance (W) and false omission rate (FOR) for all methods on the RPE1 and K562 Perturb-seq datasets. Higher Wasserstein values indicate better recovery of intervention-induced distributional shifts; lower FOR indicates fewer missed known interactions. Colors highlight the two best-performing methods for each metric (darker indicates better performance). Time in minutes. We run all the methods on the same machine using a GPU accelerator (24GB NVIDIA GeForce RTX 3090) where applicable. PACER: baseline with default hyperparameters (Appendix F). PACER (A): analytic version of PACER (model in Theorem 4.4). PACER (TF): baseline with TF mask (Bernoulli probabilities of edges outgoing from non-transcription factor nodes are set to zero). PACER (A, TF): Analytic PACER baseline with TF mask.

| | RPE1 | | | K562 | | |
|---|---|---|---|---|---|---|
| Method | W ↑ | FOR ↓ | Time | W ↑ | FOR ↓ | Time |
| DCDFG | 0.126 | 0.132 | 159.039 | 0.139 | 0.190 | 235.643 |
| DCDI-DSF | 0.173 | 0.136 | 46.884 | 0.162 | 0.184 | 110.592 |
| DCDI-G | 0.194 | 0.142 | 144.605 | 0.183 | 0.184 | 346.282 |
| GES | 0.145 | 0.154 | 91.104 | 0.163 | 0.192 | 156.843 |
| GIES | 0.144 | 0.152 | 63.865 | 0.152 | 0.184 | 55.547 |
| NOTEARS (linear) | 0.164 | 0.154 | 113.812 | 0.164 | 0.192 | 361.149 |
| NOTEARS (MLP) | 0.137 | 0.116 | 698.400 | 0.138 | 0.184 | 1005.326 |
| Sort&regress | 0.157 | 0.110 | 2.000 | 0.161 | 0.172 | 5.482 |
| SDCD | 0.126 | 0.154 | 112.455 | 0.186 | 0.192 | 183.389 |
| PACER | 0.167 | 0.134 | 15.074 | 0.150 | 0.188 | 3.109 |
| PACER (A) | 0.203 | 0.146 | 7.202 | 0.153 | 0.184 | 0.912 |
| PACER (TF) | 0.164 | 0.150 | 1.863 | 0.182 | 0.192 | 3.140 |
| PACER (A, TF) | 0.213 | 0.148 | 8.279 | 0.183 | 0.190 | 0.904 |

the strongest balance between precision and recall. In the interventional setting, methods that explicitly exploit perturbation information (such as PACER, CAM, and GIES) show substantial performance gains, highlighting the importance of interventions for resolving causal directions. PACER achieves the highest F1 score and the best FDR score while being the second best method across other metrics.

Qualitatively, the learnt interventional network captures canonical signaling interactions, such as the PKC→RAF→MEK→ERK cascade (Ueda et al., 1996) and downstream effects of ERK and PKA on AKT (Appendix M). Interestingly, we observe that PACER infers a directed edge from RAF to ERK that is not present in the consensus network (Appendix M). This finding is consistent with prior discussions suggesting the presence of a feedback loop from ERK to RAF (Mooij et al., 2020), as well as more recent studies of the RAF/MEK/ERK signaling cascade, *e.g.*, (Ullah et al., 2022) report negative feedback phosphorylation between RAF and ERK. Remaining discrepancies are typically spurious edges in densely connected regions, consistent with the known difficulty of recovering directionality in tightly coupled regions (Epskamp & Fried, 2018). Overall, PACER performs similar or better than existing methods, showing robustness across both observational and interventional settings and the ability to recover biologically meaningful signaling pathways.

**CausalBench (large-scale genetic perturbation data).** Across both datasets (RPE1 and K562), PACER achieves strong and consistent performance in terms of the Wasserstein metric, ranking among the top performing methods

(Table 2). Importantly, this performance is obtained with a substantially lower computational cost. Unlike approaches such as NOTEARS (Zheng et al., 2018), DCDI (Brouillard et al., 2020), and DCDFG (Lopez et al., 2022), which rely on continuous relaxations and acyclicity penalties, PACER explores the DAG search space directly by construction. This enables substantial computational gains: for example, on the K562 dataset, PACER achieves the same Wasserstein performance as the best competing method while being over 300 times faster, with a similar dramatic speedup on RPE1 (Table 2). The analytic version implementing Theorem 4.4, *i.e.*, PACER (A), further amplifies efficiency.

Notably, PACER operates on the full gene regulatory graph in a single optimization loop. As noted in the CausalBench benchmark (Chevalley et al., 2025), most existing methods do not computationally scale to transcriptome-wide graphs and must instead be applied to smaller subgraphs, with the final network obtained by merging independently inferred subnetworks. CausalBench runs GES and GIES on partitions of size 30, while DCDI-based methods are run on partitions of size 50. Only a small subset of approaches (NOTEARS, DCDFG, GRNBoost, and Sortnregress) oper-

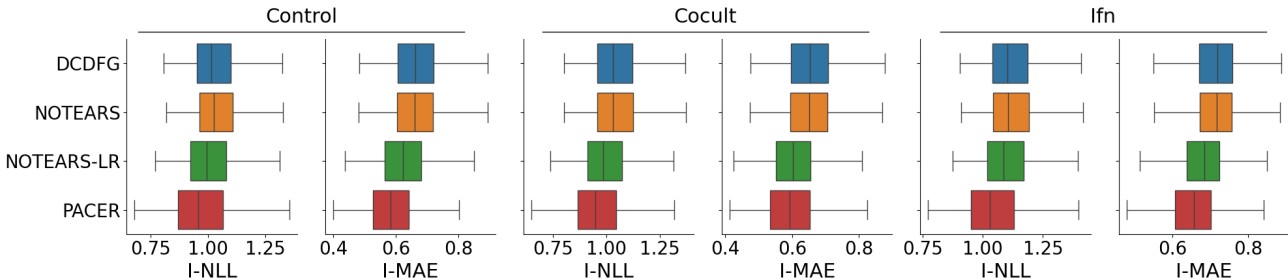

*Figure 4.* Performance on the Perturb-CITE-seq dataset. Interventional negative log-likelihood (I-NLL) and interventional mean absolute error (I-MAE) evaluated on held-out interventions across three experimental conditions: control, co-culture, and IFN-$\gamma$ stimulation. Lower values indicate better performance. Sampling units are held-out perturbations. Boxes depict distribution quartiles, with the center line corresponding to the median, and whiskers span 1.5 × interquartile range.

ate on the full graph without partitioning. Such partitioning can break global modeling assumptions and prevents methods from fully leveraging the joint interventional structure of the data. In contrast, PACER directly optimizes over all variables simultaneously, which could contribute to its strong performance on large-scale perturbation datasets.

We further assess whether injecting prior knowledge leads to performance gains (Table 2). For the PACER (TF) variants, we set the Bernoulli probabilities of edges outgoing from non-transcription factor (TF) nodes to zero, allowing only TFs to regulate other genes. This constraint yields improvements in Wasserstein performance in both RPE1 and K562 despite restricting the complexity of the model. While some methods (such as NOTEARS, DCDFG, and DCDI variants) can in principle incorporate certain forms of prior knowledge via masking or penalties, these modifications are often nontrivial and require custom adaptations. PACER enables systematic and flexible integration of prior knowledge without altering the method's optimization process, making it uniquely suited for integrating biological inductive biases similar to those used in CellOracle (Kamimoto et al., 2023) and SCENIC (Aibar et al., 2017; Bravo González-Blas et al., 2023). These frameworks leverage modalities including TF binding motifs and chromatin accessibility, and have proven effective in recent gene regulatory network inference benchmarks (Badia-i Mompel et al., 2024).

**Perturb-CITE-seq (large-scale genetic perturbation data).** Following the Perturb-CITE-seq setting of DCDFG (Lopez et al., 2022) (Appendix N), we compare PACER with DCDFG (Lopez et al., 2022), NOTEARS (Zheng et al., 2018), and NOTEARS-LR (Fang et al., 2023) using the interventional negative log-likelihood and mean absolute error as evaluation metrics. This dataset poses a particularly challenging setting for causal discovery, as it retains close to one thousand genes after preprocessing and 248 distinct perturbations, a scale at which most existing methods become computationally infeasible. Despite this, PACER achieves the best performance in terms of both metrics across the

three experimental conditions (Figure 4) while being on average over 300 times faster than DCDFG (Appendix O).

## 6. Conclusion

We introduce PACER, a scalable framework for causal discovery from observational and interventional data that guarantees acyclicity by construction. PACER parameterizes a distribution over DAGs through variable permutations and edge probabilities, eliminating the need for surrogate acyclicity penalties. This design enables a unified likelihood-based treatment of interventions while supporting flexible conditional models and prior knowledge. Empirically, PACER matches or exceeds the performance of state-of-the-art causal discovery methods on protein signaling and large-scale genetic perturbation benchmarks, while scaling efficiently to networks with thousands of variables. In particular, PACER's analytic formulation yields substantial computational gains, demonstrating that exact and scalable causal discovery is achievable when the search space is appropriately structured. PACER offers a practical causal discovery solution for modern high-dimensional settings that remain challenging for existing methods.

PACER is available on GitHub at: `https://github.com/mlbio-epfl/PACER`. Project page: `https://brbiclab.epfl.ch/pacer`.

**Limitations and future work.** PACER assumes perfect interventions and causal sufficiency, *i.e.*, it does not model latent confounders. While PACER does not provide calibrated Bayesian uncertainty, it could serve as a computationally efficient bootstrap posterior representing the interventional Markov equivalence class (Hauser & Bühlmann, 2012). Future work includes extending the framework to imperfect interventions and unknown targets (see Appendix P for an extension outline), expanding the analytic treatment to broader classes of causal mechanisms, improving variance reduction for stochastic optimization, and integrating additional biological priors from multimodal data sources.

## Acknowledgements

We thank Maxim Kodryan, Jack Naimer, Shuo Wen, Yist Yu, and the anonymous reviewers for their constructive feedback. We gratefully acknowledge the support of the Peter und Traudl Engelhorn Foundation, the Swiss National Science Foundation (SNSF) starting grant TMSGI2_226252/1, SNSF grant IC00I0_231922, SNSF grant 10.004.411, and the Swiss AI Initiative Large Call #32. M.B. is a CIFAR Fellow in the Multiscale Human Program.

## Impact statement

This work advances methodological foundations for causal discovery from observational and interventional data. By enabling scalable and principled inference of directed acyclic graphs, the proposed framework could support scientific discovery in domains such as systems biology, genomics, and medicine, where understanding causal mechanisms is critical for hypothesis generation and experimental design.

At the same time, causal discovery methods can be misapplied if their assumptions (such as causal sufficiency, correct intervention modeling, or appropriate likelihood specification) are violated. Incorrect causal conclusions could lead to misguided downstream decisions, particularly in sensitive applications like biomedical research. We therefore emphasize that PACER is intended to serve as a data analysis and hypothesis-generation tool, to be used in conjunction with domain expertise and experimental validation rather than as a standalone decision-making system.

The framework itself does not introduce new risks related to data privacy, surveillance, or automated decision-making, as it operates on pre-collected datasets and does not prescribe actions. When applied to biological or clinical data, responsible data governance and ethical data collection practices remain essential. Overall, we believe the benefits of improved scalability and principled causal inference outweigh the potential risks, provided the method is used with appropriate care and domain awareness.

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

# A. REINFORCE variance

PACER relies on REINFORCE for gradient estimation. This is attractive because the estimator is unbiased, but its variance and convergence behavior in large, high-dimensional DAG spaces are not fully understood. This motivates a more explicit analysis of the score-function estimator, its variance reduction through baselines, and the special structure of the Plackett–Luce gradient.

## A.1. REINFORCE and baseline estimators

Let $M' \sim M(\Theta)$ denote a sample / policy from the graph distribution, and let $s(M') := f(M', \Omega)$ be the scalar reward / objective signal, where $\Omega$ is a fixed parameter. Write the pure REINFORCE estimate

$$g_{\text{REINFORCE}}(M') = s(M') \, G(M'),$$

where

$$G(M') = \nabla_\Theta \log p(M'|\Theta).$$

Then

$$\mathbb{E}_{M' \sim M(\Theta)}[\, g_{\text{REINFORCE}}(M')\,] = \nabla_\Theta \mathbb{E}_{M' \sim M(\Theta)}[s(M')]. \tag{5}$$

A baseline $m$ yields the modified estimator

$$g_{\text{BASELINE}}(M') = (s(M') - m) \, G(M').$$

REINFORCE is an unbiased estimator of the gradient of the expected reward (5). The baseline estimator remains unbiased provided the baseline $m$ does not depend on the current sampled graph, since $\mathbb{E}_{M' \sim M(\Theta)} G(M') = 0$.

The covariance matrices satisfy

$$\text{Cov}(g_{\text{REINFORCE}}(M')) = \mathbb{E}_{M' \sim M(\Theta)}\big[s(M')^2 \, G(M')G(M')^\top\big] - \nabla_\Theta J(\Theta)\,\nabla_\Theta J(\Theta)^\top,$$

$$\text{Cov}(g_{\text{BASELINE}}(M')) = \mathbb{E}_{M' \sim M(\Theta)}\big[(s(M') - m)^2 \, G(M')G(M')^\top\big] - \nabla_\Theta J(\Theta)\,\nabla_\Theta J(\Theta)^\top,$$

where $J(\Theta) = \mathbb{E}_{M' \sim M(\Theta)}[s(M')]$.

The total variance of the distribution is captured by the trace of the covariance matrix. Using $\text{tr}(vv^\top) = \|v\|_2^2$, we obtain

$$\text{Var}(g_{\text{BASELINE}}(M')) = \mathbb{E}_{M' \sim M(\Theta)}\big[(s(M') - m)^2 \, \|G(M')\|_2^2\big] - \|\nabla_\Theta J(\Theta)\|_2^2. \tag{6}$$

Thus, the variance depends critically on:

- the magnitude of $(s(M') - m)^2$,

- the norm $\|G(M')\|_2^2$.

The scalar baseline that minimizes the trace of the covariance is

$$m^\star = \frac{\mathbb{E}\big[s(M') \, \|G(M')\|_2^2\big]}{\mathbb{E}[\|G(M')\|_2^2]}.$$

In practice, this quantity is often too expensive to estimate exactly, so a running average of $s(M')$ is a common and inexpensive approximation.

## A.2. Translation invariance and coordinate-wise bound

First, we consider solely the Plackett–Luce model with parameter $\theta \in \mathbb{R}^n$ and bound the norm of the corresponding component of the gradient: $\nabla_\theta \log p(M'|\Theta)$.

For ease of notation, we denote the density of the Plackett–Luce model by $p_\theta(\pi)$ and our goal is to bound $\|\nabla_\theta \log p_\theta(\pi)\|_2^2$. We denote by $\pi$ the corresponding samples from the Plackett–Luce distribution.

We observe that for the Plackett–Luce model, adding a constant to every logit does not change the distribution. This shift invariance implies that the gradient of the log-probability is orthogonal to the all-ones vector.

**Lemma A.1** (Zero-sum score function). *Let $\theta \in \mathbb{R}^n$ parameterize a Plackett–Luce distribution. Then*

$$\mathbf{1}^\top \nabla_\theta \log p_\theta(\pi) = 0.$$

*Proof.* Define

$$f(c) = \log p_{\theta + c\mathbf{1}}(\pi).$$

Because the Plackett–Luce distribution is invariant to global shifts of the logits, $f(c)$ is constant in $c$. Therefore

$$0 = f'(0) = \mathbf{1}^\top \nabla_\theta \log p_\theta(\pi).$$

$\square$

This property is useful in two ways. First, it rules out drift along the uniform direction. Second, it gives a dimension-reduction effect: the score lives on an $(n-1)$-dimensional hyperplane.

Consider a permutation $\pi = (\pi_1, ..., \pi_n)$. The Plackett-Luce log-probability can be written as

$$\log p_\theta(\pi) = \sum_{k=1}^n \left( \theta_{\pi_k} - \log \sum_{u=k}^n e^{\theta_{\pi_u}} \right)$$

Without loss of generality, we reorder the parameters $\theta$ according to this permutation, *i.e.*, $\pi = (1, \ldots, n)$. We now compute the score with respect to the reordered coordinates. For the $i$-th position we get

$$G_i(\pi) = \frac{\partial \log p_\theta(\pi)}{\partial \theta_i} = 1 - \sum_{k=1}^i \frac{e^{\theta_i}}{\sum_{u=k}^n e^{\theta_u}},$$

where

$$0 \;\leq\; \sum_{k=1}^i \frac{e^{\theta_i}}{\sum_{u=k}^n e^{\theta_u}} \;\leq\; i.$$

Therefore,

$$1 - i \;\leq\; G_i(\pi) \;\leq\; 1.$$

## A.3. Uniform bound

The zero-sum property can be combined with coordinate-wise bounds to obtain a clean quadratic bound.

**Proposition A.2.** *Suppose $v \in \mathbb{R}^n$ satisfies*

$$\mathbf{1}^\top v = 0 \qquad \text{and} \qquad 1 - i \leq v_i \leq 1 \quad \text{for all } i = 1, \ldots, n.$$

*Then*

$$\|v\|_2^2 \leq n(n-1).$$

*Moreover, this bound is tight.*

*Proof.* Let

$$u_i := 1 - v_i.$$

Then $0 \leq u_i \leq i$ and

$$\sum_{i=1}^{n} u_i = \sum_{i=1}^{n} (1 - v_i) = n - \mathbf{1}^\top v = n.$$

Also,

$$\|v\|_2^2 = \sum_{i=1}^{n} (1 - u_i)^2 = n - 2 \sum_{i=1}^{n} u_i + \sum_{i=1}^{n} u_i^2 = \sum_{i=1}^{n} u_i^2 - n.$$

Since $u_i \geq 0$,

$$\left( \sum_{i=1}^{n} u_i \right)^2 = \sum_{i=1}^{n} u_i^2 + 2 \sum_{i<j} u_i u_j \geq \sum_{i=1}^{n} u_i^2,$$

so $\sum_i u_i^2 \leq n^2$. Therefore

$$\|v\|_2^2 \leq n^2 - n = n(n-1).$$

The bound is attained by choosing

$$u = (0, 0, \ldots, 0, n), \qquad \text{equivalently} \qquad v = (1, 1, \ldots, 1, 1 - n).$$

$\square$

Hence:

$$\boxed{\|\nabla_\theta \log p_\theta(\pi)\|_2^2 \leq n(n-1).} \tag{7}$$

## A.4. Bound in expectation

Note that bound (7) holds uniformly, for all permutations $\pi$. Its right-hand side is of order $\mathcal{O}(n^2)$. However, for the expectation of $\|\nabla_\theta \log p_\theta(\pi)\|_2^2$ we can obtain a much better bound of order $\mathcal{O}(n)$.

Employing Fisher's information,

$$\mathcal{I}(\theta) = \mathbb{E}\left[ \nabla_\theta \log p_\theta(\pi) \nabla_\theta \log p_\theta(\pi)^\top \right]$$

we have

$$\text{tr}(\mathcal{I}(\theta)) = \mathbb{E}[\|\nabla_\theta \log p_\theta(\pi)\|_2^2]$$

Using the negative expectation of the second derivative of the Plackett-Luce log-probability, the trace is

$$\text{tr}(\mathcal{I}(\theta)) = \sum_{i=1}^{n} \sum_{k=1}^{i} \frac{e^{\theta_i} \sum_{u=k}^{n} e^{\theta_u} - e^{2\theta_i}}{\left( \sum_{u=k}^{n} e^{\theta_u} \right)^2}.$$

We can swap the order of summation to obtain:

$$\text{tr}(\mathcal{I}(\theta)) = \sum_{k=1}^{n} \frac{1}{\left( \sum_{u=k}^{n} e^{\theta_u} \right)^2} \left[ \sum_{i=k}^{n} \left( e^{\theta_i} \sum_{u=k}^{n} e^{\theta_u} - e^{2\theta_i} \right) \right].$$

The term in the brackets is in fact $\sum_{i=k}^{n} \left( e^{\theta_i} \sum_{u=k}^{n} e^{\theta_u} - e^{2\theta_i} \right) = \left( \sum_{i=k}^{n} e^{\theta_i} \right)^2 - \sum_{i=k}^{n} e^{2\theta_i}$. Substituting this back into the trace formula we get

$$\text{tr}(\mathcal{I}(\theta)) = \sum_{k=1}^{n} \frac{\left( \sum_{u=k}^{n} e^{\theta_u} \right)^2 - \sum_{u=k}^{n} e^{2\theta_u}}{\left( \sum_{u=k}^{n} e^{\theta_u} \right)^2} = \sum_{k=1}^{n} \left( 1 - \frac{\sum_{u=k}^{n} e^{2\theta_u}}{\left( \sum_{u=k}^{n} e^{\theta_u} \right)^2} \right).$$

Notice that each term is always between $0$ and $1$. For $k = n$, it is exactly $0$. For $k < n$, each term is $< 1$. From this we deduce

$$\boxed{\mathbb{E}[\|\nabla_\theta \log p_\theta(\pi)\|_2^2] = \mathrm{tr}(\mathcal{I}(\theta)) \leq n - 1.}$$ (8)

## A.5. The variance of Bernoulli matrix gradients

We define the probability of an edge $(i, j)$ existing via the sigmoid function $p_{ij} = \sigma(\omega_{ij})$, where $\omega_{ij}$ is a parameter. For a sampled edge indicator $b_{ij} \in \{0, 1\}$, the entry-wise score $G_{ij}$ is:

$$G_{ij} = \frac{\partial}{\partial \omega_{ij}} \log \left(\sigma(w_{ij})^{b_{ij}}(1 - \sigma(w_{ij})^{1-b_{ij}}\right) = b_{ij} - \sigma(\omega_{ij}),$$

so each element is between $-1$ and $1$.

The squared Frobenius norm of the matrix $G(b)$ is equivalent to the squared $L_2$ norm of the flattened vector:

$$\|G(b)\|_F^2 = \sum_{i=1}^n \sum_{j=1}^n G_{ij}^2 \leq \sum_{i=1}^n \sum_{j=1}^n 1 = n^2.$$

Thus,

$$\boxed{\|G(b)\|_F^2 \leq n^2.}$$ (9)

At the same time,

$$\mathbb{E}[\|G(b)\|_F^2] = \sum_{i=1}^n \sum_{j=1}^n \mathbb{E}[G_{ij}^2] = \sum_{i=1}^n \sum_{j=1}^n p_{ij}(1 - p_{ij})^2 + (1 - p_{ij})p_{ij}^2 = \sum_{i=1}^n \sum_{j=1}^n (1 - p_{ij})p_{ij}.$$

Thus,

$$\boxed{\mathbb{E}[\|G(b)\|_F^2] = \sum_{i,j}(1 - p_{ij})p_{ij}.}$$ (10)

## A.6. General estimate of the REINFORCE variance

Notice that the whole score $\|G(M')\|_2^2$ consists of two components: the Plackett-Luce component and the Bernoulli component, and it holds

$$\|G(M')\|_2^2 = \|\nabla_\theta \log p_\theta(\pi)\|_2^2 + \|G(b)\|_F^2.$$

Therefore, combining (7) and (9) we obtain:

$$\mathrm{Var}(g_{\text{BASELINE}}) = \mathbb{E}_{M' \sim M(\Theta)}\left[(s(M') - m)^2 \|G(M')\|_2^2\right] - \|\nabla_\Theta J(\Theta)\|_2^2 \leq \mathbb{E}_{M' \sim M(\Theta)}[(s(M') - m)^2] \cdot \mathcal{O}(n^2).$$ (11)

This implies that for $m = \frac{1}{k}\sum_{i=1}^k s(M_i')$, using that $Var(m) = Var(s(M'))/k$, we have

$$
\begin{aligned}
\mathrm{Var}(g_{\text{BASELINE}}) &\leq \mathbb{E}_{M' \sim M(\Theta)}[(s(M') - m)^2] \cdot \mathcal{O}(n^2) \\
&= \mathbb{E}_{M' \sim M(\Theta)}[((s(M') - \mu) - (m - \mu))^2] \cdot \mathcal{O}(n^2) \\
&= \left(\mathbb{E}_{M' \sim M(\Theta)}[(s(M') - \mu)^2] + \mathbb{E}_{M' \sim M(\Theta)}[(m - \mu)^2] - 2\mathbb{E}_{M' \sim M(\Theta)}[(s(M') - \mu)(m - \mu)]\right) \cdot \mathcal{O}(n^2) \\
&= Var_{M' \sim M(\Theta)}[s(M')]\left(1 - \tfrac{1}{k}\right) \cdot \mathcal{O}(n^2).
\end{aligned}
$$

### A.7. Estimate of the REINFORCE variance for bounded variation

Assume that there exists $V > 0$ such that

$$(s(M') - m)^2 \leq V,$$

uniformly for any DAG $M'$. Then, using (8) and (10), we conclude:

$$\text{Var}(g_{\text{BASELINE}}) \leq V \cdot \mathbb{E}[\|G(M')\|^2] \leq V \cdot \left(n + \sum_{i,j} p_{ij}(1 - p_{ij})\right),$$

which is much better explicit dependence on $n$ than that one in (11). This completes the proof of Lemma 4.3.

## B. Stability and convergence of the gradient estimator

We provide an empirical analysis of the stability and convergence behavior of PACER's gradient estimator in (i) large-scale synthetic graphs (Section B.1) and (ii) a real large-scale single-cell perturbation dataset (Section B.2).

### B.1. Controlled experiments on large synthetic graphs

To assess stability independently of modeling choices, we conduct controlled experiments using a likelihood-free objective (mean squared error to a target DAG) on graphs with up to $N = 500$ nodes, with an edge noise probability of $\rho = 0.3$ (each edge in the true DAG is independently omitted with probability $\rho$ at each training step to simulate the uncertainty inherent in real-world causal discovery). This setting isolates the effect of the DAG parameterization and optimization from any specific likelihood model or architecture.

**Convergence of topological ordering.** Figure 5 shows Kendall $\tau$ between the inferred and ground-truth partial orderings over training steps for $N = 500$ nodes. Both the analytic and REINFORCE-based variants of PACER converge stably to near-perfect ordering ($\tau \approx 1.0$) without any oscillatory or divergent behavior. The analytic variant converges substantially faster, reaching $\tau \approx 1.0$ within $\sim 200$ steps, while the REINFORCE variant converges more gradually but reliably. A comparison against DP-DAG baselines under the same objective is provided in Appendix K.

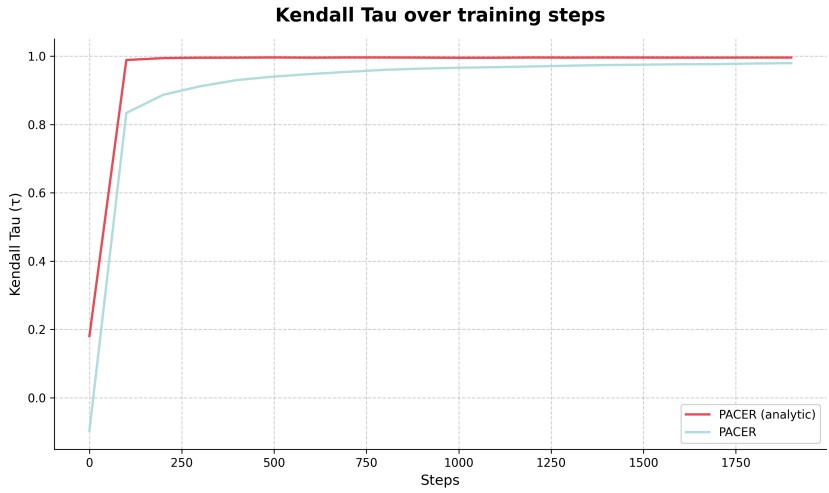

*Figure 5.* Kendall $\tau$ between inferred and ground-truth partial orderings over training steps ($N = 500$ nodes, $\rho = 0.3$). The analytic variant (red) converges rapidly to $\tau \approx 1.0$; the REINFORCE variant (teal) converges more gradually but reaches comparable accuracy. No divergence or oscillation is observed.

**Gradient variance across graph sizes (with and without control variate).** Figure 6 reports gradient variance over training steps for $N \in \{10, 50, 100, 500\}$ using the REINFORCE estimator with control variate. Despite a 50-fold increase in graph size, all curves decrease consistently and converge to comparable variance levels, demonstrating that the estimator scales stably to large graphs. For comparison, Figure 7 shows the same experiment *without* the control variate. Variance is

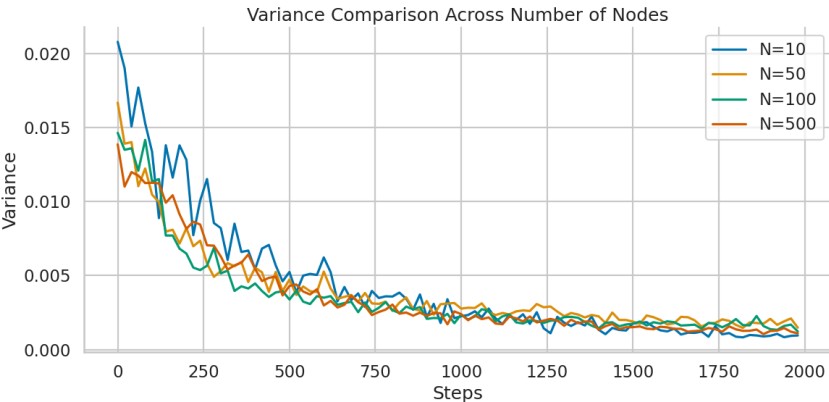

*Figure 6.* Gradient variance of REINFORCE with control variate across graph sizes $N \in \{10, 50, 100, 500\}$ (likelihood-free objective). Variance decreases monotonically for all $N$ and converges to similar magnitudes, demonstrating stable scaling with graph size.

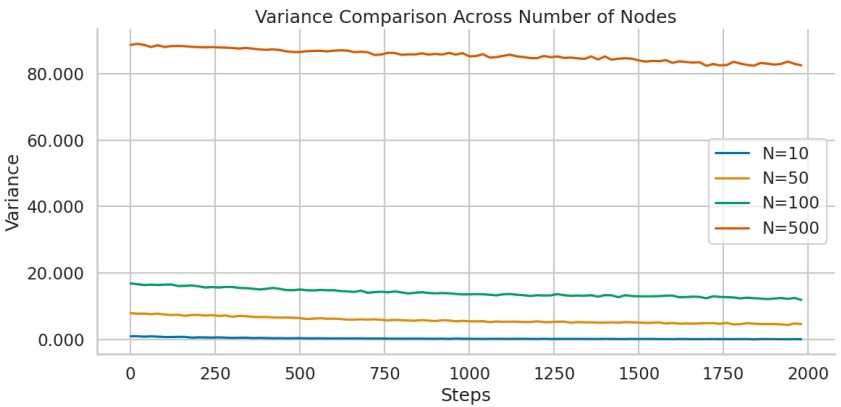

*Figure 7.* Gradient variance of REINFORCE *without* control variate across number of nodes $N \in \{10, 50, 100, 500\}$. Variance is orders of magnitude larger than with control variate (cf. Figure 6) and increases strongly with $N$, confirming the critical role of the control variate.

orders of magnitude larger and grows strongly with $N$ ($\approx$90 for $N$=500 vs. $\approx$0.002 with control variate at convergence), confirming that baseline subtraction is critical for stable large-scale training.

**Gradient variance across Monte Carlo samples.** Figure 8 shows gradient variance (mean $\pm$ std across seeds) for $N = 200$ nodes as a function of Monte Carlo sample count (10, 50, 100, and 200). Variance decreases both over training steps and as the number of Monte Carlo samples increases. Higher numbers of Monte Carlo samples also substantially tighten confidence bands across seeds.

### B.2. Variance analysis on the Replogle RPE1 dataset

We analyze gradient variance throughout training on the Replogle RPE1 dataset (Replogle et al., 2022), a large-scale CRISPR perturbation dataset with thousands of genes. We compare three estimators (Figure 9):

- **Analytic estimator** (Figure 9, left): achieves several *orders of magnitude* lower variance than stochastic estimators. Variance decreases monotonically on a log scale throughout the entire training run ($\sim$20,000 steps), reaching near-zero levels with no sign of instability.

- **REINFORCE with control variate** (Figure 9, center): variance is substantially reduced relative to the baseline without control variate. Increasing the number of Monte Carlo samples further decreases variance in a predictable manner.

- **REINFORCE without control variate** (Figure 9, right): exhibits persistently high variance ($\sim 3 \times 10^7$), confirming that the control variate is essential for stable optimization.

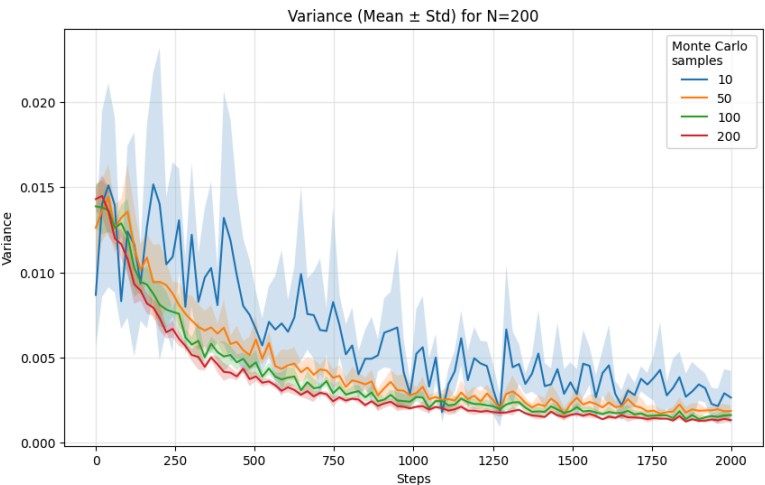

*Figure 8.* Gradient variance (mean $\pm$ std across seeds) for $N=200$ nodes and different numbers of Monte Carlo samples. Increasing the number of Monte Carlo samples consistently reduces both mean, variance, and cross-seed spread.

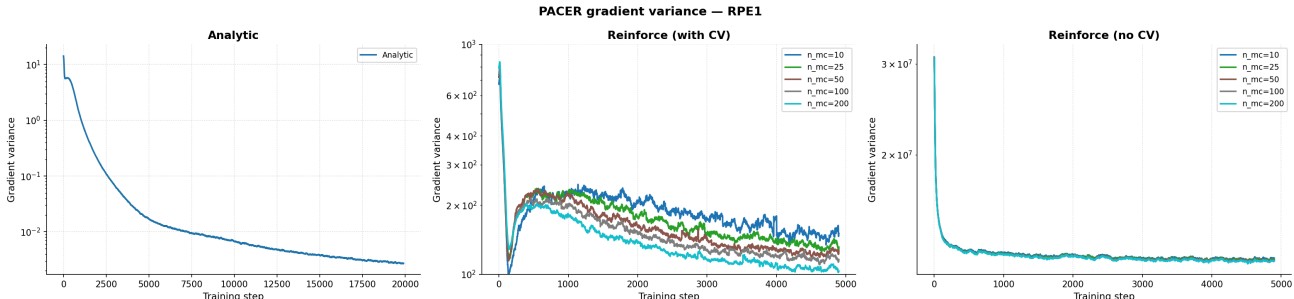

*Figure 9.* Gradient variance during training on the Replogle RPE1 dataset. *Left:* Analytic estimator (log scale): decreases monotonically to near zero over 20,000 steps. *Center:* REINFORCE with control variate for varying numbers of Monte Carlo samples; larger sample counts yield lower variance. *Right:* REINFORCE without control variate: variance is orders of magnitude higher, underscoring the necessity of variance reduction.

## C. Analytic estimators

### C.1. Lemmas

**Lemma C.1.** *Let $\pi$ be a random permutation of $\{1, \ldots, n\}$ sampled from a Plackett-Luce distribution with logits $\theta \in \mathbb{R}^n$. Then for any distinct $i, j$,*

$$\Pr(i \prec j) = \Pr(\pi_i^{-1} < \pi_j^{-1}) = \frac{e^{\theta_i}}{e^{\theta_i} + e^{\theta_j}}, \tag{12}$$

*where $i \prec j$ denotes that $i$ precedes $j$ in permutation $\pi$.*

*Proof.* We proceed by induction on $n$.

**Base case ($n = 2$).** With only two items $\{i, j\}$, the probability that $i$ precedes $j$ is simply

$$\Pr(i \prec j) = \Pr(\pi_1 = i) = \frac{e^{\theta_i}}{e^{\theta_i} + e^{\theta_j}}, \tag{13}$$

which satisfies the lemma.

**Induction step.** Suppose the statement holds for $n = N - 1$ items. Consider $n = N$ items, and let

$$Z = \sum_{k=1}^{n} e^{\theta_k}.$$

We decompose according to the first position of the permutation:

$$\Pr(i \prec j) = \Pr(\pi_1 = i) + \Pr(\pi_1 \neq i, \, i \prec j). \tag{14}$$

The first term in Equation 14 is:

$$\Pr(\pi_1 = i) = \frac{e^{\theta_i}}{Z}.$$

For the second term, note that if $\pi_1 \notin \{i, j\}$, then the relative order of $i$ and $j$ is distributed according to a Plackett–Luce model over $N - 1$ items with logits $\{\theta_k : k \neq \pi_1\}$. By the induction hypothesis,

$$\Pr(i \prec j \mid \pi_1 \notin \{i, j\}) = \frac{e^{\theta_i}}{e^{\theta_i} + e^{\theta_j}}.$$

Note also that the probability that the first element is neither $i$ or $j$ can be written as:

$$\Pr(\pi_1 \notin \{i, j\}) = 1 - \Pr(\pi_1 = i) - \Pr(\pi_1 = j) = \frac{Z - e^{\theta_i} - e^{\theta_j}}{Z}.$$

Hence, the second term in Equation 14 is:

$$\Pr(\pi_1 \neq i, \, i \prec j) = \Pr(\pi_1 \notin \{i, j\}) \Pr(i \prec j \mid \pi_1 \notin \{i, j\})$$
$$= \left( \frac{Z - e^{\theta_i} - e^{\theta_j}}{Z} \right) \left( \frac{e^{\theta_i}}{e^{\theta_i} + e^{\theta_j}} \right).$$

Putting the two terms together, the probability that $i$ precedes $j$ is:

$$\Pr(i \prec j) = \Pr(\pi_1 = i) + \Pr(\pi_1 \neq i, \, i \prec j)$$
$$= \frac{e^{\theta_i}}{Z} + \left( \frac{Z - e^{\theta_i} - e^{\theta_j}}{Z} \right) \left( \frac{e^{\theta_i}}{e^{\theta_i} + e^{\theta_j}} \right)$$
$$= \frac{e^{\theta_i}(e^{\theta_i} + e^{\theta_j}) + (Z - e^{\theta_i} - e^{\theta_j})e^{\theta_i}}{Z(e^{\theta_i} + e^{\theta_j})}$$
$$= \frac{Z(e^{\theta_i})}{Z(e^{\theta_i} + e^{\theta_j})}$$
$$= \frac{e^{\theta_i}}{e^{\theta_i} + e^{\theta_j}}.$$

Thus the claim holds for $n = N$, completing the induction. $\square$

**Corollary C.2.** *The expected probability $\mathbb{E}[a_{ij}]$ that an edge from $i$ to $j$ exists according to the Bernoulli-Plackett-Luce model is:*

$$\mathbb{E}[a_{ij}] = \left( \frac{e^{\theta_i}}{e^{\theta_i} + e^{\theta_j}} \right) p_{ij}$$

*Proof.* The proof follows from Lemma C.1. That is, the probability that an edge from $i$ to $j$ exists is given by the probability that $i$ precedes $j$ multiplied by the Bernoulli probability $p_{ij}$:

$$\mathbb{E}[a_{ij}] = \Pr(i \prec j) p_{ij} = \left( \frac{e^{\theta_i}}{e^{\theta_i} + e^{\theta_j}} \right) p_{ij} \tag{15}$$

$\square$

**Lemma C.3.** *Let $\pi$ be a random permutation of $\{1, \ldots, n\}$ sampled from a Plackett-Luce distribution with logits $\theta \in \mathbb{R}^n$. For any distinct $i, j, k$,*

$$\Pr(i \prec j, k \prec j) \;=\; \Pr(i \prec j)\Pr(k \prec j)\Big(1 + \frac{e^{\theta_j}}{e^{\theta_i} + e^{\theta_j} + e^{\theta_k}}\Big),$$

*where $i \prec j$ denotes that $i$ precedes $j$ in permutation $\pi$.*

*Proof.* We proceed by induction on $n$.

**Base case ($n = 3$).** With $n = 3$ elements, the probability that $i$ precedes $j$ and $k$ precedes $j$ is:

$$
\begin{aligned}
\Pr(i \prec j, k \prec j) &= \Pr(\pi \in \{(i, k, j), (k, i, j)\}) \\
&= \frac{e^{\theta_i}}{e^{\theta_i} + e^{\theta_j} + e^{\theta_k}}\Big(\frac{e^{\theta_k}}{e^{\theta_j} + e^{\theta_k}}\Big) + \frac{e^{\theta_k}}{e^{\theta_i} + e^{\theta_j} + e^{\theta_k}}\Big(\frac{e^{\theta_i}}{e^{\theta_i} + e^{\theta_j}}\Big) \\
&= \Big(\frac{e^{\theta_i}}{e^{\theta_i} + e^{\theta_j}}\Big)\Big(\frac{e^{\theta_k}}{e^{\theta_j} + e^{\theta_k}}\Big)\Big(\frac{2e^{\theta_j} + e^{\theta_k} + e^{\theta_i}}{e^{\theta_i} + e^{\theta_j} + e^{\theta_k}}\Big) \\
&= \Pr(i \prec j)\Pr(k \prec j)\Big(1 + \frac{e^{\theta_j}}{e^{\theta_i} + e^{\theta_j} + e^{\theta_k}}\Big)
\end{aligned}
$$

**Induction step.** Suppose the statement holds for $n = N - 1$ items. Consider $n = N$ items. We decompose the joint probability $\Pr(i \prec j, k \prec j)$ by the first position of the permutation:

$$\Pr(i \prec j, k \prec j) \;=\; \Pr(\pi_1 \in \{i, k\}, i \prec j, k \prec j) + \Pr(\pi_1 \notin \{i, k\}, i \prec j, k \prec j). \tag{16}$$

Let $C = e^{\theta_i} + e^{\theta_j} + e^{\theta_k}$ and $Z = \sum_{k=1}^{n} e^{\theta_k}$. The first term of Equation 16 can be written as:

$$
\begin{aligned}
\Pr(\pi_1 \in \{i, k\}, i \prec j, k \prec j) &= \Pr(\pi_1 = i, \, i \prec j, k \prec j) + \Pr(\pi_1 = k, \, i \prec j, k \prec j) \\
&= \Pr(\pi_1 = i)\Pr(k \prec j \,|\, \pi_1 = i) + \Pr(\pi_1 = k)\Pr(i \prec j \,|\, \pi_1 = k) \\
&= \frac{e^{\theta_i}}{Z}\Big(\frac{e^{\theta_k}}{e^{\theta_k} + e^{\theta_j}}\Big) + \frac{e^{\theta_k}}{Z}\Big(\frac{e^{\theta_i}}{e^{\theta_i} + e^{\theta_j}}\Big) \\
&= \frac{e^{\theta_i}e^{\theta_k}(e^{\theta_j} + e^{\theta_k}) + e^{\theta_i}e^{\theta_k}(e^{\theta_i} + e^{\theta_j})}{Z(e^{\theta_i} + e^{\theta_j})(e^{\theta_k} + e^{\theta_j})} \\
&= \frac{e^{\theta_i}e^{\theta_k}(C + e^{\theta_j})}{Z(e^{\theta_i} + e^{\theta_j})(e^{\theta_k} + e^{\theta_j})} \\
&= \Pr(i \prec j)\Pr(k \prec j)\Big(\frac{C + e^{\theta_j}}{Z}\Big),
\end{aligned}
$$

where the third and last equalities follow from using Lemma C.1.

The last term of Equation 16, by the induction hypothesis, is:

$$
\begin{aligned}
\Pr(\pi_1 \notin \{i, k\}, \, i \prec j, \, k \prec j) &= \Pr(\pi_1 \notin \{i, j, k\})\Pr(i \prec j, \, k \prec j \,|\, \pi_1 \notin \{i, j, k\}) \\
&= \frac{Z - C}{Z}\Pr(i \prec j)\Pr(k \prec j)\Big(\frac{C + e^{\theta_j}}{C}\Big).
\end{aligned}
$$

Combining the two terms, Equation 16 can be written as:

$$
\begin{aligned}
\Pr(i \prec j, \, k \prec j) &= \frac{1}{Z}\Pr(i \prec j)\Pr(k \prec j)\Big(C + e^{\theta_j} + \frac{(Z - C)(C + e^{\theta_j})}{C}\Big) \\
&= \Pr(i \prec j)\Pr(k \prec j)\Big(\frac{C + e^{\theta_j}}{C}\Big)
\end{aligned}
$$

Thus the claim holds for $n = N$, completing the induction. $\qquad\square$

## C.2. Isotropic Normal models

**Theorem C.4.** *Assume that each conditional distribution is Gaussian with mean $\mu_j$ and scale $\sigma_j$:*

$$p_\Omega^j(X_j \mid M_j', X_{-j}) = \mathcal{N}(X_j \mid \mu_j, \sigma_j^2), \qquad \mu_j = b_j + \sum_{i=1}^n a_{ij} w_{ij} x_i,$$

*where $a_{ij} \in \{0, 1\}$ indicates whether the edge $i \to j$ is present in the DAG, $n$ is the total number of variables, and $w_{ij}$, $b_j$, and $\sigma_j$ are learnable parameters. Assume that the edges $a_{ij}$ are distributed according to our Bernoulli-Plackett-Luce model.*

*Then, the expected log-likelihood score $S(\Theta, \Omega)$ can be expressed as:*

$$S(\Theta, \Omega) = -\frac{1}{2} \sum_{r=1}^R \mathbb{E}_{\boldsymbol{x} \sim P_{data}^{(r)}} \sum_{j \notin \mathcal{I}_r} s(\boldsymbol{x}, j),$$

*with sample- and node-specific scores $s(\boldsymbol{x}, j)$ defined as:*

$$s(\boldsymbol{x}, j) = \log(2\pi\sigma_j^2) + \frac{1}{\sigma_j^2} \left( \left( x_j - b_j - \sum_{i=1}^n \mathbb{E}[a_{ij}] w_{ij} x_i \right)^2 \right.$$
$$+ \sum_{i=1}^n \mathbb{E}[a_{ij}](1 - \mathbb{E}[a_{ij}]) w_{ij}^2 x_i^2$$
$$\left. + \sum_{i=1}^n \sum_{k=1, k \neq i}^n (\mathbb{E}[a_{ij}] w_{ij} x_i)(\mathbb{E}[a_{kj}] w_{kj} x_k) \left( \frac{e^{\theta_j}}{e^{\theta_i} + e^{\theta_j} + e^{\theta_k}} \right) \right),$$

*and expectation $\mathbb{E}[a_{ij}] = p_{ij} \left( \frac{e^{\theta_i}}{e^{\theta_i} + e^{\theta_j}} \right)$ given by the Bernoulli-Plackett-Luce model, where $p_{ij}$ are learnable Bernoulli probabilities and $\theta_i$ and $\theta_j$ are the Plackett-Luce logits for variables $i$ and $j$, respectively.*

*Proof.* Given the score function

$$S(\Theta, \Omega) = \mathbb{E}_{M' \sim M(\theta, \boldsymbol{P})} \left[ \sum_{r=1}^R \mathbb{E}_{\boldsymbol{x} \sim P_{data}^{(r)}} \sum_{j \notin \mathcal{I}_r} \log p_\Omega^j(x_j | M_j', \boldsymbol{x}_{-j}) \right],$$

we can swap the two expectations:

$$S(\Theta, \Omega) = \sum_{r=1}^R \mathbb{E}_{\boldsymbol{x} \sim P_{data}^{(r)}} [l(\boldsymbol{x})], \qquad l(\boldsymbol{x}) = \mathbb{E}_{M' \sim M(\theta, \boldsymbol{P})} \left[ \sum_{j \notin \mathcal{I}_r} \log p_\Omega^j(x_j | M_j', \boldsymbol{x}_{-j}) \right].$$

Here, we defined $l(\boldsymbol{x})$ as the expectation, with respect to the distribution over DAGs, of the interventional log-likelihood evaluated at a single observation $\boldsymbol{x}$. Given that each conditional distribution is a Gaussian, we can rewrite $l(\boldsymbol{x})$ as:

$$l(\boldsymbol{x}) = \mathbb{E}_{M' \sim M(\theta, \boldsymbol{P})} \left[ -\frac{1}{2} \left( \sum_{j \notin \mathcal{I}_r} \log(2\pi) + \log(\sigma_j^2) + \frac{1}{\sigma_j^2} \left( x_j - b_j - \sum_i a_{ij} w_{ij} x_i \right)^2 \right) \right]. \tag{17}$$

We now push the expectation over DAGs inwards:

$$l(\boldsymbol{x}) = -\frac{1}{2} \sum_{j \notin \mathcal{I}_r} \left( \log(2\pi) + \log(\sigma_j^2) + \frac{1}{\sigma_j^2} \mathbb{E}_{M' \sim M(\theta, \boldsymbol{P})} \left[ \left( x_j - b_j - \sum_i a_{ij} w_{ij} x_i \right)^2 \right] \right). \tag{18}$$

Using the identity $\mathbb{E}[X^2] = \mathbb{E}[X]^2 + \text{Var}[X]$, we now rewrite the last term of Equation 18:

$$\mathbb{E}_{M'\sim M(\theta,\boldsymbol{P})}\left[\left(x_j - b_j - \sum_i a_{ij}w_{ij}x_i\right)^2\right] = \left(\mathbb{E}_{M'\sim M(\theta,\boldsymbol{P})}\left[x_j - b_j - \sum_i a_{ij}w_{ij}x_i\right]\right)^2$$
$$+ \operatorname{Var}_{M'\sim M(\theta,\boldsymbol{P})}\left[x_j - b_j - \sum_i a_{ij}w_{ij}x_i\right]. \tag{19}$$

The expectation of Equation 19 can be expressed as:

$$\mathbb{E}_{M'\sim M(\theta,\boldsymbol{P})}\left[x_j - b_j - \sum_i a_{ij}w_{ij}x_i\right] = x_j - b_j - \left(\mathbb{E}_{M'\sim M(\theta,\mathcal{P})}\left[\sum_{i=1,i\neq j}^n a_{ij}w_{ij}x_i\right]\right)$$
$$= x_j - b_j - \left(\sum_{i=1,i\neq j}^n w_{ij}x_i\mathbb{E}_{M'\sim M(\theta,\mathcal{P})}\left[a_{ij}\right]\right) \tag{20}$$
$$= x_j - b_j - \left(\sum_{i=1,i\neq j}^n w_{ij}x_i\left(\frac{e^{\theta_i}}{e^{\theta_i}+e^{\theta_j}}\right)p_{ij}\right),$$

where the last equality follows from the fact that $\mathbb{E}[a_{ij}] = \left(\frac{e^{\theta_i}}{e^{\theta_i}+e^{\theta_j}}\right)p_{ij}$ for the Bernoulli-Plackett-Luce model (Corollary C.2).

The variance in Equation 19 can be expanded as follows:

$$\operatorname{Var}\left[x_j - b_j - \sum_i a_{ij}w_{ij}x_i\right] = \operatorname{Var}\left[\sum_{i=1,i\neq j}^n a_{ij}w_{ij}x_i\right]$$
$$= \sum_{i=1,i\neq j}^n (w_{ij}x_i)^2\operatorname{Var}[a_{ij}] + \sum_{i,k=1,i\neq k\neq j}^n (w_{ij}x_i)(w_{kj}x_k)\operatorname{Cov}(a_{ij},a_{kj})$$
$$= \sum_{i=1,i\neq j}^n (w_{ij}x_i)^2 p_{ij}\left(\frac{e^{\theta_i}}{e^{\theta_i}+e^{\theta_j}}\right)\left(1 - p_{ij}\left(\frac{e^{\theta_i}}{e^{\theta_i}+e^{\theta_j}}\right)\right)$$
$$+ \sum_{i,k=1,i\neq k\neq j}^n (w_{ij}x_i)(w_{kj}x_k)(p_{ij}p_{kj})\left(\frac{e^{\theta_i}}{e^{\theta_i}+e^{\theta_j}}\right)\left(\frac{e^{\theta_k}}{e^{\theta_k}+e^{\theta_j}}\right)\left(\frac{e^{\theta_j}}{e^{\theta_i}+e^{\theta_j}+e^{\theta_k}}\right)$$
$$= \sum_{i=1,i\neq j}^n (w_{ij}x_i)^2\mathbb{E}[a_{ij}]\left(1 - \mathbb{E}[a_{ij}]\right)$$
$$+ \sum_{i=1}^n\sum_{k=1,k\neq i}^n (\mathbb{E}[a_{ij}]w_{ij}x_i)(\mathbb{E}[a_{kj}]w_{kj}x_k)\left(\frac{e^{\theta_j}}{e^{\theta_i}+e^{\theta_j}+e^{\theta_k}}\right)\right). \tag{21}$$

We decomposed the covariance in Equation 21 as follows:

$$\operatorname{Cov}(a_{ij},a_{kj}) = \mathbb{E}[a_{ij}a_{kj}] - \mathbb{E}[a_{ij}]\mathbb{E}[a_{kj}]$$
$$= \Pr(i \prec j, k \prec j)p_{ij}p_{kj} - \Pr(i \prec j)\Pr(k \prec j)p_{ij}p_{kj}$$
$$= p_{ij}p_{kj}\Big(\Pr(i \prec j, k \prec j) - \Pr(i \prec j)\Pr(k \prec j)\Big)$$
$$= p_{ij}p_{kj}\left(\frac{e^{\theta_i}}{e^{\theta_i}+e^{\theta_j}}\right)\left(\frac{e^{\theta_k}}{e^{\theta_k}+e^{\theta_j}}\right)\left(\frac{e^{\theta_j}}{e^{\theta_i}+e^{\theta_k}+e^{\theta_j}}\right),$$

where the last equality follows from applying Lemmas C.1 and C.3.

Plugging Equations 20 and 21 into Equation 19, we obtain the following expected interventional log-likelihood (Equation 17):

$$l(\boldsymbol{x}) = -\tfrac{1}{2} \sum_{j \notin \mathcal{I}_r} \log(2\pi\sigma_j^2) + \frac{1}{\sigma_j^2} \Bigg( \Big( x_j - b_j - \sum_{i=1}^{n} \mathbb{E}[a_{ij}]w_{ij}x_i \Big)^2$$

$$+ \sum_{i=1}^{n} \mathbb{E}[a_{ij}](1 - \mathbb{E}[a_{ij}])w_{ij}^2 x_i^2$$

$$+ \sum_{i=1}^{n} \sum_{k=1,k\neq i}^{n} (\mathbb{E}[a_{ij}]w_{ij}x_i)(\mathbb{E}[a_{kj}]w_{kj}x_k) \Big( \frac{e^{\theta_j}}{e^{\theta_i} + e^{\theta_j} + e^{\theta_k}} \Big) \Bigg),$$

completing the proof. □

## C.3. Multivariate Normal models

In this section, let us show how to extend our result to more general *multivariate* Normal models. Namely, we assume that the variables are now vectors in $d$-dimensional space: $\boldsymbol{x}_1, \dots, \boldsymbol{x}_n \in \mathbb{R}^d$.

**Theorem C.5.** *Assume that each conditional distribution is multivariate Gaussian with mean $\boldsymbol{\mu}_j \in \mathbb{R}^d$ and covariance matrix $\boldsymbol{\Sigma}_j \in \mathbb{R}^{d \times d}$ that is symmetric and positive definite: $\boldsymbol{\Sigma}_j = \boldsymbol{\Sigma}_j^\top \succ 0$:*

$$p_\Omega^j(\boldsymbol{x}_j \,|\, M_j', \boldsymbol{x}_{-j}) \;=\; \mathcal{N}(\boldsymbol{x}_j \,|\, \boldsymbol{\mu}_j, \boldsymbol{\Sigma}_j), \qquad \boldsymbol{\mu}_j \;=\; \boldsymbol{b}_j + \sum_{i=1}^{n} a_{ij} w_{ij} \boldsymbol{x}_i, \tag{22}$$

*where $a_{ij} \in \{0,1\}$ indicated the edge $i \to j$ in the DAG, and $w_{ij}$, $\boldsymbol{b}_j$, and $\boldsymbol{\Sigma}_j$ are learnable parameters. Then, the expected log-likelihood score $S(\Theta, \Omega)$ can be expressed as:*

$$S(\Theta, \Omega) \;=\; -\tfrac{1}{2} \sum_{r=1}^{R} \mathbb{E}_{\boldsymbol{x} \sim P_{data}^{(r)}} \sum_{j \notin \mathcal{I}_r} s(\boldsymbol{x}, j)$$

*with sample- and node-specific scores $s(\boldsymbol{x}, j)$ defined as:*

$$s(\boldsymbol{x}, j) \;=\; d\log(2\pi) + \log\det\boldsymbol{\Sigma}_j + \tfrac{1}{2}\Big\langle \boldsymbol{\Sigma}_j^{-1}\Big( \boldsymbol{x}_j - \boldsymbol{b}_j - \sum_{i=1}^{n} \mathbb{E}[a_{ij}]w_{ij}\boldsymbol{x}_i \Big), \boldsymbol{x}_j - \boldsymbol{b}_j - \sum_{i=1}^{n} \mathbb{E}[a_{ij}]w_{ij}\boldsymbol{x}_i \Big\rangle$$

$$+ \sum_{i=1}^{n} \mathbb{E}[a_{ij}](1 - \mathbb{E}[a_{ij}])w_{ij}^2 \langle \boldsymbol{\Sigma}_j^{-1}\boldsymbol{x}_i, \boldsymbol{x}_i \rangle$$

$$+ \sum_{i=1}^{n} \sum_{k=1,k\neq i}^{n} (\mathbb{E}[a_{ij}]w_{ij})(\mathbb{E}[a_{kj}]w_{kj}) \Big( \frac{e^{\theta_j}}{e^{\theta_i} + e^{\theta_j} + e^{\theta_k}} \Big) \langle \boldsymbol{\Sigma}_j^{-1}\boldsymbol{x}_i, \boldsymbol{x}_k \rangle,$$

*where, as in the previous case, the expectation $\mathbb{E}[a_{ij}] = p_{ij}\Big( \frac{e^{\theta_i}}{e^{\theta_i} + e^{\theta_j}} \Big)$ is given by the Bernoulli-Plackett-Luce model, where $p_{ij}$, $\theta_i$, and $\theta_j$ are learnable parameters.*

*Proof.* The proof repeats the reasoning of Theorem C.4. Let us fix $1 \le r \le R$ and consider the expectation of the log-likelihood evaluated at a single observation $\boldsymbol{x}$:

$$l(\boldsymbol{x}) \;=\; \mathbb{E}_{M' \sim M(\theta, \boldsymbol{P})}\Big[ \sum_{j \notin \mathcal{I}_r} \log p_\theta^j(\boldsymbol{x}_j \,|\, M_j', \boldsymbol{x}_{-j}) \Big]$$

$$\overset{(22)}{=} \;-\tfrac{1}{2} \sum_{j \notin \mathcal{I}_r} \Big( d\log(2\pi) + \log\det\boldsymbol{\Sigma}_j + \mathbb{E}_{M' \sim M(\theta, \boldsymbol{P})}\Big[ \langle \boldsymbol{\Sigma}_j^{-1}(\boldsymbol{x}_j - \boldsymbol{\mu}_j), \boldsymbol{x}_j - \boldsymbol{\mu}_j \rangle \Big] \Big),$$

and it remains to compute the following expectation, denoting $\bar{\boldsymbol{\mu}}_j := \mathbb{E}_{M' \sim M(\theta, \boldsymbol{P})}[\boldsymbol{\mu}_j] = \boldsymbol{b}_j + \sum_{i=1}^{n} \mathbb{E}[a_{ij}] w_{ij} \boldsymbol{x}_i$,

$$
\begin{aligned}
& \mathbb{E}_{M' \sim M(\theta, \boldsymbol{P})} \Big[ \langle \boldsymbol{\Sigma}_j^{-1}(\boldsymbol{x}_j - \boldsymbol{\mu}_j), \boldsymbol{x}_j - \boldsymbol{\mu}_j \rangle \Big] \\
=\ & \mathbb{E}_{M' \sim M(\theta, \boldsymbol{P})} \Big[ \langle \boldsymbol{\Sigma}_j^{-1}(\boldsymbol{x}_j - \bar{\boldsymbol{\mu}}_j + \bar{\boldsymbol{\mu}}_j - \boldsymbol{\mu}_j), \boldsymbol{x}_j - \bar{\boldsymbol{\mu}}_j + \bar{\boldsymbol{\mu}}_j - \boldsymbol{\mu}_j \rangle \Big] \\
=\ & \langle \boldsymbol{\Sigma}_j^{-1}(\boldsymbol{x}_j - \bar{\boldsymbol{\mu}}_j), \boldsymbol{x}_j - \bar{\boldsymbol{\mu}}_j \rangle + \mathbb{E}_{M' \sim M(\theta, \boldsymbol{P})} \Big[ \langle \boldsymbol{\Sigma}_j^{-1}(\boldsymbol{\mu}_j - \bar{\boldsymbol{\mu}}_j), \boldsymbol{\mu}_j - \bar{\boldsymbol{\mu}}_j \rangle \Big],
\end{aligned}
\tag{23}
$$

where we used in the last equation that the expectation of the cross term is zero:

$$
\mathbb{E}_{M' \sim M(\theta, \boldsymbol{P})} \Big[ \langle \boldsymbol{\Sigma}_j^{-1}(\boldsymbol{x}_j - \bar{\boldsymbol{\mu}}_j), \boldsymbol{\mu}_j - \bar{\boldsymbol{\mu}}_j \rangle \Big] = \langle \boldsymbol{\Sigma}_j^{-1}(\boldsymbol{x}_j - \bar{\boldsymbol{\mu}}_j), \mathbb{E}_{M' \sim M(\theta, \boldsymbol{P})} \big[ \boldsymbol{\mu}_j - \bar{\boldsymbol{\mu}}_j \big] \rangle = 0.
$$

It remains to compute the last term in equation 23, which is the variance:

$$
\begin{aligned}
& \mathbb{E}_{M' \sim M(\theta, \boldsymbol{P})} \Big[ \langle \boldsymbol{\Sigma}_j^{-1}(\boldsymbol{\mu}_j - \bar{\boldsymbol{\mu}}_j), \boldsymbol{\mu}_j - \bar{\boldsymbol{\mu}}_j \rangle \Big] \\
=\ & \mathbb{E}_{M' \sim M(\theta, \boldsymbol{P})} \Big[ \langle \boldsymbol{\Sigma}_j^{-1} \sum_{i=1}^{n}(a_{ij} - \mathbb{E}[a_{ij}]) w_{ij} \boldsymbol{x}_i, \sum_{i=1}^{n}(a_{ij} - \mathbb{E}[a_{ij}]) w_{ij} \boldsymbol{x}_i \rangle \Big] \\
=\ & \sum_{i=1}^{n} \mathrm{Var}(a_{ij}) w_{ij}^2 \langle \boldsymbol{\Sigma}_j^{-1} \boldsymbol{x}_i, \boldsymbol{x}_i \rangle + \sum_{i=1}^{n} \sum_{k=1, k \neq i}^{n} \mathrm{Cov}(a_{ij}, a_{kj}) w_{ij} w_{kj} \langle \boldsymbol{\Sigma}_j^{-1} \boldsymbol{x}_i, \boldsymbol{x}_k \rangle.
\end{aligned}
$$

To complete the proof, we use the formulas, $\mathrm{Var}(a_{ij}) = \mathbb{E}[a_{ij}](1 - \mathbb{E}[a_{ij}])$ and $\mathrm{Cov}(a_{ij}, a_{kj}) = \mathbb{E}[a_{ij}] \mathbb{E}[a_{kj}] \big( \frac{e^{\theta_j}}{e^{\theta_i} + e^{\theta_j} + e^{\theta_k}} \big)$, that were established in Theorem C.4. $\qquad\square$

### C.4. Second-order approximation for general models

Let us discuss a general technique that estimates the expectation of the model with respect to the Bernoulli-Plackett-Luce distribution *beyond Normal assumption*, as in the previous section. Our approach is based on *second-order* approximation of the log-likelihood function as a function of the adjacency matrix $\boldsymbol{A} = (a_{ij}) \in \{0, 1\}^{n \times n}$ around *null graph*, $\bar{\boldsymbol{A}} = \boldsymbol{0} \in \mathbb{R}^{n \times n}$, that is the *model with no causal relationships*. In the particular case of Normal models, our derivations recover the exact formulas from the previous sections. However, these estimates can be used for a gerenal, non-Gaussian setting.

Recall that our unconstrained objective (excluding the regularizer for simplicity) is given by:

$$
S(\Theta, \Omega) = \mathbb{E}_{\boldsymbol{A} \sim M(\Theta)} \Big[ f(M', \Omega) = \sum_{r=1}^{R} \mathbb{E}_{X \sim P_{\mathrm{data}}^{(r)}} \sum_{j \notin \mathcal{I}_r} \log p_\Omega^j (X_j | \boldsymbol{A}_j, X_{-j}) \Big],
\tag{24}
$$

where $\boldsymbol{A}_j$ is the $j$-th column of the sampled adjacency matrix. Denote, for a fixed $1 \leq r \leq R$ and $j \notin \mathcal{I}_r$, the log-likelihood as a function of $\boldsymbol{a} := \boldsymbol{A}_j \in \mathbb{R}^n$

$$
\varphi(\boldsymbol{a}) := \log p_\Omega^j (X_j | \boldsymbol{a}, X_{-j}),
\tag{25}
$$

and consider its second-order Taylor expansion around $\boldsymbol{a} := \boldsymbol{0}$:

$$
\varphi(\boldsymbol{a}) \approx q(\boldsymbol{a}) := \varphi(\boldsymbol{0}) + \langle \nabla \varphi(\boldsymbol{0}), \boldsymbol{a} \rangle + \tfrac{1}{2} \langle \nabla^2 \varphi(\boldsymbol{0}) \boldsymbol{a}, \boldsymbol{a} \rangle.
\tag{26}
$$

Note that for Normal models from the previous sections, this approximation is *exact*. The value $\varphi(\boldsymbol{0})$ corresponds to the log-likelihood of our model with no relations between $j$-th variable and others:

$$
\varphi(\boldsymbol{0}) := \log p_\Omega^j (X_j | \boldsymbol{0}, X_{-j}) = \log p_\Omega^j (X_j)
$$

and can be seen as an apriori distribution with no causal relationships, modeling the data for a fixed value of parameters $\Omega$. Using automatic differentiation, we can compute the gradient $\nabla \varphi(\boldsymbol{0}) \in \mathbb{R}^n$ and the Hessian matrix $\nabla^2 \varphi(\boldsymbol{0}) \in \mathbb{R}^{n \times n}$ at zero, for a specifically given model in (25).

**Example 1.** *Consider the multivariate Normal model (22). In this case, $\varphi(\boldsymbol{a})$ can be expressed as*

$$\varphi(\boldsymbol{a}) = -\tfrac{1}{2}\langle \boldsymbol{\Sigma}^{-1}(\bar{\boldsymbol{X}}\boldsymbol{a} + \boldsymbol{b}_j), \bar{\boldsymbol{X}}\boldsymbol{a} + \boldsymbol{b}_j \rangle + c,$$

*where $c = -\tfrac{d}{2}\log(2\pi) - \log\det\boldsymbol{\Sigma}$ is the normalizing constant, and the matrix $\bar{\boldsymbol{X}} \in \mathbb{R}^{d\times n}$ is composed by a reweighted data:*

$$\bar{\boldsymbol{X}} = \left[ w_{1j}\boldsymbol{x}_1 \,\middle|\, \dots \,\middle|\, w_{nj}\boldsymbol{x}_n \right].$$

*Note that in this case,*

$$\varphi(\boldsymbol{0}) = -\tfrac{1}{2}\langle \boldsymbol{\Sigma}^{-1}\boldsymbol{b}_j, \boldsymbol{b}_j \rangle + c$$

*is the log-likelihood of our model with no causalities, and the gradient and the Hessian at zero can be computed as:*

$$\nabla\varphi(\boldsymbol{0}) = -\bar{\boldsymbol{X}}^\top\boldsymbol{\Sigma}^{-1}(\boldsymbol{b}_j - \boldsymbol{x}_j) \in \mathbb{R}^n,$$

$$\nabla^2\varphi(\boldsymbol{0}) = -\bar{\boldsymbol{X}}^\top\boldsymbol{\Sigma}^{-1}\bar{\boldsymbol{X}} \in \mathbb{R}^{n\times n}.$$

*Note that both $\nabla\varphi(\boldsymbol{0})$ and $\nabla^2\varphi(\boldsymbol{0})$ do not depend on a sampled adjacency matrix $\boldsymbol{A}$, and describe the local geometry of the log-likelihood with no causal relationships. In this example, expansion (26) is exact.* $\square$

Note that in general, (26) is a local approximation of the log-likelihood. However, under mild assumptions on our model, we can equip it with a *global guarantee*, as in the following lemma.

**Lemma C.6.** *Let the third derivative of (25) be bounded, for some $L \geq 0$:*

$$L \geq \|\nabla^3\varphi(\boldsymbol{a})\| \equiv \|\nabla^3_{\boldsymbol{a}} \log p^j_\Omega(X_j \mid \boldsymbol{a}, X_{-j})\|. \tag{27}$$

*Then,*

$$|\varphi(\boldsymbol{a}) - q(\boldsymbol{a})| \leq \tfrac{L}{6}n^{3/2}. \tag{28}$$

*Proof.* The bound immediately follows from the integral form of the Taylor theorem,

$$|\varphi(\boldsymbol{a}) - q(\boldsymbol{a})| = |\varphi(\boldsymbol{a}) - \varphi(\boldsymbol{0}) - \langle\nabla\varphi(\boldsymbol{0}), \boldsymbol{a}\rangle - \tfrac{1}{2}\langle\nabla^2\varphi(\boldsymbol{0})\boldsymbol{a}, \boldsymbol{a}\rangle|$$

$$= \tfrac{1}{2}\Big|\int_0^1 (1-\tau)^2 \langle\nabla^3\varphi(\tau\boldsymbol{a})\boldsymbol{a}, \boldsymbol{a}, \boldsymbol{a}\rangle\Big| d\tau$$

$$\leq \tfrac{1}{2}\int_0^1 (1-\tau)^2 \|\nabla^3\varphi(\tau\boldsymbol{a})\| \cdot \|\boldsymbol{a}\|^3 d\tau \overset{(27)}{\leq} \tfrac{L}{6}\|\boldsymbol{a}\|^3.$$

It remains to note that, since all entries of the adjacency matrix are from $\{0,1\}$, we have

$$\|\boldsymbol{a}\|^3 = \Big(\sum_{i=1}^n a_i^2\Big)^{3/2} \leq n^{3/2},$$

which completes the proof of (28). $\square$

Now, we observe that we can compute an *explicit formula* for the expectation of the second-order approximate score, $\mathbb{E}_{\boldsymbol{A}\sim M(\Theta)}[q(\boldsymbol{A}_j)]$, under the Bernoulli-Plackett-Luce distribution, using our previous results:

$$\mathbb{E}_{\boldsymbol{A}\sim M(\Theta)}[q(\boldsymbol{A}_j)] = \mathbb{E}_{\boldsymbol{A}\sim M(\Theta)}\big[\varphi(\boldsymbol{0}) + \langle\nabla\varphi(\boldsymbol{0}), \boldsymbol{A}_j\rangle + \tfrac{1}{2}\langle\nabla^2\varphi(\boldsymbol{0})\boldsymbol{A}_j, \boldsymbol{A}_j\rangle\big]$$

$$= \varphi(\boldsymbol{0}) + \sum_{i=1}^n \mathbb{E}[a_{ij}]\cdot(\nabla\varphi(\boldsymbol{0}))_i + \tfrac{1}{2}\sum_{i=1}^n\sum_{k=1}^n \mathbb{E}[a_{ij}a_{kj}](\nabla^2\varphi(\boldsymbol{0}))_{ik}$$

$$= \varphi(\boldsymbol{0}) + \sum_{i=1}^n \mathbb{E}[a_{ij}]\cdot\big((\nabla\varphi(\boldsymbol{0}))_i + \tfrac{1}{2}(\nabla^2\varphi(\boldsymbol{0}))_{ii}\big) \tag{29}$$

$$+ \sum_{1\leq i<k\leq n} \mathbb{E}[a_{ij}]\cdot\mathbb{E}[a_{kj}]\cdot\Big(1 + \frac{e^{\theta_j}}{e^{\theta_i}+e^{\theta_j}+e^{\theta_k}}\Big)\cdot(\nabla^2\varphi(\boldsymbol{0}))_{ik},$$

where we have the exact expression for the expecation $\mathbb{E}[a_{ij}]$.

Therefore, we can use exact formula (29) for the approximation of our score, and Lemma C.6 provides us with the following uniform bound:

$$\left| \mathbb{E}_{\boldsymbol{A}\sim M(\Theta)}[\varphi(\boldsymbol{A}_j)] - \mathbb{E}_{\boldsymbol{A}\sim M(\Theta)}\big[q(\boldsymbol{A}_j)\big] \right| \leq \frac{L}{6}n^{3/2}.$$

Combining these observations together, we establish the following approximation result.

**Theorem C.7.** *Let the third derivative of the log-likelihood function be bounded by $L \geq 0$, as in (27). Then, the expected log-likelihood score (24) under the Bernoulli-Plackett-Luce distribution over DAGs, can be approximated by*

$$\bar{S}(\Theta,\Omega) \quad := \quad \sum_{r=1}^{R} \mathbb{E}_{X\sim P_{data}^r} \sum_{j\notin\mathcal{I}_r} \ell_{\Omega}^j(X), \tag{30}$$

*where*

$$
\ell_{\Omega}^j(X) \quad := \quad \log p_{\Omega}^j(X_j \,|\, \mathbf{0}, X_{-j}) + \sum_{i=1}^{n} \mathbb{E}[a_{ij}] \cdot \left( \frac{\partial}{\partial a_{ij}} \log p_{\Omega}^j(X_j \,|\, \mathbf{0}, X_{-j}) + \frac{1}{2}\frac{\partial^2}{\partial a_{ij}^2} \log p_{\Omega}^j(X_j \,|\, \mathbf{0}, X_{-j}) \right)
$$

$$
+ \sum_{1\leq i<j\leq n} \mathbb{E}[a_{ij}] \cdot \mathbb{E}[a_{kj}] \cdot \left( 1 + \frac{e^{\theta_j}}{e^{\theta_i}+e^{\theta_j}+e^{\theta_k}} \right) \cdot \frac{\partial^2}{\partial a_{ij}\partial a_{kj}} \log p_{\Omega}^j(X_j \,|\, \mathbf{0}, X_{-j}), \tag{31}
$$

*where* $\mathbb{E}[a_{ij}] = p_{ij}\left( \frac{e^{\theta_i}}{e^{\theta_i}+e^{\theta_j}} \right)p_{ij}$. *And we have the global approximation bound:*

$$|S(\Theta,\Omega) - \bar{S}(\Theta,\Omega)| \quad \leq \quad \delta \quad := \quad \sum_{r=1}^{R} |\mathcal{I}_r| \cdot \frac{L}{6}n^{3/2}. \tag{32}$$

Note that the value of $\delta$ in (32) can be improved if we replace the Taylor expansion around $\mathbf{0}$ (no causal relationships) in (26) to a graph $\boldsymbol{A} \sim M(\Theta)$ sampled from the Bernoulli-Plackett-Luce distribution using the latest values of the trained parameters $\Theta$. In case of Normal models, we have $L = 0$, which gives exact bound in (32).

In order to use approximation (31), we have to compute the Hessian of the score: $\left[ \frac{\partial^2}{\partial a_{ij}\partial a_{kj}} \log p_{\Omega}^j(X_j \,|\, \mathbf{0}, X_{-j}) \right]_{ik} \in \mathbb{R}^{n\times n}$, which, in general, requires computing $O(n^2)$ entries, for each of $1 \leq j \leq n$ nodes. Therefore, this approach is less scalable than REINFORCE estimator, and we keep it for future research.

## D. Runtime and memory complexity

**Computational complexity.** PACER can be trained using either a REINFORCE estimator or the analytic objective (Theorem 4.4). In the former version, each optimisation step involves $M$ Monte Carlo samples of permutations and edge masks. For every sample, the method constructs a permutation, builds the DAG mask, evaluates the likelihood-based objective on a minibatch of size $B$ and performs backpropagation through the neural networks. The overall complexity of each step is $\mathcal{O}(MBn^2)$, where $n$ is the number of variables. The analytic version does not require Monte Carlo sampling and directly optimizes the expected objective. The main cost comes from the covariance term which is cubic with respect to the number of nodes. However, at each step we sample $n^{2/3}$ nodes keeping the complexity quadratic. Consequently, the analytic method has a per-step complexity of $\mathcal{O}(Bn^2)$. In both cases, considering $M$ and $B$ constants, PACER has a computational complexity of $\mathcal{O}(n^2)$.

**Memory complexity.** PACER's memory complexity is dominated by storing model parameters and intermediate activations. In particular, the adjacency-related parameters (*i.e.*, edge probabilities and weights) scale as $\mathcal{O}(n^2)$, which is standard for methods that model dense graphs. During training, additional memory is required for mini-batch data and gradients, resulting in an overall memory complexity of $\mathcal{O}(Bn + n^2)$, where $B$ is the batch size and $n$ the number of variables. For the REINFORCE-based variant, storing $M$ Monte Carlo samples per mini-batch example introduces an additional $\mathcal{O}(MBn^2)$ factor in the worst case, although in practice samples could, if necessary, be processed sequentially to keep memory usage effectively at $\mathcal{O}(n^2)$. The analytic linear-Gaussian variant does not require Monte Carlo sampling and therefore avoids this overhead.

# E. Baselines

**Summary of baselines.** We compare PACER with a range of established causal discovery methods spanning constraint-based, score-based, and differentiable approaches. Constraint-based methods include GS (Margaritis, 2003), IAMB (Tsamardinos et al., 2003a), and MMPC (Tsamardinos et al., 2003b), which infer graph structure via conditional independence tests. Score-based methods include GES (Chickering, 2002), GIES (Hauser & Bühlmann, 2012), GRaSP (Lam et al., 2022), BOSS (Andrews et al., 2023), and CAM (Bühlmann et al., 2014), which search for graphs that optimize a likelihood-based or penalized score. LiNGAM (Shimizu et al., 2006) exploits non-Gaussianity to identify causal directions under linear models. Sortnregress (Reisach et al., 2021) sorts variables by increasing marginal variance and uses parent search to infer DAG structure. NOTEARS (Zheng et al., 2018), NOTEARS-LR (Fang et al., 2023), DCDI (Brouillard et al., 2020) formulate structure learning as continuous optimization problems with acyclicity constraints, with several variants explicitly leveraging interventional data. DCDFG (Lopez et al., 2022) is a recent method designed for large-scale perturbation settings that jointly models causal structure and interventional effects. SDCD (Nazaret et al., 2024) is a method that improved stability and efficiency of DCD-based methods using an alternative acyclicity constraint.

**Baselines implementations.** GS, GES, GIES, IAMB, MMPC, GRaSP, BOSS, and LiNGAM are benchmarked using the implementation of the Causal Discovery Toolbox (Kalainathan et al., 2020). We use the original implementation of NO-TEARS (Zheng et al., 2018). For the CausalBench benchmark, we use the default CausalBench implementations (Chevalley et al., 2025). In terms of the Perturb-CITE-seq results, we use the implementations of DCDFG, NOTEARS, and NOTEARS from the DCDFG codebase (Lopez et al., 2022). For the synthetic dataset from SDCD and DCDI, we used the default implementations of the baseline models. Unless otherwise stated, we use the default hyperparameters.

*Table 3.* Summary of differentiable baseline methods. We highlight the complexity of their acyclicity constraints and whether they can incorporate prior knowledge. $d$: number of variables, $m$: number of factors.

| Method | Acyclicity constraint | Scalable constraint | Interventions | Prior knowledge |
|---|---|---|---|---|
| NOTEARS | ✓ | $\mathcal{O}(d^3)$ | X | X |
| NO-BEARS | ✓ | $\mathcal{O}(d^2)$ | X | X |
| DAGMA | ✓ | $\mathcal{O}(d^3)$ | X | X |
| DCDI | ✓ | $\mathcal{O}(d^3)$ | ✓ | X |
| DCDFG | ✓ | $\mathcal{O}(md)$ | ✓ | X |
| SDCD | ✓ | $\mathcal{O}(d^2)$ | ✓ | X |
| **PACER** | X | $\mathcal{O}(1)$ | ✓ | ✓ |

# F. PACER hyperparameters

We use the following default hyperparameters across all experiments throughout the manuscript unless stated otherwise:

- Number of layers for each variable-specific MLP: 2

- Layer dimension: 4

- Number of training steps: 5,000

- Batch size: 64

- Learning rate: 0.01

- Number of Monte Carlo samples: 200

- Edge regularization coefficient $\lambda$: 1

We use the default threshold of 0.5 on the expected edge probabilities (Equation 15) to define the inferred graph. PACER uses 20% of the input data to select the checkpoint that minimizes the validation loss. For the analytic version of PACER we lowered the learning rate to 0.001 and increased the number of training steps to 20,000. For the synthetic dataset experiments, we adopt hyperparameters similar to prior work to ensure a fair comparison in terms of runtime and scalability. We follow the settings used in DCDI and SDCD, with a learning rate of 0.001, hidden layer dimension of 16, batch size of 256, and 10,000 training steps.

# G. Synthetic dataset

We evaluate PACER on two synthetic benchmarks for causal discovery with interventions: the DCDI synthetic dataset and SDCD synthetic dataset, which differ in both their data-generating mechanisms and intervention designs.

The DCDI synthetic dataset evaluates causal discovery under diverse causal mechanisms to compare robustness and is commonly used to benchmark differentiable and constraint-based approaches.

For each dataset, a DAG is sampled from an Erdős–Rényi distribution. Interventions are applied differently depending on graph size. For 10-node graphs, single-node interventions are performed on all variables, while for 20-node graphs, interventions target one or two nodes selected uniformly at random. Samples are evenly distributed across interventional settings and subsequently normalized.

Three types of causal mechanisms are considered:

- Linear mechanisms: variables are generated as linear combinations of their parents with additive Gaussian noise.

- Additive noise models (ANM): causal functions are nonlinear neural networks with additive noise.

- Nonlinear non-additive (NN) mechanisms: noise enters the causal function non-additively.

We use perfect interventions, which replace the distribution of intervened nodes with a shifted Gaussian.

The SDCD synthetic dataset is designed to evaluate causal discovery under large-scale and nonlinear settings to compare scalability. Data are generated according to the following steps.

First, an undirected graph is sampled from an Erdős–Rényi distribution and converted into a ground-truth DAG $G$ by assiging edge directions according to a random node permutation. Each node $j$ is associated with a nonlinear causal mechanism parameterized by a fully connected MLP with one hidden layer of 100 units, defining the conditional mean of the observational distribution:

$$p_j(x_j|x_{\mathrm{pa}_G(j)};0) \sim \mathcal{N}(\mathrm{MLP}^{(j)}(x_{\mathrm{pa}_G(j)}), 0.5) \tag{33}$$

Hard interventions are applied by replacing the conditional distribution of the intervened variables $k$ with

$$p_k(x_k;k) \sim \mathcal{N}(0, 0.1) \tag{34}$$

The data comprise 10,000 observational samples and 500 interventional samples per variable, followed by standardization. To assess scalability, we vary the number of variables $d$ while fixing the edge density to $p = 0.05$.

# H. Evaluation metrics

We employ different evaluation metrics depending on the availability of ground-truth causal structure and the scale of the dataset, following established practices for each benchmark.

**Structure recovery metrics (Sachs *et al.*).**    For the (Sachs et al., 2005) protein signaling dataset, a curated ground-truth causal graph is available. We therefore use standard structure recovery metrics that directly compare the inferred graph to the ground truth:

- **Structural Hamming Distance (SHD).** SHD counts the minimum number of edge additions, deletions, and reversals required to transform the inferred graph into the ground-truth graph. Lower values indicate better structural recovery.

- **Structural Intervention Distance (SID).** The Structural Intervention Distance quantifies the closeness between two graphs in terms of their corresponding causal inference statements (Peters & Bühlmann, 2015).

- **False Discovery Rate (FDR).** FDR is the fraction of predicted edges that are not present in the ground-truth graph, measuring the propensity of a method to introduce spurious causal relations.

- **True Positive Rate (TPR).** TPR (recall) is the fraction of ground-truth edges that are correctly recovered by the inferred graph.

- **F1 score.** The F1 score is the harmonic mean of precision (1-FDR) and recall (TPR), providing a single summary statistic that balances false positives and false negatives.

**CausalBench metrics (large-scale perturbation data).** For the CausalBench benchmark (Chevalley et al., 2025), complete ground-truth causal graphs are not available at scale. Instead, evaluation is based on how well inferred graphs explain observed interventional effects. We therefore adopt the metrics proposed in the benchmark:

- **Mean Wasserstein distance.** This metric measures the agreement between the distributional shifts induced by interventions in the data and those implied by the inferred causal graph. Higher values indicate better recovery of intervention-induced effects.

- **False Omission Rate (FOR).** FOR quantifies the fraction of known or curated interactions that are missed by the inferred graph, measuring the tendency of a method to omit relevant causal edges.

These metrics emphasize functional correctness under interventions rather than exact structural recovery, which is appropriate for large-scale gene regulatory networks where only partial ground truth is available.

**Predictive interventional metrics (Perturb-CITE-seq).** For the Perturb-CITE-seq dataset (Frangieh et al., 2021), no ground-truth causal graph is provided. Following DCDFG (Lopez et al., 2022), we therefore evaluate models based on their ability to predict held-out interventional outcomes:

- **Interventional negative log-likelihood (I-NLL).** I-NLL measures how well the learned causal model explains observed data under unseen interventions. Lower values indicate better predictive performance.

- **Interventional mean absolute error (I-MAE).** I-MAE evaluates the absolute deviation between predicted and observed intervention responses, providing an interpretable measure of prediction accuracy.

These metrics assess the quality of the learned causal mechanisms in terms of predictive validity under interventions, which is the primary objective in the absence of a known ground-truth graph.

## I. Scalability experiments

We evaluate six interventional causal discovery models on the SDCD synthetic dataset across a wide range of problem sizes, varying the number of variables $d$ while fixing the edge density to $0.05$. For each setting, experiments are conducted over three randomly generated datsets. We report SHD, SID, precision, recall for $d \in \{10, 20, 30, 40, 50\}$, and measure training time for $d \in \{10, 20, 30, 40, 50, 100, 200, 300\}$. Runs exceeding six hours are treated as timeouts (Figure 3, 10, Table 4).

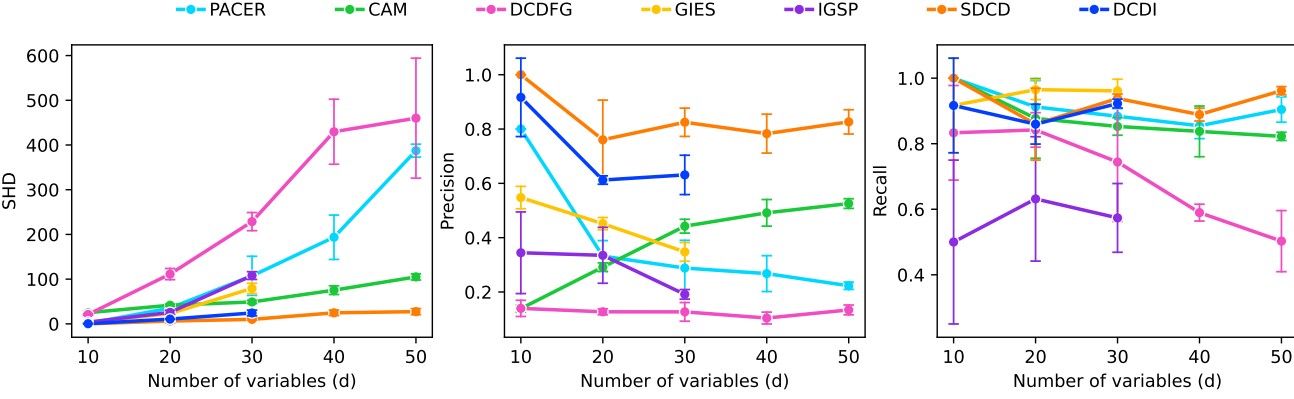

*Figure 10.* Performance across simulations on interventional dataset with increasing numbers of variables $d \leq 50$. SHD, SID, precision, and recall are reported under interventional settings with a fixed edge density of 0.05. Lower values are better for SHD, while higher values are better for precision and recall. Error bars indicate the standard deviation over three random datasets. Missing data points indicate runs that exceeded the 6 hr time limit.

*Table 4.* Average runtime (seconds) and standard deviation across different numbers of variables $d$ on the SDCD synthetic dataset. Runs exceeding six hours are treated as timeouts and marked as "–".

| $d$ | CAM | DCDFG | DCDI | GIES | IGSP | PACER | SDCD |
|---|---|---|---|---|---|---|---|
| 10 | $270.2_{\pm 57.9}$ | $521.5_{\pm 16.9}$ | $837.8_{\pm 102.6}$ | $0.2_{\pm 0.0}$ | $0.0_{\pm 0.0}$ | $507.5_{\pm 276.8}$ | $1645.6_{\pm 261.9}$ |
| 20 | $831.6_{\pm 129.6}$ | $702.3_{\pm 281.6}$ | $4271.9_{\pm 457.1}$ | $12.1_{\pm 2.1}$ | $1.8_{\pm 2.4}$ | $705.1_{\pm 264.4}$ | $2530.8_{\pm 67.4}$ |
| 30 | $954.5_{\pm 156.8}$ | $1476.6_{\pm 63.4}$ | $6654.6_{\pm 754.7}$ | $96.1_{\pm 38.8}$ | $96.9_{\pm 96.0}$ | $654.6_{\pm 220.1}$ | $2949.3_{\pm 154.6}$ |
| 40 | $1959.6_{\pm 97.7}$ | $2080.0_{\pm 81.1}$ | – | – | – | $610.5_{\pm 35.2}$ | $3594.7_{\pm 418.0}$ |
| 50 | $2750.5_{\pm 311.6}$ | $2077.7_{\pm 755.7}$ | – | – | – | $862.9_{\pm 138.7}$ | $4440.1_{\pm 537.6}$ |
| 100 | $8415.7_{\pm 512.1}$ | $2556.6_{\pm 289.0}$ | – | – | – | $3230.2_{\pm 568.6}$ | $6188.6_{\pm 38.8}$ |
| 200 | – | – | – | – | – | $1018.5_{\pm 170.0}$ | $9999.1_{\pm 371.9}$ |
| 300 | – | – | – | – | – | $1877.4_{\pm 628.1}$ | – |

# J. Robustness Experiments

## J.1. Sensitivity to the sparsity coefficient $\lambda$

We perform additional experiments using Erdos-Renyi (ER) graphs of varying average degree (degree $\in \{1, 2, 3, 4\}$), sweeping $\lambda$ over $\{0.01, 0.1, 0.5, 1.0, 2.0\}$ and reporting SHD and SID averaged over 10 random seeds ($\pm$ standard deviation). Tables 5 and 6 summarize the results. PACER's performance is stable across a wide range of $\lambda$ values, with only minor sensitivity in dense graphs.

*Table 5.* SHD for varying $\lambda$ and ER graph average degree. Results are mean $\pm$ std over 10 random seeds. **Bold**: best (lowest SHD) per column.

| $\lambda \setminus$ **degree** | **degree = 1** | **degree = 2** | **degree = 3** | **degree = 4** |
|---|---|---|---|---|
| 0.01 | $32.4 \pm 3.9$ | $36.8 \pm 2.1$ | $31.8 \pm 2.3$ | $30.4 \pm 2.5$ |
| 0.1 | $24.2 \pm 7.7$ | $34.4 \pm 3.9$ | $32.2 \pm 1.7$ | $\mathbf{29.8 \pm 2.8}$ |
| 0.5 | $18.0 \pm 6.9$ | $29.4 \pm 2.6$ | $31.4 \pm 2.1$ | $31.4 \pm 2.9$ |
| 1.0 | $15.4 \pm 6.2$ | $24.2 \pm 2.8$ | $30.6 \pm 1.5$ | $31.4 \pm 3.1$ |
| 2.0 | $\mathbf{12.2 \pm 5.0}$ | $\mathbf{21.2 \pm 4.8}$ | $\mathbf{28.4 \pm 2.9}$ | $33.6 \pm 2.4$ |

*Table 6.* SID for varying $\lambda$ and ER graph average degree. Results are mean $\pm$ std over 10 random seeds. **Bold**: best (lowest SID) per column.

| $\lambda \setminus$ **degree** | **degree = 1** | **degree = 2** | **degree = 3** | **degree = 4** |
|---|---|---|---|---|
| 0.01 | $31.2 \pm 16.5$ | $\mathbf{56.6 \pm 11.5}$ | $\mathbf{69.2 \pm 3.7}$ | $\mathbf{73.0 \pm 2.4}$ |
| 0.1 | $33.4 \pm 16.1$ | $56.6 \pm 11.6$ | $70.0 \pm 3.1$ | $73.2 \pm 2.7$ |
| 0.5 | $31.2 \pm 19.6$ | $60.0 \pm 12.0$ | $72.2 \pm 0.7$ | $74.0 \pm 5.2$ |
| **1.0** | $\mathbf{28.6 \pm 18.2}$ | $62.2 \pm 13.8$ | $72.4 \pm 5.4$ | $73.4 \pm 5.4$ |
| 2.0 | $33.0 \pm 19.4$ | $62.2 \pm 9.5$ | $75.8 \pm 8.1$ | $77.6 \pm 6.0$ |

## J.2. Intervention robustness

To assess robustness with respect to different intervention types, we also report results under multiple intervention settings following DCDI. For these experiments, we followed the hyperparameter setting from DCDI having learning rate=0.001, layer dimension=16, batch size=256, number of training steps=10,000, and choosing the best edge regularization coefficient $\lambda$ based on the validation set.

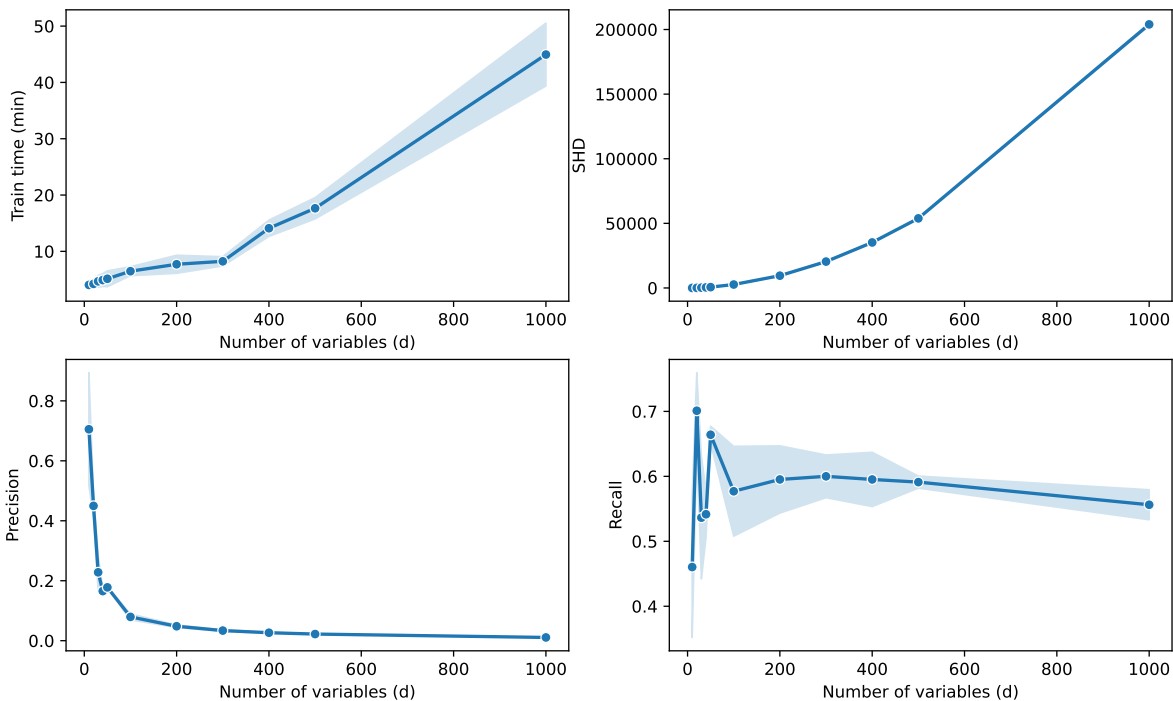

*Figure 11.* Performance of PACER across simulations on interventional linear dataset with increasing numbers of variables $d \leq 1000$. Runtime, SHD, precision, and recall are reported under interventional settings with a fixed connectivity of 4. Lower values are better for SHD, while higher values are better for precision, and recall. Error bars indicate the standard deviation over three random datasets.

*Table 7.* Results for linear dataset with perfect interventions. Lower SHD and SID indicate better performance. Best scores are bolded and second best scores are underlined.

| Method | 10 nodes, $e=1$ | | 10 nodes, $e=4$ | | 20 nodes, $e=1$ | | 20 nodes, $e=4$ | |
|---|---|---|---|---|---|---|---|---|
| | SHD | SID | SHD | SID | SHD | SID | SHD | SID |
| IGSP | $4.0_{\pm 4.8}$ | $15.7_{\pm 15.4}$ | $28.8_{\pm 2.0}$ | $72.2_{\pm 5.1}$ | $9.7_{\pm 8.7}$ | $45.1_{\pm 45.4}$ | $68.1_{\pm 13.6}$ | $295.4_{\pm 27.6}$ |
| GIES | $\mathbf{0.3}_{\pm \mathbf{0.5}}$ | $\mathbf{0.0}_{\pm \mathbf{0.0}}$ | $4.0_{\pm 6.5}$ | $\mathbf{6.7}_{\pm \mathbf{17.7}}$ | $\mathbf{1.5}_{\pm \mathbf{1.2}}$ | $\mathbf{0.3}_{\pm \mathbf{0.9}}$ | $49.4_{\pm 22.2}$ | $\mathbf{111.9}_{\pm \mathbf{51.4}}$ |
| CAM | $0.6_{\pm 1.0}$ | $\mathbf{0.0}_{\pm \mathbf{0.0}}$ | $11.8_{\pm 4.3}$ | $32.2_{\pm 17.2}$ | $6.3_{\pm 7.4}$ | $7.6_{\pm 9.8}$ | $91.4_{\pm 21.3}$ | $181.7_{\pm 60.5}$ |
| DCD | $6.3_{\pm 3.4}$ | $14.8_{\pm 10.6}$ | $26.1_{\pm 3.3}$ | $66.4_{\pm 11.4}$ | $11.1_{\pm 4.7}$ | $45.8_{\pm 22.8}$ | $49.0_{\pm 12.0}$ | $258.6_{\pm 41.6}$ |
| DCDI-G | $\underline{0.4}_{\pm 0.7}$ | $1.3_{\pm 2.1}$ | $\underline{7.5}_{\pm 1.4}$ | $\underline{29.7}_{\pm 8.2}$ | $3.2_{\pm 3.2}$ | $12.1_{\pm 11.2}$ | $\mathbf{21.0}_{\pm \mathbf{4.9}}$ | $\underline{147.6}_{\pm 49.5}$ |
| PACER | $1.5_{\pm 1.3}$ | $3.5_{\pm 4.0}$ | $10.6_{\pm 3.2}$ | $33.1_{\pm 8.4}$ | $\underline{2.5}_{\pm 2.3}$ | $\underline{6.9}_{\pm 9.1}$ | $\underline{42.5}_{\pm 8.6}$ | $190.1_{\pm 39.5}$ |

*Table 8.* Results for nonlinear additive noise model dataset with perfect interventions. Lower SHD and SID indicate better performance. Best scores are bolded and second best scores are underlined.

| Method | 10 nodes, $e=1$ | | 10 nodes, $e=4$ | | 20 nodes, $e=1$ | | 20 nodes, $e=4$ | |
|---|---|---|---|---|---|---|---|---|
| | SHD | SID | SHD | SID | SHD | SID | SHD | SID |
| IGSP | $5.7_{\pm 2.3}$ | $23.4_{\pm 13.6}$ | $32.8_{\pm 2.4}$ | $79.3_{\pm 3.2}$ | $14.9_{\pm 8.1}$ | $78.8_{\pm 64.6}$ | $80.5_{\pm 6.4}$ | $337.6_{\pm 27.3}$ |
| GIES | $7.5_{\pm 5.1}$ | $2.3_{\pm 2.5}$ | $9.2_{\pm 2.9}$ | $27.1_{\pm 11.5}$ | $23.8_{\pm 18.4}$ | $\underline{3.1}_{\pm 4.4}$ | $89.6_{\pm 14.7}$ | $143.9_{\pm 53.1}$ |
| CAM | $6.3_{\pm 6.9}$ | $\mathbf{0.0}_{\pm \mathbf{0.0}}$ | $\underline{6.3}_{\pm 3.8}$ | $\mathbf{14.6}_{\pm \mathbf{20.1}}$ | $9.2_{\pm 14.3}$ | $13.5_{\pm 25.1}$ | $106.2_{\pm 14.6}$ | $\underline{96.2}_{\pm 57.9}$ |
| DCD | $6.4_{\pm 4.6}$ | $22.0_{\pm 14.7}$ | $31.1_{\pm 3.4}$ | $77.4_{\pm 3.1}$ | $18.1_{\pm 8.0}$ | $51.5_{\pm 41.5}$ | $55.7_{\pm 8.3}$ | $261.3_{\pm 22.5}$ |
| DCDI-G | $\mathbf{0.9}_{\pm \mathbf{1.2}}$ | $3.9_{\pm 6.4}$ | $\mathbf{5.2}_{\pm \mathbf{1.9}}$ | $24.0_{\pm 9.3}$ | $\underline{6.5}_{\pm 5.6}$ | $17.9_{\pm 19.1}$ | $\mathbf{26.8}_{\pm \mathbf{7.0}}$ | $\mathbf{94.4}_{\pm \mathbf{41.5}}$ |
| PACER | $\underline{2.6}_{\pm 1.5}$ | $\underline{1.1}_{\pm 1.4}$ | $7.7_{\pm 3.8}$ | $\underline{19.3}_{\pm 14.5}$ | $\mathbf{4.4}_{\pm \mathbf{2.7}}$ | $\mathbf{5.8}_{\pm \mathbf{4.0}}$ | $\underline{36.8}_{\pm 11.0}$ | $156.6_{\pm 59.8}$ |

*Table 9.* Results for nonlinear non-additive noise dataset with perfect interventions. Lower SHD and SID indicate better performance. Best scores are bolded and second best scores are underlined.

| Method | 10 nodes, $e=1$ | | 10 nodes, $e=4$ | | 20 nodes, $e=1$ | | 20 nodes, $e=4$ | |
|---|---|---|---|---|---|---|---|---|
| | SHD | SID | SHD | SID | SHD | SID | SHD | SID |
| IGSP | $6.6_{\pm3.9}$ | $25.8_{\pm17.9}$ | $31.1_{\pm3.3}$ | $77.1_{\pm5.7}$ | $14.4_{\pm4.8}$ | $63.8_{\pm26.5}$ | $79.7_{\pm8.1}$ | $341.4_{\pm18.1}$ |
| GIES | $6.2_{\pm3.5}$ | $\mathbf{0.9}_{\pm\mathbf{1.5}}$ | $9.5_{\pm3.6}$ | $29.0_{\pm17.7}$ | $12.2_{\pm2.1}$ | $\mathbf{3.4}_{\pm\mathbf{3.2}}$ | $63.8_{\pm11.1}$ | $\underline{124.9}_{\pm36.9}$ |
| CAM | $4.1_{\pm3.8}$ | $2.3_{\pm3.4}$ | $11.3_{\pm4.2}$ | $35.4_{\pm20.8}$ | $\mathbf{4.2}_{\pm\mathbf{2.3}}$ | $\underline{10.9}_{\pm10.3}$ | $106.6_{\pm15.7}$ | $144.2_{\pm51.8}$ |
| DCD | $6.6_{\pm3.5}$ | $18.1_{\pm8.1}$ | $20.6_{\pm3.9}$ | $65.8_{\pm9.9}$ | $9.4_{\pm4.9}$ | $25.6_{\pm16.2}$ | $\underline{28.6}_{\pm6.8}$ | $188.0_{\pm28.7}$ |
| DCDI-G | $\underline{2.1}_{\pm1.5}$ | $4.6_{\pm5.4}$ | $\underline{5.0}_{\pm4.3}$ | $\underline{28.8}_{\pm17.6}$ | $6.4_{\pm3.8}$ | $15.1_{\pm8.0}$ | $\mathbf{12.2}_{\pm\mathbf{2.7}}$ | $\mathbf{96.1}_{\pm\mathbf{18.9}}$ |
| PACER | $\mathbf{1.3}_{\pm\mathbf{1.5}}$ | $\underline{1.9}_{\pm2.5}$ | $\mathbf{4.5}_{\pm\mathbf{2.7}}$ | $\mathbf{18.6}_{\pm\mathbf{10.6}}$ | $\underline{4.2}_{\pm3.4}$ | $13.1_{\pm12.3}$ | $28.9_{\pm7.9}$ | $160.1_{\pm29.3}$ |

## J.3. Intervention robustness over sparsity coefficient

Similar to DCDI, we perform a hyperparameter search over the sparsity coefficient $\lambda$. Although $\lambda$ plays different roles in DCDI and PACER, in both cases it controls the trade-off between model fit and graph sparsity. Following DCDI, we conduct a grid search over 10 values of $\lambda$ for known interventions, using 3 random seeds on the DCDI synthetic dataset (Figure 12, 13, 14, 15).

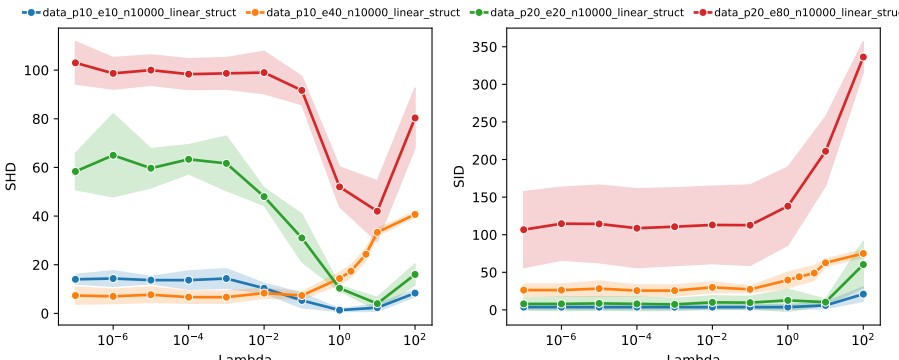

*Figure 12.* Sensitivity of PACER to the sparsity hyperparameter $\lambda$ on linear datasets. SHD and SID are shown for $\lambda$ values ranging from $10^{-7} - 10^2$ for three random seeds.

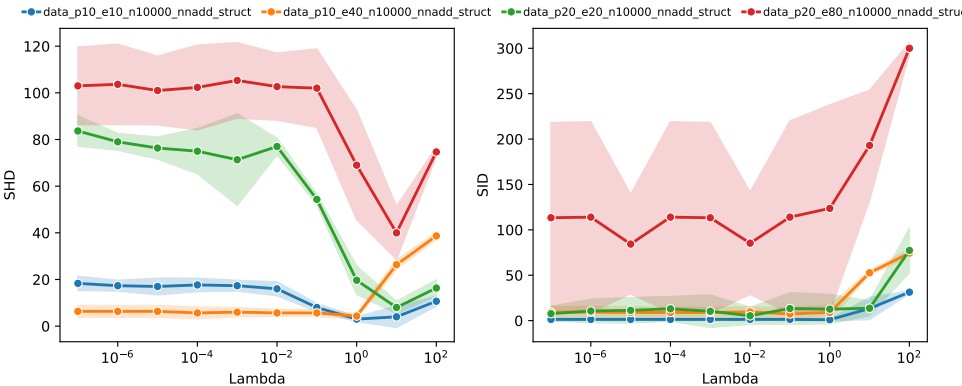

*Figure 13.* Sensitivity of PACER to the sparsity hyperparameter $\lambda$ on additive noise model (ANM) datasets. SHD and SID are shown for $\lambda$ values ranging from $10^{-7} - 10^2$ for three random seeds.

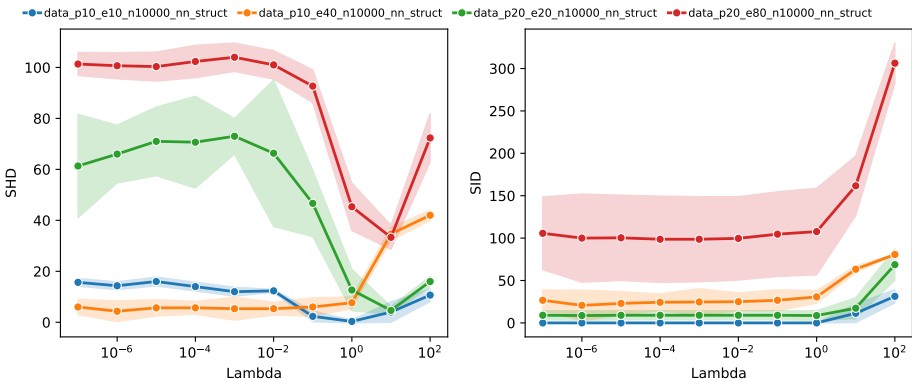

*Figure 14.* Sensitivity of PACER to the sparsity hyperparameter $\lambda$ on nonlinear non-additive (NN) datasets. SHD and SID are shown for $\lambda$ values ranging from $10^{-7} - 10^2$ for three random seeds.

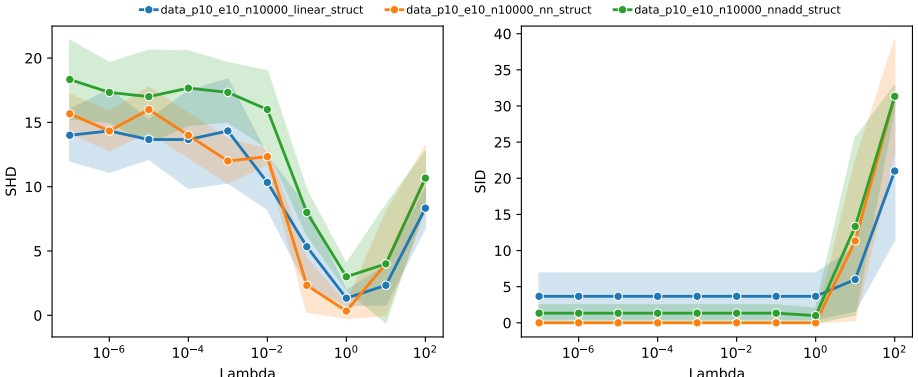

*Figure 15.* Sensitivity of PACER to the sparsity hyperparameter $\lambda$ on different causal mechanism datasets with same number of nodes and edges. SHD and SID are shown for $\lambda$ values ranging from $10^{-7} - 10^2$ for three random seeds.

## J.4. Robustness to imperfect and off-target interventions

We evaluate PACER's robustness under two practically motivated deviations from the perfect-intervention assumption, directly inspired by realistic biological perturbation settings such as CRISPR knockdowns.

### J.4.1. OFF-TARGET INTERVENTIONS

For each interventional sample the intervention target label is randomly mislabeled with probability $p \in \{0.0, 0.1, 0.2, 0.3, 0.5\}$, simulating scenarios in which a fraction of interventions are attributed to the wrong node.

Figure 16 shows that PACER's performance degrades gracefully as $p$ increases. Both SHD and SID remain nearly constant across the tested mislabeling rates, confirming that PACER retains strong causal discovery performance even at moderate off-target rates.

### J.4.2. SOFT (PARTIAL) INTERVENTIONS

We generate soft interventions by interpolating between the fully intervened and observational mechanisms via a blending parameter $\alpha \in [0, 1]$:

$$p_\alpha(X_i \mid \mathrm{pa}(X_i)) = \alpha\, p_{\mathrm{int}}(X_i) + (1 - \alpha)\, p_{\mathrm{obs}}(X_i \mid \mathrm{pa}(X_i)), \tag{35}$$

where $\alpha{=}1$ corresponds to a perfect do-intervention and $\alpha{=}0$ to purely observational data.

Figure 17 shows that PACER maintains robust performance across all tested values of $\alpha$.

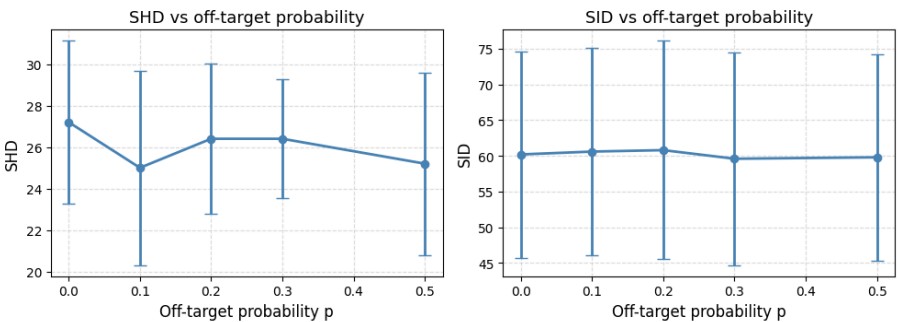

*Figure 16.* Sensitivity of PACER to off-target interventions. SHD and SID are shown for three random seeds.

*Figure 17.* Sensitivity of PACER to imperfect interventions. SHD and SID are shown for three random seeds.

## J.5. Hyperparameter sensitivity analysis

We conduct a thorough sensitivity analysis over all major hyperparameters on a synthetic dataset. We systematically vary batch size, learning rate, number of MC samples $K$, number of layers, and hidden dimension on the linear Gaussian dataset:

- **Hidden dimension.** Increasing hidden dimension improves both SHD and F1, with diminishing returns beyond moderate sizes.

- **Number of layers.** Reducing the number of layers is detrimental, possibly due to limited representational capacity of shallow networks for this data modality.

- **Number of MC samples $K$.** Performance improves and variance decreases as $K$ increases.

- **Batch size and learning rate.** Performance is robust to moderate variations within standard ranges.

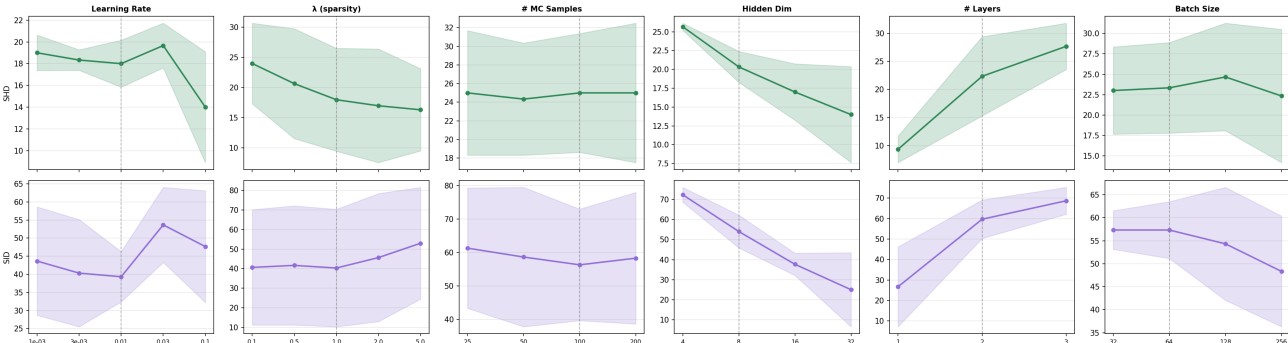

*Figure 18.* Sensitivity of PACER to various hyperparameter settings. SHD and SID are shown for three random seeds.

## K. Comparison with differentiable DAG sampling approach (DP-DAG)

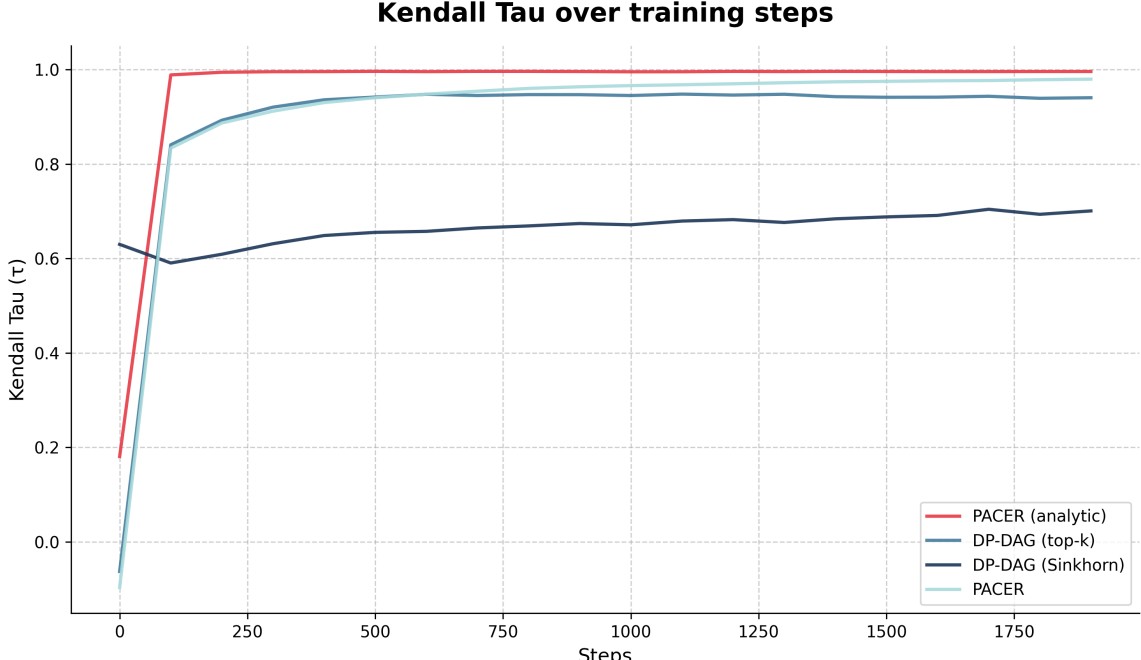

*Figure 19.* Kendall Tau coefficient calculated between ground-truth and inferred partial orderings by training step.

We compare PACER to the differentiable DAG sampling approach of DP-DAG (Charpentier et al., 2022), including both top-k and Sinkhorn-based methods (Charpentier et al., 2022). To isolate the efficacy of the DAG parameterization (independent of specific likelihood modeling or architectural choices) we conduct a controlled experiment targeting the reconstruction of a fully-connected DAG with $d = 500$ nodes. To simulate the uncertainty inherent in real-world causal discovery, we introduce an edge noise probability $\rho = 0.3$. At each training step, a corrupted target is generated where each edge in the true DAG has a 30% probability of being omitted (flipped from 1 to 0). We evaluate performance using the Kendall Tau ($\tau$) correlation to measure topological ordering accuracy. In this setting, all methods optimize a common Mean Squared Error (MSE) objective against a noisy version of the target adjacency matrix at each iteration. Our results (Figure 19) indicate that:

- Both PACER variants outperform DP-DAG. Notably, the analytic version of PACER achieves near-perfect recovery of the ground-truth ordering ($\tau \approx 1.0$), while DP-DAG baselines exhibit slower convergence and lower asymptotic accuracy in this regime.

- PACER is substantially more efficient, offering an order-of-magnitude speedup. The analytic variant is approximately $10\times$ faster than DP-DAG (top-$k$) and $20\times$ faster than DP-DAG (Sinkhorn) per iteration.

- While the REINFORCE-based PACER (using 100 Monte Carlo samples) robustly outperforms the baselines, the analytic variant provides the most stable and rapid convergence, demonstrating the algorithmic advantages of the Bernoulli-Plackett-Luce parameterization for large-scale discovery.

## L. Sachs dataset and benchmark

We use observational (Sachs et al., 2005) data from `causallearn` (Zheng et al., 2024). The dataset contains 7467 measurements across 11 proteins. GS, GES, GIES, IAMB, MMPC, GRaSP, BOSS, and LiNGAM are benchmarked using the implementation of the Causal Discovery Toolbox (Kalainathan et al., 2020) with default parameters. We use the original implementation of NO-TEARS (Zheng et al., 2018). In terms of the interventional scenario, we use the (Sachs et al., 2005) data processed by DCDI (Brouillard et al., 2020), containing 5846 measurements for the same 11 proteins across 6 interventional regimes. We use the IGSP, GIES, CAM, DCDI-G, and DCDI-DSF results from Appendix C1 of DCDI

([Brouillard et al., 2020](#)). For both settings and all experiments throughout the manuscript, we use the default PACER hyperparameters detailed in Section F.

## M. Inferred Sachs network

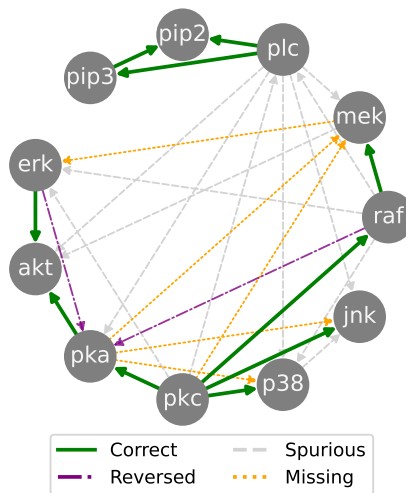

*Figure 20.* Interventional network inferred by PACER on flow cytometry data from ([Sachs et al., 2005](#)). PACER correctly recovers many canonical signaling interactions.

## N. Perturb-CITE-seq dataset

The Perturb-CITE-seq dataset ([Frangieh et al., 2021](#)) comprises gene expression profiles from 218,331 melanoma cells subjected to CRISPR-based perturbations targeting 248 genes. The data was generated to identify gene regulatory programs underlying resistance or sensitivity to T cell-mediated killing, with the goal of uncovering potential therapeutic targets in cancer. We treat the three experimental conditions, control, co-culture, and IFN-$\gamma$ stimulation, as separate datasets.

We apply the same preprocessing pipeline as DCDFG, Appendix F ([Lopez et al., 2022](#)), retaining genes with sufficient signal and constructing interventional training and test splits by holding out all cells corresponding to 20% of the intervention regimes for evaluation. We use the NOTEARS and NOTEARS-LR implementations provided by DCDFG, which extend the original methods to handle perfect interventions and employ a Gaussian likelihood with heterogeneous variances. We use the hyperparameter grids reported in the DCDFG paper. We use the default PACER hyperparameters detailed in Section F.

## O. Perturb-CITE-seq runtime analysis

Across the three Perturb-CITE-seq conditons, PACER is orders of magnitude faster than NOTEARS, NOTEARS-LR and DCDFG. We run all the methods on the same machine using a GPU accelerator (24GB NVIDIA GeForce RTX 3090).

*Table 10.* Average runtime (minutes) and standard deviation across the three Perturb-CITE-seq datasets (Control, Cocult, IFN), each containing approximately 960 variables.

|  | Dataset | | |
| --- | --- | --- | --- |
| Method | Control | Cocult | IFN |
| NOTEARS | $276.7_{\pm 85.6}$ | $161.8_{\pm 46.8}$ | $162.9_{\pm 50.3}$ |
| NOTEARS-LR | $329.0_{\pm 16.0}$ | $314.6_{\pm 123.0}$ | $366.4_{\pm 65.2}$ |
| DCDFG | $850.0_{\pm 253.9}$ | $1031.5_{\pm 311.0}$ | $949.6_{\pm 428.8}$ |
| PACER | $6.4_{\pm 1.6}$ | $7.5_{\pm 1.6}$ | $11.0_{\pm 0.3}$ |

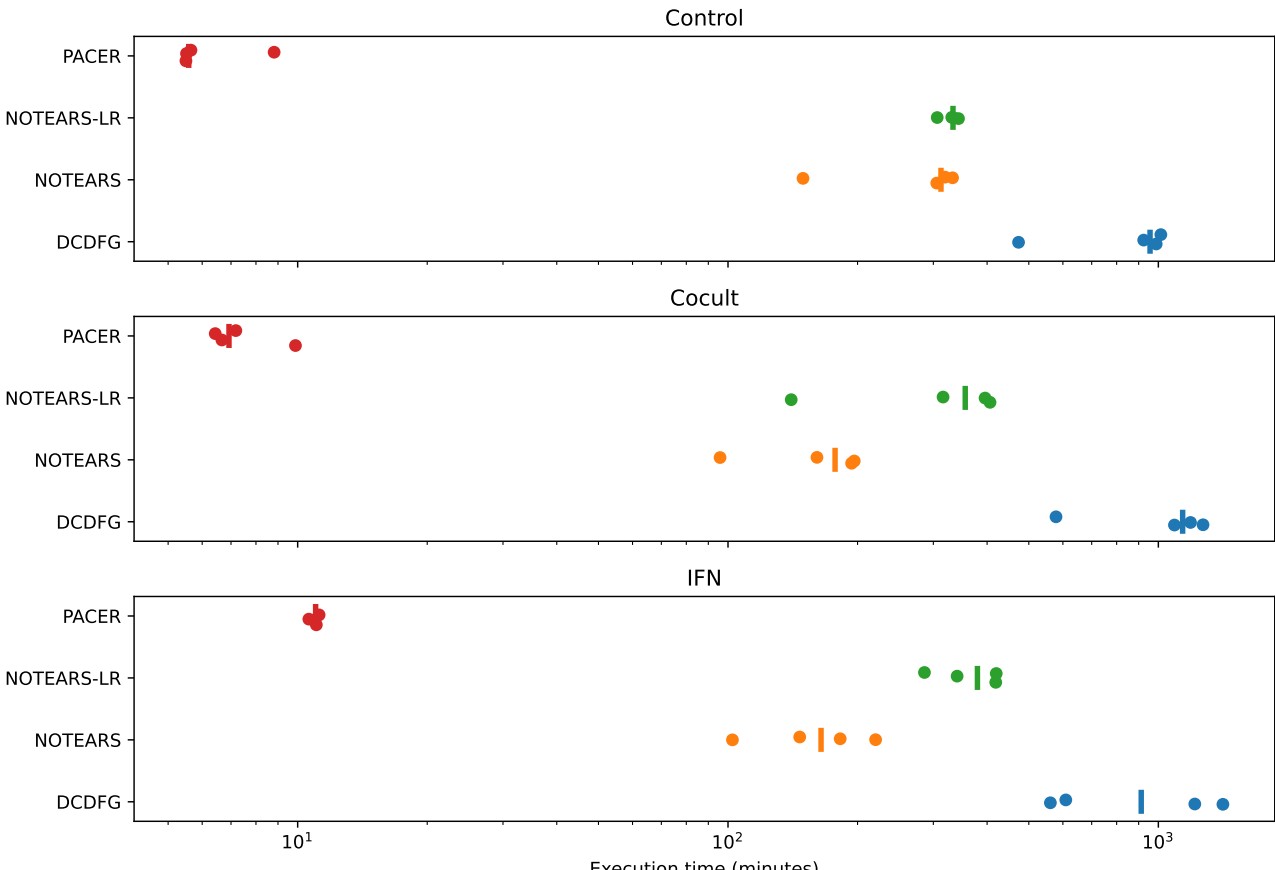

*Figure 21.* Execution times (minutes, log scale) for DCDFG, NOTEARS, NOTEARS-LR and PACER across Control, Cocult and IFN conditions. Points represent runs, lines represent median time.

## P. Extensions to the intervention model

The objective in Equation 3 treats interventions as stochastic perfect interventions with known targets: for each regime $r$, the variables in $\mathcal{I}_r$ are excluded from the likelihood sum, and a single set of conditional parameters $\Omega$ is shared across regimes. We outline here how PACER extends to two settings beyond this default: (i) soft interventions, where the intervened conditional is replaced rather than dropped, and (ii) unknown intervention targets, where $\mathcal{I}_r$ is itself inferred. Both extensions preserve PACER's core design choices.

### P.1. Soft interventions

Under perfect interventions the conditional $p(X_j \mid \mathrm{pa}(X_j))$ for an intervened node is dropped from the likelihood entirely. Under soft interventions (Eberhardt & Scheines, 2007), this conditional is instead *replaced* by a regime specific conditional $p^{(r)}(X_j \mid \mathrm{pa}(X_j))$ that retains the same parental structure but uses different parameters. Yang et al. (2018) characterize the interventional Markov equivalence class under soft interventions with known targets, extending the perfect-intervention interventional Markov equivalence class framework of Hauser & Bühlmann (2012). PACER's identifiability therefore carries over in the sense that the model still recovers a member of an interventional Markov equivalence class, though the specific equivalence class is determined by the intervention type. Perfect interventions can be viewed as the limit of soft interventions where the post-intervention conditional becomes independent of parents

**Modified likelihood.** Concretely, this corresponds to a small modification of the interventional objective $f(M', \Omega)$ in Equation 3. We introduce per-regime conditional parameters $\Omega_j^{(r)}$ for each intervened node $j \in \mathcal{I}_r$, and include those nodes

in the likelihood sum:

$$f_{\text{soft}}(M', \Omega, \{\Omega^{(r)}\}) = \sum_{r=1}^{R} \mathbb{E}_{X \sim P_{\text{data}}^{(r)}} \left[ \sum_{j \notin \mathcal{I}_r} \log p_{\Omega}^j(X_j \mid M'_j, X_{-j}) \ + \ \sum_{j \in \mathcal{I}_r} \log p_{\Omega_j^{(r)}}^j(X_j \mid M'_j, X_{-j}) \right].$$

This places soft interventions naturally between the two regimes already covered by PACER, which include the observational case (no terms dropped, all nodes share $\Omega$) and perfect interventions (intervened-node terms dropped entirely). Only the per-regime conditional parameters and the index sets in the inner sums change.

**Inheritance from the base model.** Crucially, the other components of PACER remain unaffected. The Bernoulli-Plackett-Luce parameterization of the DAG distribution $M(\Theta)$, the acyclicity-by-design property, the REINFORCE estimator and its variance bound (Section 4.3), and the integration of structural priors via $\theta$ and $p_{ij}$ all remain identical. The closed-form expected log-likelihood score for linear-Gaussian mechanisms (Theorem 4.4) also extends to the soft-intervention setting by splitting the inner sum to mirror $f_{\text{soft}}$. Writing $s(\boldsymbol{x}, j; \Omega_j)$ for the per-sample, per-node score from Theorem 4.4 with explicit dependence on the linear-Gaussian parameters $\Omega_j = (b_j, w_{1j}, \ldots, w_{nj}, \sigma_j)$ of node $j$, the soft-intervention expected log-likelihood score reads

$$S_{\text{soft}}(\Theta, \Omega, \{\Omega^{(r)}\}) = -\frac{1}{2} \sum_{r=1}^{R} \mathbb{E}_{\boldsymbol{x} \sim P_{\text{data}}^{(r)}} \left[ \sum_{j \notin \mathcal{I}_r} s(\boldsymbol{x}, j; \Omega_j) \ + \ \sum_{j \in \mathcal{I}_r} s(\boldsymbol{x}, j; \Omega_j^{(r)}) \right],$$

where each $s(\boldsymbol{x}, j; \cdot)$ retains the same structure as in Theorem 4.4, with $b_j$, $w_{ij}$, $\sigma_j$ replaced by $b_j^{(r)}$, $w_{ij}^{(r)}$, $\sigma_j^{(r)}$ when $j \in \mathcal{I}_r$. The Bernoulli-Plackett-Luce expectations $\mathbb{E}[a_{ij}] = p_{ij} \, e^{\theta_i}/(e^{\theta_i} + e^{\theta_j})$ and the parent-pair Plackett-Luce factor $e^{\theta_j}/(e^{\theta_i} + e^{\theta_j} + e^{\theta_k})$ are unchanged across regimes, since the DAG distribution itself does not depend on the intervention type.

**Modeling partial perturbation efficiency.** A further extension could target biological artifacts such as incomplete CRISPRi knockdowns, where a perturbation only partially silences its target across cells. The regime conditional can be modeled as a mixture between the observational mechanism and the soft-intervention mechanism,

$$p_j^{(r,\text{mix})}(X_j \mid \text{pa}(X_j)) = \alpha_j^{(r)} \, p_{\Omega}^j(X_j \mid \text{pa}(X_j)) + (1 - \alpha_j^{(r)}) \, p_{\Omega_j^{(r)}}^j(X_j \mid \text{pa}(X_j)),$$

with a learnable mixing weight $\alpha_j^{(r)} \in [0, 1]$. Setting $\alpha_j^{(r)} = 1$ recovers the observational mechanism (no perturbation effect), while $\alpha_j^{(r)} = 0$ recovers the soft intervention. Unlike the soft-intervention case, this mixture is not linear-Gaussian even when both components are, so Theorem 4.4 no longer applies and training falls back on REINFORCE. We leave a full empirical study of this parameterization to future work.

## P.2. Unknown intervention targets

PACER as presented assumes the intervention sets $\mathcal{I}_r$ are provided. In some settings, however, the targeted variables may be uncertain or only partially known. PACER could in principle be extended to learn intervention targets jointly with the graph structure, following the approach of DCDI (Brouillard et al., 2020). Concretely, one could introduce per-regime binary indicators $\rho_j^{(r)} \in \{0, 1\}$, with $\rho_j^{(r)} = 1$ specifying that node $j$ is targeted in regime $r$, and parameterize each indicator as an independent Bernoulli variable with success probability $\sigma(\beta_j^{(r)})$, analogous to DCDI's relaxation of the intervention matrix $R^{\mathcal{I}}$ via logits $\beta_{kj}$. The interventional likelihood in Equation 3 would be evaluated jointly over $M' \sim M(\Theta)$ and the $\rho_j^{(r)}$, with $\mathcal{I}_r$ effectively replaced by $\{j : \rho_j^{(r)} = 1\}$, and an additional sparsity penalty $\lambda_R \sum_{r,j} \sigma(\beta_j^{(r)})$ would encourage sparse interventional families. Under suitable identifiability assumptions, and provided the observational regime is known a priori, according to Brouillard et al. (2020, Theorem 2), this joint formulation recovers both the interventional Markov equivalence class of the true graph and the true interventional family.

The intervention parameters $\beta_j^{(r)}$ could be optimized jointly with the DAG parameters using the REINFORCE estimator, which is consistent with PACER's existing gradient approach for the Bernoulli-Plackett-Luce parameters, though introducing the additional Bernoulli variables $\rho_j^{(r)}$ would further increase the gradient variance (cf. Section 4.3). Alternatively, a

continuous relaxation such as the straight-through Gumbel estimator (used by DCDI for both the graph and intervention parameters), which trades variance for relaxation bias, could also be applied. Both estimators are conceptually compatible with PACER. We view this as a promising direction with direct connections to existing work and leave a careful empirical comparison to future work.

