# OpenReview forum: "PACER: Acyclic Causal Discovery from Large-scale Interventional Data"
_ICML.cc/2026/Conference — ICML 2026 regular_

### Official Review · Reviewer_mxVp · 2026-02-19

**Soundness:** 3
**Presentation:** 3
**Significance:** 3
**Originality:** 3
**Overall Recommendation:** 5
**Confidence:** 3

**Summary:**

This paper studies the problem of large-scale causal discovery from interventional data. The authors propose an optimization based method named PACER (Perturbation driven Acyclic Causal Edge Recovery). Instead of enforcing acyclicity via soft constraints (as in common existing works), PACER guarantees the acyclic search space by design, by jointly optimizing the topological ordering of the variables and the edges under this ordering. Specifically, the node ordering is parameterized by a Plackett–Luce distribution over permutations, and the edges are pruned by independent Bernoulli edge masks. A likelihood objective is optimized over observational and interventional data. For linear Gaussian setting, a closed-form expression for the expected interventional likelihood is derived. Empirically, PACER matches or exceeds other methods in accuracy and speedup.

**Compliance With Llm Reviewing Policy:**

Affirmed.

**Final Justification:**

I keep my (high) score of acceptance. I cannot score higher mainly due to the lack of identifiability discussion, as also noted by the authors. I acknowledge the authors' efforts in optimization and scalability issues, though.

**Key Questions For Authors:**

see "weaknesses".

**Limitations:**

see "weaknesses".

**Strengths And Weaknesses:**

## Strengths:

1. The acyclicity by design is a clean and principled idea. It avoids the scalability and instability issues of soft DAG constraints in existing differentiable DAG learning methods. It also avoids the known issues of ill-defined likelihood evaluations on cyclic graphs, and post-hoc pruning of them.
2. The use of Plackett–Luce models to represent distributions over topological orders is new to me (though I am not that familiar with the differentiable DAG learning literature). The derivations look coherent to me.
3. In the linear Gaussian setting, the closed form expression for the expected interventional likelihood is derived. This is a strong theoretical results that eliminates unscalable exact search or Monte Carlo sampling, and yields runtime improvements.
4. The experimental results are comprehensive in terms of both the competitors and the benchmark datasets, as well as the ablation studies in the appendix. The scalability up to hundreds or thousands of variables is strong.

## Weaknesses:

1. One major concern I have is the lack of identifiability discussion:
   - This work focuses heavily on the acyclicity enforcement and the related computational advantages, but the identifiability issues are not discussed.
   - Several important questions remain unaddressed, for example,
   - What if the interventions are not sufficient to identify the exact DAG? How is that interventional equivalence class represented in the final output?
   - Does one always need to know the intervention targets, and do the interventions have to be targeted on a singleton variable in each domain?
   - From the optimization perspective, is the solution unique, or can multiple θ,P combinations represent the same DAG distribution?
   - Are there any identifiability guarantees (at the asymptotic level) and the consistency guarantees (at the finite-sample level)?
   - The faithfulness assumption seems not mentioned in the "Assumptions" paragraph. Note that permutation based methods were first proposed to resolve certain faithfulness violation cases. It would be interesting to see which extent of faithfulness assumption is required for this work.
   - Also, it is expected to discuss the trade-offs against conventional permutation based methods (constraint-based or exact search based ones). These methods are usually quite slow. Thus it would be valuable to mention which trade-offs are made for the scalability here.

2. It remains unclear how the non-analytic cases work. Though the paper mentions that in settings other than linear Gaussian, other proper likelihood functions can be used together with REINFORCE, it remains unclear how stable is training in those settings, how many samples are typically required, or how fast/slow PACER with REINFORCE typically works.

---

> ### Author Rebuttal · Authors · 2026-03-31
>
> **Identifiability and equivalence classes.** Please refer to the response to Reviewer vX4S under "Identifiability".
>
> **Intervention assumptions.** PACER assumes known intervention targets and supports multi-variable interventions. PACER could in principle be extended to learn intervention targets jointly with the graph structure (e.g. via latent binary variables analogous to the β variables in DCDI) indicating whether a node is intervened upon in a given regime. These variables could be optimized jointly with the DAG parameters, either via REINFORCE or via continuous relaxations (such as a straight-through Gumbel estimator, similar to DCDI). While both approaches are conceptually compatible with PACER, these approaches introduce additional challenges (e.g., increased variance in gradient estimates or bias from relaxations) and we therefore leave this extension to future work. We will clarify these points in the revised manuscript and that extending the framework to unknown or uncertain intervention targets is a promising direction, with connections to existing approaches such as DCDI.
>
> **Parameter non-identifiability and optimization.** The parameterization is not identifiable: multiple ($θ$,$P$) combinations can induce the same distribution over DAGs. This non-uniqueness arises from several sources (permutation invariance, factorization, and structural non-identifiability). However, this does not pose a problem for optimization itself, as the Plackett-Luce parameters $\theta$ are identifiable up to an additive constant (the $\theta$ lies on an (n-1) dimensional hyperplane). Once this invariance is accounted for, each $\theta$  corresponds to a unique distribution over permutations and so it induces a well-defined distribution over DAGs. As a result, while multiple ($θ$,$P$) combinations may represent equivalent DAG distributions at a global level, the optimization landscape is well-defined and does not suffer from pathological degeneracies beyond standard invariances. We will clarify this point in the revised manuscript.
>
> **Faithfulness.** We assume standard causal sufficiency and (implicit) faithfulness conditions required by likelihood-based methods. PACER does not explicitly address faithfulness violations; we will clarify the assumptions.
>
> **Trade-offs made for scalability.** Classical permutation-based methods guarantee acyclicity but typically incur a computational cost that scales super-exponentially in the number of variables (in the worst case $\mathcal{O}(n!)$), limiting scalability. Continuous optimization methods (e.g. NOTEARS) improve scalability, but require costly constraints and may score cyclic graphs. This improves scalability compared to exact search, but introduces its own challenges: enforcing acyclicity requires evaluating costly constraints, intermediate solutions may correspond to cyclic graphs, and post-hoc thresholding is often required. Compared to classical permutation search, PACER sacrifices exact enumeration in favor of stochastic optimization, but gains substantial scalability (quadratic rather than factorial complexity). Compared to continuous methods, PACER avoids acyclicity constraints and their associated numerical issues, at the cost of introducing stochastic gradient estimation in the general case. We will expand the trade-offs made for scalability with respect to conventional permutation methods.
>
> **REINFORCE stability, variance reduction, and number of samples.** In our experiments, we have found that training with the REINFORCE estimator with a mean baseline is fairly stable. To investigate this, we conducted a synthetic experiment on a 1,000-node complete DAG. We compared 3 approaches: 1) REINFORCE without baseline, 2) REINFORCE with average baseline, and 3) REBAR (Tucker et al., 2017). To evaluate performance, we employed the Kendall-Tau coefficient between the inferred and ground-truth partial variable orderings. Overall, we observed that REINFORCE without baseline performed poorly, while the other 2 variable stabilization techniques attained substantially improved performance (https://ibb.co/t0jpzjg, https://ibb.co/WWbx59B3). As expected, as we increased the number of Monte Carlo samples, the performance also improved, albeit with diminishing returns. Through this synthetic experiment, we identified M=200 Monte Carlo samples as an optimal balance between downstream performance and computational efficiency. Empirically, we did not observe any benefit in using the more sophisticated REBAR technique. We agree this is an important consideration and will include these experiments in the revised manuscript. For a more theoretical answer, please see response to mxVp (lack of space).

---

> > ### Author Rebuttal · Reviewer_mxVp · 2026-04-01
> >
> > Thanks for the response. I keep my (high) score of acceptance. I cannot score higher mainly due to the lack of identifiability discussion, as also noted by the authors. I acknowledge the authors' efforts in optimization and scalability issues, though.

---

> > > ### Author Response · Authors · 2026-04-03
> > >
> > > We thank the Reviewer for their constructive feedback and positive evaluation of our work. We will add an extended identifiability discussion in the revised manuscript.

---

### Official Review · Reviewer_SHVL · 2026-03-10

**Soundness:** 3
**Presentation:** 3
**Significance:** 3
**Originality:** 3
**Overall Recommendation:** 3
**Confidence:** 2

**Summary:**

This paper introduces PACER (Perturbation-driven Acyclic Causal Edge Recovery), a framework for scalable causal discovery from observational and interventional data. The central idea is to parameterize a distribution over DAGs via a joint Bernoulli–Plackett–Luce model, thereby guaranteeing acyclicity by construction rather than enforcing it through soft constraints. This work discuss the explicit restriction of the search space to valid DAGs during optimization, eliminating the need for surrogate acyclicity penalties. They also focus on the scalability and numerical instability of differentiable causal discovery methods when applied to high-dimensional interventional datasets.

**Compliance With Llm Reviewing Policy:**

Affirmed.

**Key Questions For Authors:**

1. How stable is training under the stochastic gradient estimator in large graphs? Are variance reduction strategies beyond a baseline employed?
2. Does the Bernoulli–Plackett–Luce parameterization cover all DAG distributions, or are there representational biases?
3. How would PACER adapt to soft or imperfect interventions where parent distributions are partially modified?
4. How does PACER perform in regimes where the ground-truth graph is moderately dense?
5. Is it possible to extend the exact gradient derivation beyond linear-Gaussian mechanisms?

**Limitations:**

PACER assumes causal sufficiency and perfect interventions, and its analytic scalability benefits are currently restricted to linear-Gaussian mechanisms. In nonlinear regimes, reliance on stochastic gradient estimators may limit stability. Addressing these aspects would further enhance robustness and applicability.

**Strengths And Weaknesses:**

Strengths:
1. The Bernoulli–Plackett–Luce parameterization provides a clean and principled way to restrict optimization to the DAG space. This addresses a longstanding weakness of NOTEARS-style approaches that evaluate cyclic graphs during training.
2. Empirical runtime results convincingly demonstrate improved scaling compared to DCDI and DCDFG, particularly in the thousands-of-variables regime.
3. Theorem 4.3 is technically non-trivial and offers a clear computational advantage in the linear-Gaussian setting. Avoiding Monte Carlo sampling of DAGs in this regime is an elegant contribution.
4. The ability to encode prior biological knowledge via the Bernoulli mask (e.g., TF constraints) is practically useful and well demonstrated in CausalBench.
5. The evaluation spans synthetic, medium-scale (Sachs), and large-scale perturbation datasets, providing evidence across multiple regimes.

Major Weaknesses:
1. In the general nonlinear setting, PACER relies on REINFORCE for gradient estimation. The variance properties and convergence behavior of this estimator in high-dimensional DAG spaces are not thoroughly analyzed.
2.  While acyclicity is guaranteed, restricting exploration to permutation-induced DAGs may implicitly bias the search. It would be helpful to discuss whether this parameterization limits expressivity or introduces optimization bottlenecks in dense graphs.
3. Some baselines (e.g., DCDI-based methods) are run on partitioned graphs due to scalability limits, whereas PACER operates on the full graph. Although this highlights PACER’s scalability, it complicates direct fairness comparisons.
4. Although Appendix G indicates robustness, the dependence of performance on $\lambda$ could be more systematically characterized, especially for dense graphs.
5. The framework assumes stochastic perfect interventions and causal sufficiency. In realistic perturbation experiments (e.g., CRISPR knockdowns), interventions are often imperfect or off-target. The robustness of PACER under such deviations is not evaluated.

Minor Weaknesses:
1. The manuscript emphasizes scalability to thousands of variables, but memory complexity analysis is limited.
2. In the Sachs observational case, PACER does not uniformly dominate across metrics (Table 1), suggesting trade-offs that could be discussed more explicitly.
3. The analytic linear-Gaussian formulation is compelling, but its applicability beyond Gaussian assumptions remains limited.

---

> ### Author Rebuttal · Authors · 2026-03-31
>
> **Variance and convergence (MJ1, Q1).** We reduce the high variance of REINFORCE using standard baseline subtraction, $g_b(b)=(S(b)-B)G(b)$, with B implemented as a running average. It is expressed as $Var(g_b(b)) = E_{b \sim \mathcal{G}} \left[ (S(b) - B)^2 \parallel G(b) \parallel_2^2 \right] - \parallel \nabla_\theta J(\theta) \parallel_2^2$. For the Plackett-Luce model, translation invariance induces a zero-sum constraint on the score: gradients lie in an (n−1)-dimensional subspace. This property can be combined with coordinate-wise bounds to obtain a clean bound $||\nabla_\theta \log p_\theta(b)||^2 = \mathcal{O}(n^2)$. Empirically, training is stable with baseline subtraction, minibatching, and sufficient Monte Carlo samples. (empirical analysis in response to mxVp; lack of space). In the revised manuscript, we will elaborate on the score-function estimator, its variance reduction through baselines, and the structure of the Plackett-Luce gradient.
>
> **Expressivity and biases of Bernoulli–Plackett–Luce (BPL; MJ2, Q2).** This parameterization does not represent all possible distributions, i.e., dependencies between edges that cannot be captured through a shared node ordering or independent edge probabilities are not explicitly modeled. This design is intentional and reflects a trade-off between expressivity and scalability. We will elaborate on the potential optimization bottlenecks in dense graphs, noting that while BPL does not cover all distributions, it provides a flexible approximation that is well-suited for large-scale causal discovery.
>
> **DCDI (MJ3).** We follow the standard protocol of CausalBench, where some methods (e.g., DCDI) operate on partitioned graphs due to scalability constraints. This is a design choice of CausalBench that we follow. However, we  additionally compare all methods on data without partitioning in controlled settings (Fig. 2-4, Table 1). We will acknowledge this limitation of CausalBench and emphasize that full-graph scalability is a key advantage of PACER.
>
> **Sensitivity to λ (MJ4, Q4).** We perform additional experiments using Erdős-Rényi graphs of varying density (i.e. average degree). PACER’s performance is stable across a wide range of λ values (https://ibb.co/S7cjd3P8, https://ibb.co/wF54qTNK), with only minor sensitivity in dense graphs.
>
> **Imperfect and off-target interventions (MJ5, Q3).** We now evaluated robustness to (a) off-target interventions and (b) partial interventions (https://ibb.co/8LTqdHpw, https://ibb.co/KprPSLZF, https://ibb.co/MkqVgc1B).
> For each interventional example, we randomly mislabel the intervention with probability p. PACER’s performance degrades gracefully as off-target probability increases, retaining reasonable performance even at moderate mislabeling rates.
> Using parameter $\alpha$ to blend intervened values with parental influence $\alpha * interventionval + (1 - \alpha) * f(parents)$, PACER remains robust to partial interventions across varying $\alpha$ values.
> Following DCDI, PACER could be extended to accommodate soft interventions via regime-specific conditionals and off-target interventions via mixture models. The entire DAG parameterization would remain unchanged.
>
> **Memory complexity (MI1).** $O(n^2)$ (will include this analysis)
>
> **Metric trade-offs in the Sachs (MI2).** There are trade-offs between different metrics. Some metrics like SHD can be trivially minimized by aggressively reducing the number of edges. We will clarify trade-offs in the revised manuscript and note that PACER’s strength lies in providing balanced performance across multiple metrics.
>
> **Beyond linear-Gaussian mechanisms (MI3, Q5).** We acknowledge this assumption does not always hold. While full empirical validation remains an objective for future work, we have now formulated an objective that extends beyond linear-Gaussian mechanisms. Briefly, we utilize a second-order Taylor expansion of the log-likelihood with respect to the adjacency matrix $M$ around a null graph, i.e. model with no causal relationships, $p_{\Theta}(X_j | 0, X_{-j}) = p_{\Theta}(X_j)$, for each node $j$:
> $$
> \log p_{\Theta}^{j}(X_j | M_j, X_{-j}) \approx \log p_{\Theta}^{j}(X_j|0,X_{-j})
> +\langle \nabla_{M_j} \log p_{\Theta}^{j}(X_j|0,X_{-j}), M_j \rangle
> +\frac12 \langle \nabla_{M_j}^2 \log p_{\Theta}^{j}(X_j|0,X_{-j}) M_j, M_j \rangle.
> $$
> This approach can be viewed as iteratively fitting a Normal distribution to a general log-likelihood and applying our previous result regarding exact expectations. We derive a uniform non-asymptotic bound for this approximation that recovers the exact Normal case when $L = 0$, where $L$ is a bound on the third-derivative of the log-likelihood. Current limitation: requires computing the Hessian of the log-likelihood for each node. This is possible using automatic differentiation, but is less scalable than our REINFORCE estimator. We will include the full derivations of the extended objective.

---

> > ### Author Rebuttal · Reviewer_SHVL · 2026-04-03
> >
> > Thanks for the detailed rebuttal. I appreciate the additional theoretical clarification on the REINFORCE estimator and the empirical analysis on robustness to imperfect and off-target interventions. The discussion on the expressivity of the Bernoulli–Plackett–Luce parameterization is also helpful. However, while variance reduction techniques are discussed, the stability and convergence behavior of the stochastic gradient estimator in large graphs remain unclear in practice. The expressivity limitations of the parameterization are acknowledged, but their impact in dense or complex graph settings is still not fully characterized. Also, aspects such as fairness of comparisons, sensitivity to hyperparameters, and robustness under realistic intervention settings would benefit from more systematic evaluation rather than preliminary results.

---

> > > ### Author Response · Authors · 2026-04-08
> > >
> > > We thank the Reviewer for the thoughtful follow-up. We address the remaining concerns with additional experiments and clarifications that will be included in the revision.
> > >
> > > **Stability and convergence of REINFORCE in large graphs.** We analyzed PACER’s gradient variance in a real large-scale perturbation dataset (Replogle RPE1: [figure](https://ibb.co/Pv4Gfw8M)). We note that:
> > > * The naive REINFORCE estimator exhibits high variance; baseline subtraction and increased Monte Carlo (MC) samples reduce it substantially,
> > > * The variance decreases over training with no instability,
> > > * The analytic variant achieves several orders of magnitude lower variance than stochastic estimators.
> > >
> > > To assess stability independently of modeling choices, we conducted controlled experiments in large graphs (up to 500 nodes) using a likelihood-free objective (squared error to a target DAG). This isolates the effect of the DAG parameterization and optimization. Across runs, we consistently observe:
> > > * Stable convergence of both analytic and REINFORCE estimators ([Kendall τ on partial orderings](https://ibb.co/HpxrqyCy), n=500 nodes),
> > > * Low variance of our REINFORCE estimator ([variance by number of nodes](https://ibb.co/m52rhMqx), [variance by number of Monte Carlo samples](https://ibb.co/dJ3Sj7XR)), in contrast with the high-variance no-baseline estimator ([variance by number of nodes](https://ibb.co/N6ZGN9YL)),
> > > * No divergence or oscillatory behavior even in high dimensions.
> > >
> > > All the above experiments will be added to the revised manuscript, including:
> > > * Detailed theoretical derivation of the variance of our REINFORCE estimator,
> > > * Empirical variance analysis in the large-scale Replogle RPE1 dataset,
> > > * Training and convergence dynamics across steps,
> > > * Variance bands across graphs of different sizes,
> > > * Ablations on the number of Monte Carlo samples.
> > >
> > > **Expressivity, robustness, and behavior in dense graphs.** We performed systematic experiments on Erdős-Rényi graphs of varying density. Our results show:
> > > * The effect of the regularization parameter λ is particularly marked in sparse regimes ([figure](https://ibb.co/wF54qTNK), [table](https://ibb.co/S7cjd3P8)),
> > > * Mild and progressive degradation of graph recovery metrics in denser regimes ([figure](https://ibb.co/HLmQqrgx)),
> > > * No evidence of optimization failure or instability.
> > >
> > > In addition, we performed robustness experiments under realistic intervention deviations:
> > > * Off-target interventions (random mislabeling of intervention targets: [figure](https://ibb.co/KprPSLZF), [table](https://ibb.co/8LTqdHpw)),
> > > * Soft interventions (interpolating between observational and intervened mechanisms: [figure](https://ibb.co/MkqVgc1B), [table](https://ibb.co/8LTqdHpw)).
> > >
> > > In both cases, PACER degrades gracefully and maintains strong performance across a wide range of perturbation strengths.
> > >
> > > In terms of hyperparameter sensitivity:
> > > * We conducted a thorough hyperparameter search over the sparsity coefficient λ across multiple datasets ([linear](https://ibb.co/Fb5RscrK), [nonlinear with additive noise](https://ibb.co/B5Pkptxb), [nonlinear with non-additive noise](https://ibb.co/zTD7kFdZ)) using graphs of different densities. Across all configurations, we note that PACER achieves near-optimal performance around λ=1, which also corresponds to the value employed in the rest of our experiments.
> > > * We extensively explored the influence of all hyperparameters (batch size, learning rate, sparsity, number of Monte Carlo samples, number of layers, hidden dimension) on the linear Gaussian dataset ([figure](https://ibb.co/Tx3PPn9M)). Increasing the hidden dimension improves both metrics, while reducing the number of layers appears to be detrimental (possibly due to the linear nature of this data).
> > >
> > > All these analyses will be added to the revised manuscript, including:
> > > * Performance vs. graph density plots,
> > > * Performance vs regularization hyperparameter λ,
> > > * Performance vs number of Monte Carlo samples,
> > > * Discussion of the Bernoulli-Plackett-Luce factorization and its implications in dense settings,
> > > * Empirical evidence on robustness to imperfect and off-target interventions.
> > >
> > > **Fairness of comparisons.** The use of graph partitioning in CausalBench for methods such as DCDI follows the standard protocol and is not a design choice specific to our work. We attempted to run DCDI on larger partitions and on the full graph; however, this led to out-of-memory errors and numerical instability. These experiments were conducted using the same DCDI implementation and standard evaluation settings as in CausalBench.
> > >
> > > We will revise the manuscript to:
> > > * Clearly separate benchmark-driven comparisons (i.e., CausalBench) from controlled experiments,
> > > * Emphasize settings where all methods operate on the same data without partitioning,
> > > * Explicitly discuss the limitations of benchmark protocols,
> > > * Clearly describe the benchmarks, metrics, and hyperparameters for all our experiments and baselines.

---

### Official Review · Reviewer_J597 · 2026-03-10

**Soundness:** 4
**Presentation:** 4
**Significance:** 3
**Originality:** 4
**Overall Recommendation:** 5
**Confidence:** 3

**Summary:**

The paper introduces a new method to learn the structure of DAGs. Compared with the traditional method, the proposed method considers the distribution over DAGs and the objective function is defined through DAGs sampled from this distribution. This strategy enables an optimization over the DAG space directly, thereby avoiding a regularization term to enforce DAGs. Actually, as pointed out by the authors, the traditional regularization is achieve through soft acyclicity constraint, which therefore leads to optimization over invalid
cyclic graphs, numerical instability, and limited scalability. In contrast, the proposed method is free from this soft constraint and is easily scalable.

**Compliance With Llm Reviewing Policy:**

Affirmed.

**Key Questions For Authors:**

I am wondering whether you need to assume that the conditional density of $X_j$ given $M_j^{\prime}, X_{-j]$ needs to be correctly specified in your synthetic data. If it is not correctly specified, would this lead to any undesirable results?

In real-world applications, any guidelines on choosing this conditional density? Are there any more robust methods for the specification of the density compared with the parametric ones mentioned in the paper?

**Limitations:**

Yes

**Strengths And Weaknesses:**

The proposed method is clearly presented, and is supported with comprehensive empirical studies.
Compared with traditional methods, the proposed method directly sampled from the distribution over DAGS, thus ensuring the solution must be a valid DAG. This merit is clearly presented in the paper. Moreover, the paper provides a special case, where the the conditional distribution is Gaussian, to demonstrate that the method can be efficiently implemented without sampling. This may provide some theoretical guarantees. However, a more general discussion might be needed regarding the sampling DAGS since in general, we can hardly which parametric densities are suitable for the conditional density. This may be a potential limitation of the current framework.  In particular, is it possible to extend the current framework to allow for non-parametric conditional densities?

---

> ### Author Rebuttal · Authors · 2026-03-31
>
> **On model misspecification.** Our framework, like most likelihood-based causal discovery methods, does not require the conditional densities to be perfectly specified, but performance can degrade under model misspecification. In particular, if the assumed parametric family (e.g., linear-Gaussian) deviates significantly from the true data-generating process, the learned structure may favor graphs that better fit the misspecified likelihood rather than the true causal graph. This is a standard limitation shared by approaches such as NOTEARS and related methods.
>
> That said, PACER is flexible in the choice of conditional models. In practice, using more expressive function classes (e.g., nonlinear neural networks or normalizing flows) can mitigate misspecification and improve robustness. We will clarify this point in the revised manuscript.
>
> **On guidelines for choosing conditional densities.** In real-world applications, the choice of conditional density should be guided by domain knowledge and the expected data characteristics:
> Linear-Gaussian models are often effective in high-dimensional settings due to their simplicity and stability.
> Nonlinear parametric models (e.g., neural networks) are preferable when strong nonlinearities are expected.
> In settings where we have knowledge about the target distribution, PACER allows using custom likelihood functions. For example, when modelling single-cell data (e.g. scRNA-seq read counts) one may choose to use a (zero-inflated) negative binomial likelihood, as done commonly in existing probabilistic approaches for modelling single-cell data (e.g. scVI; Lopez, Romain, et al. "Deep generative modeling for single-cell transcriptomics." Nature methods 15.12 (2018): 1053-1058.).
> In settings with complex or unknown noise distributions, more flexible likelihoods (e.g., heteroscedastic models or flow-based densities) may be beneficial.
>
> Importantly, PACER is agnostic to the specific choice of conditional density and can accommodate a wide range of parametric models within the same framework.
>
> **On non-parametric extensions.** We thank the Reviewer for this insightful suggestion. The Reviewer is correct that the current implementation focuses on parametric densities for computational efficiency. However, the PACER framework is modular by design: the distribution over DAGs (the structural part) is mathematically decoupled from the conditional density model (the functional part). To extend PACER to non-parametric settings, one could for example replace the Gaussian log-likelihood with:
> Spline-based or Kernel-based estimators: Which avoid global parametric assumptions.
> Conditional Normalizing Flows (CNFs): These allow for highly flexible, non-parametric-like density estimation while remaining fully differentiable, fitting naturally into our Sinkhorn-based optimization.
>
> While a fully non-parametric approach (e.g., using Kernel Density Estimation) would increase the computational complexity, the core 'DAG-sampling' logic of PACER would remain unchanged. We will include a discussion regarding this possible extension and the inherent trade-offs between model flexibility and statistical efficiency in high-dimensional regimes.
>
> Following the Reviewer’s inquiries, we will expand the discussion in the paper to better highlight these trade-offs and position PACER as a flexible framework that can incorporate increasingly expressive conditional models.

---

> > ### Author Rebuttal · Reviewer_J597 · 2026-04-03
> >
> > Using the nonparametric methods may suffer from the curse of dimensionality, though it can partially address the issue of model misspecification. Also, I believe a sensitivity analysis for the choice is desirable to see the impact of parametric specifications.

---

> > > ### Author Response · Authors · 2026-04-08
> > >
> > > We thank the Reviewer for this insightful comment. We agree that nonparametric methods are appealing as they can mitigate model misspecification. As noted by the Reviewer, one potential challenge is that they may be difficult to scale to the high-dimensional regimes that PACER targets. Nonetheless, we acknowledge that this is a promising direction for future work and will highlight it in our revised manuscript.

---

### Official Review · Reviewer_vX4S · 2026-03-11

**Soundness:** 2
**Presentation:** 3
**Significance:** 2
**Originality:** 1
**Overall Recommendation:** 5
**Confidence:** 4

**Summary:**

The authors focus on causal discovery with a mix of interventional and observational data. Whereas previous methods rely on continuous optimisation that regularise to obtain an acyclic causal graph, the authors propose parametrising the space of DAGs directly. This ensures that sampled graphs are acyclic throughout optimisation. The authors propose multiple causal models, and test their method on synthetic and real benchmarks. They show that their method recovers scales better than baselines, both in terms of compute and performance.

**Compliance With Llm Reviewing Policy:**

Affirmed.

**Final Justification:**

Happy to increase my score based on the proposed changes by the authors.

I'm not sure I understand the figure provided in the latest response, it would be nice to see actual performance in terms of the causal discovery metrics in the paper. These kinds of experiments would convince a reader that this method is worth applying.

**Key Questions For Authors:**

Q1. Why is SDCD not included in all the experiments?

Q2. Although the runtime is interesting, what is the actual computational cost in terms of the sample size and variables?

Q3. Could you compare and contrast against other approaches that use permutation and lower triangular representation of DAGs?

**Limitations:**

Yes

**Strengths And Weaknesses:**

**Soundness:** The paper makes certain claims that do not seem well supported:
1. L103: They claim that baselines provide 'limited uncertainty quantification' in contrast to their method. Having a distribution over graphs is not sufficient for this, and no experiment is done to show that their method provides any meaningful uncertainty. For example, consider a non-identifiable case with X->Y and X<-Y. Here, as their functions are point estimates, I suspect that the distribution over graphs will not represent both graphs with equal probability. For each graph, the function must change. This is the reason in fully Bayesian approaches that functional distributions are dependent on the graph [1].
2. Figure 2 and Figure 5 show that SDCD outperforms their method in every metric. However, SDCD is missing in their experiments in Table 1 and Table 2. As SDCD is a variant of the continuous optimisation method with a regulariser, this weakens their argument slightly. I don't see a discussion of this method or why it was not included in all experiments (as it can handle both observational and interventional data). In the SDCD paper they do also scale up to thousands of variables (Figure 8 in their paper).
3. The correctness of the Sachs ground truth graph is disputed, with multiple works positing a different ground truth graph. For example, see 5.8 of [2]. Which graph

Claims that are well supported:
1. Their method does have better runtime. However, I don't see a direct comparison of the computational cost (in terms fo number of nodes and samples) of their method along with competing methods.


**Presentation:** The paper is well written and the ideas are easy to follow. Its slightly unclear which model instantiation is used in each experiment - this could be signposted a little better.

**Significance:** The paper addresses scalable causal discovery methods which have an impact on scientific discovery. Although the paper does not discuss any identifiability issues. For example, if the interventions are not sufficient to recover the complete DAG, what do the authors propose doing then? The answer provided will surely be incorrect in that case.

**Originality:** The problem on parametrising DAGs using lower (upper) triangular matrices and permutation matrices is not a new idea, and there are signiifcant works missing. For example, [3,4,5,6] all use variants of this idea.



[1] Toth, Christian, et al. "Active bayesian causal inference." _Advances in Neural Information Processing Systems_ 35 (2022): 16261-16275.

[2] Mooij, Joris M., Sara Magliacane, and Tom Claassen. "Joint causal inference from multiple contexts." _Journal of machine learning research_ 21.99 (2020): 1-108.

[3] Charpentier, Bertrand, Simon Kibler, and Stephan Günnemann. "Differentiable dag sampling." _arXiv preprint arXiv:2203.08509_ (2022).

[4] Cundy, Chris, Aditya Grover, and Stefano Ermon. "Bcd nets: Scalable variational approaches for bayesian causal discovery." _Advances in Neural Information Processing Systems_ 34 (2021): 7095-7110.

[5] Annadani, Yashas, et al. "Bayesdag: Gradient-based posterior inference for causal discovery." _Advances in Neural Information Processing Systems_ 36 (2023): 1738-1763.

[6] Dhir, Anish, et al. "A Meta-Learning Approach to Bayesian Causal Discovery." _The Thirteenth International Conference on Learning Representations_.

---

> ### Author Rebuttal · Authors · 2026-03-31
>
> **Uncertainty quantification.** In related work, we mentioned that existing methods provide limited uncertainty quantification as the Reviewer highlighted. Our intention was not to claim calibrated Bayesian uncertainty, but rather that PACER represents structural variability via a distribution over DAGs, unlike methods that return a single graph. We agree that this does not guarantee calibrated uncertainty in a Bayesian sense. As the Reviewer correctly points out, functional parameters are shared across structures. Fully capturing uncertainty over causal structures would require graph-specific parameterizations, as in Bayesian approaches, e.g. [1]. We will revise our manuscript to clarify this distinction.
>
> **SDCD.** We agree that SDCD is an important baseline and have now included it in Tables 1 and 2. On the Sachs dataset ([Table 1](https://ibb.co/CKKMXjg0)), PACER outperforms SDCD in SHD, FDR, TPR, and F1 score, while SDCD achieves slightly better SID (second-best). On CausalBench ([Table 2](https://ibb.co/Lzkc8H4Q)), all flavours of PACER outperform SDCD in terms of the Wasserstein metric across both datasets (RPE1 and K562), while SDCD attains the lowest FOR score in K562. We note that PACER remains substantially faster in these large-scale regimes. We will update the manuscript accordingly and expand the discussion to better contextualize the strengths and limitations of SDCD relative to PACER across different regimes.
>
> **Correctness of Sachs.** We used the consensus network from Sachs et al. (Fig. 11). The Sachs data has been employed in multiple previous works (including [3], [4], and [5]) and the reference [2] provided by the Reviewer is very insightful in this regard. The paper notes that ‘in the literature regarding this [Raf/Mek/Erk] pathway, it is often suggested that there should be a feedback loop back from Erk to Raf’. We did notice that PACER infers a directed edge from Raf to Erk (Figure 11) that is not present in the consensus network but appears consistent with [2] as well as more recent studies of the Raf/Mek/Erk signaling cascade (Ullah, Rahim, et al. 2022.). Indeed, Ullah et al. report negative feedback phosphorylation between Raf and Erk, a loop that is not captured in the consensus network and that may compromise its completeness. We will clarify that we used the consensus network from Sachs et al., acknowledge that its correctness is disputed, and cite the reference provided by the Reviewer.
>
> **Originality, related work, and computational cost.** We agree that modelling DAGs via permutations and triangular structures is not new and we do not claim this as our contribution. Rather, PACER builds on this general order-based view of DAG parameterizations, explored in prior work [3–6], to introduce a fundamentally different probabilistic formulation and optimization strategy. Our contribution lies in:
> * Probabilistic model over permutations and edges via the Bernoulli–Plackett–Luce model, a conceptually and computationally distinct approach to DAG modeling.
> * Unlike prior permutation-based approaches, which primarily focus on observational settings, PACER is explicitly designed to handle interventional data within a single likelihood-based objective.
> * PACER admits an exact expression for the expected objective under linear-Gaussian mechanisms, eliminating the need for Monte Carlo sampling. To our knowledge, this form of exact expectation over graph distributions has not been explored in prior permutation-based methods.
> * The combination of the above design choices enables PACER to scale to hundreds or thousands of variables, whereas related methods rely on O(n³) permutation operators (e.g., Sinkhorn/Hungarian) or MCMC-based inference, which can limit scalability in practice.
>
> We summarize the main differences between PACER and these existing approaches  [3–6] and compare computational cost in [this table](https://ibb.co/4k0K5MS). We will revise the manuscript to discuss these differences.
>
> **Identifiability.** We agree that the causal structure is generally not identifiable without sufficiently informative interventions, i.e., to restrict the interventional Markov equivalence class to a single graph.  In these settings, PACER recovers high-scoring structures that are consistent with the data, rather than the true DAG. Importantly, PACER does not resolve identifiability limitations inherent to the data; instead, it provides a scalable framework for exploring plausible causal structures under these constraints. In the revised manuscript we will advocate interpreting the output as an element from the set of plausible structures rather than a single definitive graph and, where possible, leveraging additional domain knowledge or designing more informative interventions to further reduce ambiguity. We will also include this as a limitation of our approach as well as other existing methods as no method can reliably recover the true underlying structure in such a case.

---

> > ### Author Rebuttal · Reviewer_vX4S · 2026-04-02
> >
> > After reading the authors' response, most of my concerns have been adequately addressed.
> >
> > To summarise, I feel the paper will benefit from (and it seems the authors agree):
> > - Dialing down claims about uncertainty quantification: Unless there are substantial experiments to back this up. If the distribution over graphs only returns one element of an MEC, as the functions can only represent one of them, this can be misleading for a reader.
> > - Discussion on identifiability: Explicitly stating that the method returns a member from the Interventional-MEC (and citing the relevant paper). Its important to be upfront about this so that the reader is not confused. I can imagine a method like this being very useful as a bootstrap posterior representing the I-MEC.
> >
> > Finally, as the main contribution is the Plackett-Luce model for parametrising DAGs, it would be very useful and convincing if there was a direct comparison against other methods that represent DAGs with permutation+Bernoulli. Particularly relevant is the DP-DAG way of parametrising DAGs, or the extension in Bayes-DAG. It would be very convincing for the reader if an ablation was done where the same likelihood model (with handling interventions) is used and the parametrisation of the DAG is changed. This would clearly show the benefit of the proposed parametrisation of the DAG, controlling for other aspects of the model.

---

> > > ### Author Response · Authors · 2026-04-03
> > >
> > > We thank the Reviewer for their thoughtful reviews and constructive criticism. In the revised manuscript, we will explicitly state that, while PACER represents structural variability via a distribution over DAGs, it _does not_ provide calibrated Bayesian uncertainty. We also commit to being upfront about the identifiability challenges, stating clearly that the method returns a member from the Interventional-MEC (citing for example: Alain Hauser and Peter Bühlmann, _Characterization and Greedy Learning of Interventional Markov Equivalence Classes of Directed Acyclic Graphs_, 2011).  We will also include the Reviewer's insightful observation that PACER could serve as a computationally efficient "bootstrap posterior representing the I-MEC".
> > >
> > > In terms of comparing our method to existing approaches that represent DAGs with permutations and Bernoulli probabilities, we will include detailed qualitative and quantitative comparisons of the methods. To directly isolate the effect of the DAG parameterization (independent of likelihood modeling), we have now conducted a controlled experiment where all methods optimize the same objective: reconstruction of a target fully-connected DAG (n=500 nodes) under an MSE loss. Each method is optimized against the same noisy version of the target adjacency at each iteration. As shown in [this figure](https://ibb.co/DHf1Z2Jb), the REINFORCE-based PACER baseline performs better than DP-DAG (top-k). Notably, the analytic version of PACER converges the fastest, achieves near-perfect recovery of the optimal DAG, and is the most efficient method by a large margin (~10x faster than DP-DAG top-k, ~20x faster than DP-DAG Sinkhorn). We hope that this controlled experiment, combined with our clarified theoretical scope, addresses your concerns and demonstrates the practical and algorithmic advantages of the Bernoulli-Plackett-Luce parameterization for large-scale causal discovery.
> > >
> > > As per ICML policy, we are unfortunately unable to continue the rebuttal discussion beyond two answers. We hope that our responses have adequately addressed the Reviewer's remaining concerns.

---

### Decision · Program_Chairs · 2026-04-30

**Decision:**

Accept (regular)

**Comment:**

The reviewers aknowledge the soundness, novelty and usefulness of the methodology for large scale inference of DAG causal models based on interventional data. Main concerns raised during the discuss have been addressed by the authors with explanations and additional experiments. I therefore recommend acceptance.